



# An 18-year climatology of derechos in Germany

Christoph P. Gatzen[1], Andreas H. Fink[2], David M. Schultz[3], and Joaquim G. Pinto[2]

[1]Institut für Meteorologie, Freie Universität Berlin, Berlin, Germany

[2]Institute of Meteorology and Climate Research - Department Troposphere Research, Karlsruhe Institute of Technology, Karlsruhe, Germany

[3] Center for Atmospheric Science, School of Earth and Environmental Sciences, University of Manchester, Manchester, United Kingdom

**Correspondence:** Christoph Gatzen (gatzen@met.fu-berlin.de)

**Abstract.**

Derechos are high-impact convective wind events that can cause fatalities and widespread losses. In this study, 40 derechos affecting Germany between 1997 and 2014 are analysed to estimate the derecho risk. Similar to the United States, Germany is affected by two derecho types. The first derecho type forms in south-westerly 500-hPa flow downstream of intense west-European troughs and accounts for 22 of the 40 derechos. These derechos are named warm-season type due to their peak occurrence in June and July. Warm-season type derechos frequently start over southwestern Germany in the afternoon and move either eastward along the Alpine forelands or north-eastward across southern central Germany. Only one warm-season derecho moved across the North Sea and one moved across the Baltic Sea in the 18-year period. Proximity soundings of German warm-season type derechos indicate strong deep-layer vertical wind shear with a median of 20 ms$^{-1}$ 0–6-km shear and mixed-layer Convective Available Potential Energy (mixed-layer CAPE) between 20 and 2600 Jkg$^{-1}$ with a median around 500 Jkg$^{-1}$. The second derecho type forms in north-westerly 500-hPa flow and accounts for 18 of the 40 derechos. These derechos form in strong north-westerly flow, frequently in association with mid-tropospheric PV intrusions. They are named cold-season type because they are associated with a secondary peak from December to February. Cold-season type derechos start over or close to the North Sea and primarily affect north and central Germany; their start time is not strongly related to the peak of diurnal heating. Proximity soundings indicate high-shear–low-CAPE environments with a median 0–6-km shear of 35 ms$^{-1}$ and a median mixed-layer CAPE of 3 Jkg$^{-1}$. Environmental CAPE is zero in almost half of cold-season type proximity soundings. Fifteen warm-season type and nine cold-season type derechos had wind gusts reaching 33 ms$^{-1}$ in at least at three locations. Although warm-season derechos are more frequent, the path length of cold-season type derechos is on average 1.4 times longer. Thus, these two types of German derechos are likely to have similar impacts.

## 1 Introduction

Convective wind events can produce high losses and fatalities in Germany. One example is the Pentecost storm in 2014 (Mathias et al., 2017) with six fatalities in the region of Düsseldorf in north-western Germany and high impacts in particular to the





railway network. Trains were stopped due to blocked tracks and trees were blown down and hit overhead power lines, severing connections (Die Welt, 2014). Another example is the convective windstorm that killed eight in Berlin and Brandenburg on 10 July 2002 (Gatzen, 2004). The high number of casualties is caused by the sudden and frequently unexpected occurrence of severe convective storms due to their rapid development (Doswell, 2001), and there are indications that people caught outdoors

are not well-prepared. Open-air events are at risk in particular as demonstrated by incidents such as on 06 July 2001 near Strasbourg, France, when trees fell on tents where visitors took cover, killing 12 (European Severe Weather Database (ESWD), CNN, 2001; Dotzek et al., 2009) and at the Pukkelpop open-air festival in Hasselt, Belgium, when estimated wind gusts of 47 ms$^{-1}$ during the 18 August 2011 storm caused five fatalities due to the collapse of tents and parts of the stage (BBC, 2011). A common characteristic of such events is that they travel over large distances of several hundred kilometers, which greatly

increases their potential impact due to the large area affected.

Data from the European Severe Storms Laboratory (ESSL) can be used to determine the impact of large-scale convective wind events. In 2017, there were 131 fatalities and 783 injuries due to severe weather (including severe winds, heavy rain, tornadoes, and large hail, ESSL, 2018a, b). Of these, 41 fatalities and 456 injuries were caused by the most significant 12

large-scale convective windstorms (ESSL, 2018c, d). The high potential for fatalities and substantial economic losses due to widespread high-impact weather in combination with their rapid development make these convective windstorms among the most challenging forecast situations for weather services and their forecasters.

Similar convective windstorms are known in the United States, where the most intense are called *derechos* (after Hinrichs,

1888) (Johns and Hirt, 1987). To classify derechos based on severe wind gust reports and radar reflectivity data, Johns and Hirt (1987) require that

– deep moist convection is associated with concentrated wind occurrence over a path that extends over at least 400 km along the major length axis. Wind gusts must exceed 26 ms$^{-1}$ as indicated by measurements and/or damage reports,

– associated wind gusts can be related to the same event, so that a chronological progression of singular wind swaths or a

series of wind swaths is indicated,

– there must be at least 3 reports of wind gusts of at least 33 ms$^{-1}$ (and/or related wind damage) separated by 64 km or more within the area of severe wind gusts, and

– all wind reports have to occur within 3h of the other wind reports of the same derecho event.

For the United States, Ashley and Mote (2005) estimate that the derecho hazard potential can be as high as that from trop-

ical storms and tornadoes except for the highest impact tornadoes and hurricanes. Up to four derechos in three years can be expected for 200 km x 200 km grid boxes in the most affected region south-west of the Great Lakes (Coniglio and Stensrud, 2004). Derechos form in association with mesoscale convective systems that develop bow echoes on plan-view radar displays (Fujita, 1978). Bow echoes indicate strong flow from the rear to the front of a convective system, a so-called rear-inflow jet (e.g.



Smull and Houze, 1987; Przybylinski, 1995). The development of rear-inflow jets is fostered by persistent convection initiation along the gust front of a convective system (Mahoney et al., 2009). Studies on the environmental conditions of convective storms have shown that a combination of high Convective Available Potential Energy (CAPE) and strong vertical wind shear is one environment that can support this upscale growth (e.g. Rotunno et al., 1988; Corfidi, 2003; Klimowski et al., 2003). In particular, a combination of strong shear and high CAPE can occur during May to July in the United States when derechos commonly occur (Ashley and Mote, 2005). The environment may become so favourable (as during heat waves) that derechos may occur in clusters of several events called *derecho families* during the same large-scale flow situation (Bentley and Sparks, 2003; Ashley et al., 2004). However, derechos can also develop when instability is weak (Johns, 1993; Burke and Schultz, 2004). Instead of high shear and high CAPE, these derechos form in situations with high shear and low CAPE (Sherburn and Parker, 2014) called high-shear–low-CAPE environments (e.g. Evans, 2010; King and Parker, 2014).

In contrast to the United States, the *derecho* definition is still not commonly used in Europe although several events have been classified in the past decades (Gatzen, 2004; Punkka et al., 2006; López, 2007; Gatzen et al., 2011; Pistotnik et al., 2011; Púčik et al., 2011; Simon et al., 2011; Hamid, 2012; Celiński-Mysław and Matuszko, 2014; Toll et al., 2015; Gospodinov et al., 2015; Mathias et al., 2018). Four of these derechos affected large portions of Germany (Gatzen, 2004; Gatzen et al., 2011; Pistotnik et al., 2011; Mathias et al., 2018). Before 2004, German convective windstorms were not classified as derechos, but there are indications of potential derechos in the previous literature (e.g. Köppen, 1882, 1896; Faust, 1948; Kurz, 1993; Haase-Straub et al., 1997; Kaltenböck, 2004). The low number of classified events does not allow for a risk estimate of derechos in Germany, because their characteristics (e.g. the spatio-temporal distribution, typical environmental conditions, large-scale flow patterns) are not known.

Based on the case studies listed above, European derechos possess some similar characteristics to derechos in the United States. For example, they can be divided into a "warm-season type" and a "cold-season type". The majority of European derecho case studies can be attributed to the warm-season type. These are frequently associated with rather high CAPE (Púčik et al., 2011; Hamid, 2012; Celiński-Mysław and Matuszko, 2014) compared to the majority of severe convective wind environments (Púčik et al., 2015). Four case studies of the warm-season derecho type also mention strong vertical wind shear (Gatzen, 2004; Púčik et al., 2011; Hamid, 2012; Celiński-Mysław and Matuszko, 2014). Publications with radar images describe sustained convection initiation along or ahead of a gust front (Gatzen, 2004; Celiński-Mysław and Matuszko, 2014), bow echoes (Gatzen, 2004; Púčik et al., 2011; Hamid, 2012; Celiński-Mysław and Matuszko, 2014), and rear-inflow jets (Gatzen, 2004). Finally, the large-scale flow was frequently from the south-west (e.g. Gatzen, 2004).

Next to the warm-season type derechos described above, three case studies refer to cold-season type derechos (Gatzen et al., 2011; Pistotnik et al., 2011; Mathias et al., 2018). In contrast to the examples of German warm-season type derechos, German cold-season type derechos form within north-westerly large-scale flow (Gatzen et al., 2011; Pistotnik et al., 2011; Mathias et al., 2018). In particular, Pistotnik et al. (2011) describe a mid-troposphere PV intrusion in one of the cold-season type





derechos. Cold-season type derechos form along narrow cold-frontal rainbands in an environment similar to high-shear–low-CAPE situations described for the United States (e.g. Burke and Schultz, 2004; van den Broeke et al., 2005). Classification of cold-season type derechos can be difficult because strong synoptic-scale winds frequently overlap with convective gusts (e.g. van den Broeke et al., 2005).

Given that only a few European derechos have been analysed, in particular of the cold-season type, it is not clear if the described characteristics are typical for German derechos and thus can be useful in operational forecasting. The purpose of this study is to contribute to European convective storms research and present a derecho climatology for Germany. Creating accurate derecho climatologies is laborious due to manual identification, and this work goes beyond the climatology of warm-

season wind gusts presented in Gatzen (2013). Our work investigates how often derechos occur across Germany, how intense they are, and which parts of the country are most affected. Another main research question is to determine the differences between derechos that occur in the warm and cold seasons, specifically differences in the large-scale flow patterns and the local thermodynamic environment as given by observational data such as rawinsondes. The benefit of this analysis to support operational weather services in potential derecho situations is assessed. Moreover, results are compared to the derecho climatology

of the United States, for which more information is available.

Section 2 introduces the data we used and explains the methods to identify and analyse derechos in Germany. Section 3 presents our results and interpretation of the German derecho climatology and compares it with the derecho climatology of the United States. This section also aids forecasters to support them in identifying potential derecho situations. Finally, section 4

provides the conclusions and an outlook to future work.

## 2    Data and Methods

For the derecho classification, we used maximum wind gusts together with radar data. Maximum wind gusts are the highest 3-seconds averaged wind speed measurements within an hour taken of the German Weather Service (DWD) surface station net-

work. Additionally, 10-minute maximum wind-gust measurements were obtained from the Meteomedia/MeteoGroup network. Wind-measurement sites increased from 260 to more than 1000 sites between 1997 and 2011. This corresponds to a density between 0.7 and nearly three measurement points per 1000 $km^2$ across Germany (approximately 360,000 $km^2$), with an average station spacing between 10 and 20 km. Beyond the borders of Germany, internationally distributed SYNOP and other data such as data from buoys and ships sometimes allowed for following derecho tracks farther into neighbouring countries

and across the North Sea. We used data from the radar network of the German Weather Service, including radar displays in 15-minute intervals, showing radar reflectivity of the lowest elevation scans. This data was interpolated on a Cartesian grid with a horizontal grid spacing of 2 km, a vertical grid spacing of 1 km, and seven intensity thresholds (Schreiber, 2000). Starting in 2009, we used radar displays available at MeteoGroup with cell sizes of 1 km x 1° azimuth angle, time intervals of 5 minutes,



and intensity classes in 2-dBZ steps (DWD, 1997).

To identify potential derechos, DWD surface wind measurements have been analysed with respect to concentrated occurrence of severe wind gusts. Following Evans and Doswell (2001), we used a threshold of 25 ms$^{-1}$ to define a severe wind gust as opposed to 26 ms$^{-1}$ as proposed by Johns and Hirt (1987). When more than five stations reported a daily maximum wind gust of 25 ms$^{-1}$ or more, this date was chosen for further analysis. Given the average station spacing of 20 km or less, we expect a high detection rate of potential derechos using this criterion. However, derechos that start or end close to the borders of Germany may affect fewer than five stations. We expect a decreasing detection rate close to the German borders for this reason. In the months from October to March, when high winds frequently occur with extratropical cyclones, it was more effective to identify potential derechos first based on radar displays, choosing only those dates that indicated the presence of mesoscale convective systems or narrow cold-frontal rainbands (NCFR). Convective systems were identified on radar displays by high reflectivity and large reflectivity gradients that typically form linear structures such as bowing line segments. Narrow cold-frontal rainbands identified for cold seasons from 1998/99 to 2008/09 in Gatzen (2011) were completed with events back to 1997 and until 2014 using subjective analysis following Gatzen (2011). A NCFR had to exceed 100 km in length at one time during its lifetime; its width should be small compared to their length axis (i.e. typically about one to four pixels on the radar composite image which corresponds to 2–8 km). An aspect ratio was not used. NCFRs with a lifetime less than 1 h were excluded from the analysis. Bands of precipitation that were likely not related to cold fronts (e.g. convective gust fronts) were also excluded. For dates with more than five severe wind gust measurements (April to September) or indications of mesoscale convective systems or NCFRs (October to March), hourly maximum wind gusts of the networks from DWD and Meteomedia/MeteoGroup were analysed together with plan-view radar images. Only severe wind gusts were considered that could be followed chronologically with the associated mesoscale convective system or NCFR.

In some of the events, severe wind gusts occurred in the environment before the mesoscale convective system or NCFR approached. This happened, for example, when isolated thunderstorms produced wind gusts in the pre-derecho area or when severe non-convective winds occurred ahead of the NCFR. Due to the comparatively large time intervals between available wind gust measurements (i.e. 10 or 60 minutes), some of these wind gusts could not be clearly assigned to the derecho. As a first approximation, we assumed the wind gust intensity to be nearly constant ahead of the convective gust front. When a measurement site reported severe winds prior to the approach of the convective system, it was only used for the derecho classification when it recorded a further increase of the maximum wind gust of at least 3 ms$^{-1}$ during the passage of the convective system or NCFR. The passage of the convective system or NCFR was analysed on radar display, as well as observations at the SYNOP station such as wind direction, temperature, or dewpoint temperature. This increase in wind speed is about 10% or roughly 1 increment on the Beaufort scale and is used to exclude reports with almost constant severe wind gusts ahead of the derecho and no further increase in speed with the derecho passage. This method is also used to estimate the contribution of convection to wind gusts assigned to a derecho, although it is sometimes not possible to clearly distinguish between the contributions of non-convective and convective processes as discussed in van den Broeke et al. (2005). Based on the resulting





subset of severe wind reports, an event was classified as a derecho when there was a concentrated area that could be followed for at least 400 km along the centre line of the associated linear radar structure. All wind measurements had to occur within a distance of less than 167 km from the other wind measurements. Also, no gaps of more than 2 h between successive wind reports were allowed, to ensure concentrated severe wind occurrence within the derecho path. This procedure is similar to that

introduced by Evans and Doswell (2001).

This study differs from previous derecho climatologies because we included over-water stations such as measurements from buoys, ships, and oil rigs. The threshold of 25 ms$^{-1}$ might be more easily surpassed at open-water stations, because the effect of friction on the wind speed is weaker over the water compared to the land. Wind measurements on oil rigs and ships are not

taken at 10 m but at heights up to 100 m above the sea level. This difference in wind measurements might lead to a higher rating of a wind event when it moves across open water compared to the land. Furthermore, the density of wind measurements affects the detection rate of derechos. Derechos can be followed across the North Sea where the measurement density is high enough, in particular across the southern North Sea. Across the central and northern North Sea, there are fewer wind measurements, and it was more difficult to follow derecho tracks farther out than 100 km away from the German coast. For this reason, derecho

paths across the North Sea may be shorter and/or less reliable compared to derecho paths over land.

Finally, we used lightning data to follow derecho events into areas where radar data was not available, for example over eastern Europe. We used data from the Arrival Time Difference (ATD) system operated by the Met Office (Lee, 1986) available at wetterzentrale.de (2016) until the year 2000 and of the Siemens Blids lightning network (Siemens, 2019) for events after

the year 2000. We decided to expand a derecho path into regions where lighting data allowed a chronological detection of the associated convective system together with the severe wind-gust occurrence.

We took the first and the last gust measurement of a derecho to define its location and time of start and dissipation, respectively. The derecho duration was defined as the corresponding time difference. The path length of derechos was calculated by

the direct distance between the first and the last wind measurements close to the derecho centre line. Following Coniglio and Stensrud (2004), we distinguished between high-end, moderate, and low-end intensity derechos. If a derecho met the requirements by Johns and Hirt (1987) and included three or more wind measurements of at least 38 ms$^{-1}$, two of them occurring during the mature phase of the convective system, then it was classified a high-end intensity event; if it met the requirements by Johns and Hirt (1987) (i.e. that at least three wind reports of 33 ms$^{-1}$ or greater that are separated by 64 km or more occur

within the derecho path), then the event was classified as moderate intensity; the remaining were classified as low-end intensity events. Wind damage was not used to classify the derecho intensity. To allow for a comparison of the derecho density to that of the United States, we calculated the spatial derecho density of moderate and high-end intensity events for grid boxes of 200 km x 200 km similar to Coniglio and Stensrud (2004), counting every derecho that at least partly affected a box and computed the spatio-temporal density by dividing the number of derechos per box by the number of analysed years.




We analysed the large-scale flow in derecho situations indicated by the 500-hPa geopotential charts following Evans and Doswell (2001), Burke and Schultz (2004), and Coniglio and Stensrud (2004). We used reanalysis data (ERA-Interim; Dee et al., 2011) displayed together with the derecho position using the wetter4 visualization tool of MeteoGroup. Because PV intrusions have been observed in association with cold-season derechos (e.g. Pistotnik et al., 2011), we also analysed fields of poten-

tial vorticity (PV) on the 320 K isentropic surface. We used archived GFS charts of wetter3.de (http://www1.wetter3.de/Archiv/), concentrating on PV intrusions as indicated by sharp PV gradients and large PV values of more than 1.5 PV units (PVU; 1 PVU $= 10^{-6}\mathrm{Kkg^{-1}m^2s^{-1}}$); PV was only available for years after 1998.

Similar 500-hPa flow situations were grouped together using the agglomerative hierarchical clustering of the scipy.cluster

python package (Berkhin, 2006; Müllner, 2011). Base fields for the clustering were the NCEP NCAR Reanalysis 2 (available at 00, 06, 12, and 18 UTC[1]; Kalnay et al., 1996) 500-hPa geopotential height fields. We used the analysis time closest before the start time of the derecho and normalized the geopotential fields with respect to their minimum and maximum geopotential heights in the region of interest. The clustering was done in a latitude–longitude box between 20°W–20°E and 40°N–60°N. The distance between two normalized geopotential height fields is calculated as the sum of the Gaussian distance between the

single grid points. Clusters were joined agglomeratively (bottom-up) by calculating the average distance between clusters. We used 500 hPa to analyse synoptic-scale situations to compare the clusters with results from the United States (Coniglio et al., 2004).

To study thermodynamic environments in which derechos in Germany form, we used proximity soundings determined from

criteria used by Evans and Doswell (2001): soundings had to be taken within 150 km and 2 h of the derecho path. We used these criteria to allow for a larger number of proximity soundings although small-scale changes of the environment may be missed that could be significant to the development of convective storms (e.g. Brooks et al., 1994; Potvin et al., 2010; Apsley et al., 2016). Specifically, Markowski et al. (1998) found meso- to storm-scale changes of the low-level vertical wind shear near supercells, and Potvin et al. (2010) found that the best distance of proximity soundings from tornadoes is in the range

40–80 km and less than 2 h. Moreover, Apsley et al. (2016) analysed a simulation of a NCFR that showed a narrow zone of CAPE in a narrow strip just ahead of the rainband that would be not indicated by soundings taken more than 100 km away. For this reason, proximity soundings will be interepreted with care.

Every proximity sounding had to be launched ahead of the convective system or NCFR. To ensure that soundings were re-

leased ahead of the derecho gust front, we took the release time that is given in the raw data available at University of Wyoming sounding data archive (http://weather.uwyo.edu/upperair/sounding.html) and analysed the derecho position as indicated by radar data and ground observations such as wind direction and temperature at the release time. Parameters such as mixed-layer CAPE were taken from the University of Wyoming sounding data archive (http://weather.uwyo.edu/upperair/sounding.html).

---

[1]Central European Time CET = UTC + 1 in the cold season and Central European Summer Time CEST = UTC + 2 in the warm season when daylight savings time is in effect.





Here, parameters are based on the lifting curve of the lowest 500-m mixed-layer parcel using the virtual-temperature correction of Doswell and Rasmussen (1994). Additionally, we computed 0–1-km vertical wind shear as the magnitude of the vector difference between the wind at 1 km altitude above ground level and the 10-m (or surface) wind. Similarly, we computed the 0–6-km vertical wind shear by the magnitude of the vector difference between the winds at 6 km altitude above ground level

and the surface. We linearly interpolated the wind vectors at 1 km and 6 km based in the closest wind measurements near the corresponding heights.

## 3    Results

Between 1997 and 2014, 40 derechos at least partly affected Germany (Table 1). This is more than two derechos every year on

average, with a range from zero derechos in1998 and 2012 to four derechos in the year 2003. In the following paragraphs, we present the characteristics of German derechos.

### 3.1    Derecho occurrence across Germany

The regional derecho frequency was analysed based on the areas of each of the 40 derecho paths. Some parts of Germany are

much more affected than others. In general, the derecho density increases from the low lands in the north to the higher terrain in the south, with the maximum density across southern central Germany (Fig. 1). Here, the derecho density of moderate and high-end intensity events reaches 0.72 per year or about three derechos every four years for 200 km x 200 km grid boxes. Derechos are less frequent across Germany compared to the eastern United States, where four moderate and high-end intensity derechos can be expected every three years for similar grid boxes in the most affected region from Minnesota and Iowa to

Pennsylvania and from the eastern parts of the southern Plains to the lower Mississippi Valley (Coniglio and Stensrud, 2004). The derecho frequency of southern Germany is comparable to that of the Appalachian Mountains in the United States (Coniglio and Stensrud, 2004). Fewer derechos have been found across northern Germany. Along the coasts of the North and Baltic Seas derechos are rare with only 0.28 to 0.33 events per year or about one event in three to four years. The rarity of derechos along the coast is similar to the rarity of thunderstorms along the coast (Wapler, 2013) and the low mixed-layer CAPE along the coast

relative to locations farther south (Siedlecki, 2009).

German derechos are less intense compared to the United States. Five of 40 (13%) German derechos reached high-end intensity according to the definition of Coniglio and Stensrud (2004), with 19 (48%) being moderate and 16 (40%) being low-end intensity. A relatively small fraction of German derechos is of high-end intensity compared to the 23% in the United States

(55 of 244 events, Coniglio and Stensrud, 2004)[2]. Also, a relatively large fraction of German derechos is of low-end intensity

---

[2]The intensity classification by Coniglio and Stensrud (2004) also includes damage reports that could result in a larger fraction of moderate and high-end intensity events.





compared to the 30% in the United States (73 of 244 events analysed by Coniglio and Stensrud, 2004).

The path length of German derechos is between 400 and 600 km for 15 of 40 events (38%) and between 600 and 800 km for another 15 of 40 events (38%). Ten of the 40 derecho paths (25%) exceed 800 km. Germany extends about 600 km from west to east and 850 km from north to south, and it is possible that its limited areal extent and the lower detection rate of derecho paths beyond the German borders leads to shorter derecho paths for some of the analysed events. However, it was possible to follow derecho paths farther into neighbouring countries based on internationally distributed wind reports. Some events could be tracked as far as Scotland or Serbia.

German derechos occur most commonly in June and July, as 16 of 40 derechos (40%) fall within these two months (Fig. 2). In late summer, the derecho frequency drops rapidly, and September is the only month with no recorded derecho during the 18-year period. This clear seasonal derecho frequency maximum is comparable to that in the United States (e.g. Coniglio et al., 2004). From December to February, there is another peak in the German derecho distribution. These three months account for 12 derechos (30%), and January is the month with the third-highest derecho number during the analysed time period. The secondary maximum during the cold season also occurs in the United States, although it is less pronounced; October to March contribute to less than 25% of the United States derecho number (Ashley and Mote, 2005), compared to 38% in Germany.

The two seasonal peaks of derecho occurrence are associated with distinct flow patterns at mid levels. A cluster analysis of the 500-hPa geopotential fields indicates two main clusters that account for 39 of 40 derechos (Fig. 3). There is a warm-season cluster that represents 22 derechos that occurred between May and August. The 20 May 2006 event is the only one out of 23 events that occurred between May to August and is not included in this cluster. The cold-season cluster contains 16 of 17 derechos that occurred between October and April plus the one on 20 May 2006. There is one event that is not represented in these two clusters, 11 April 1997. From a manual analysis, this event resembles the cold-season cluster, with high geopotential across the Atlantic Ocean to the west and lower geopotential across eastern Europe. In contrast, the warm-season cluster indicates low geopotential across western Europe and higher geopotential across eastern Europe. We therefore assigned the remaining event to the cold-season cluster. In the following sections, we name derechos of the warm-season cluster *warm-season type derechos*; derechos of the other cluster (including the 20 May 2006 event) and the event of 11 April 1997 are named *cold-season type derechos*.

## 3.2 Warm-season type derechos

In the 18-year analysis, 22 of 40 derechos (55%) are of the warm-season type. They form from May to August with a frequency peak in early July (Fig. 2). The diurnal cycle of diabatic heating is likely important for these events. Seventeen of the 22 warm-season type derechos (77%) start in the afternoon and early evening between 12 and 17 UTC, corresponding to 14 and 19 Central European Summer Time, and a further three (14%) between 9 and 12 UTC (11 and 14 Central European





Summer Time, respectively) (Fig. 4a). Their intensity is high-end for three of 22 (14%) and low-end for seven of 22 (32%); their average path length is 620 km. Almost all of these derechos move from the south-west to the north-east across Germany (Fig. 5a). They frequently start north of the Swiss Alps or over the northern slopes of the Jura Mountains across Switzerland and France (Fig. 1a). Over Germany, 12 of 22 warm-season type derechos follow either eastward paths along the northern

slopes of the Alps eventually entering Austria or paths along the north-western slopes of the Swabian and Frankonian Jura mountains toward central and eastern Germany. In contrast to the relatively high derecho frequency in southern Germany, only two warm-season type events affect the northern Germany in the 18-year period (Fig. 5a). Warm-season type derechos that affect northern Germany move from south to north, whereas the events close to the Alps have a large eastward component of motion.

The average synoptic-scale 500-hPa flow of the warm-season type cluster shows an intense west-European trough and a strong south-westerly flow across Germany (Fig. 3a). The warm-season type thus typically occurs downstream of intense troughs to the west of Germany. As indicated by the anticyclonic curvature of the average geopotential contour lines over eastern Germany, some derechos move into the downstream ridge during their lifetime. These events can be a challenge to

operational forecasters because they occur in regions where severe storms might not be necessarily expected. Furthermore, derechos of the warm-season type frequently occurr in derecho families with a recurrent large-scale weather pattern (Bentley and Sparks, 2003; Ashley et al., 2004). There are two events that follow each other on 02 and 04 July 2000, 12 and 14 July 2010, and 04 and 06 August 2013. Moreover, there are two derechos per day on 02 June 1999, 23 July 2009, and 22 June 2011.

In addition to the synoptic-scale flow on derecho days, the thermodynamic environment can help to assess the derecho potential. We analysed 23 proximity soundings for the warm-season derecho type (right columns of Table 2) with respect to sounding parameters such as mixed-layer CAPE and vertical wind shear (Fig. 6). We used Púčik et al. (2015) and Taszarek et al. (2017) to compare the results with a larger set of sounding parameters across central Europe. Púčik et al. (2015) identified 1135 proximity soundings by using a maximum distance of 150 km to severe wind reports occurring up to 3 h after the

sounding time. His study covered western and central Europe for March–October of 2008–2013. Taszarek et al. (2017) found 828 proximity soundings by using a maximum distance of 125 km and time differences of up to 2 h before and 4 h after the sounding time. This study covered March–October 2009–2015 warm season across western and central Europe. Additionally, Taszarek et al. (2017) present further categories of proximity soundings such as non-severe thunderstorms (8361 soundings), and extremely severe convective wind gusts (33+ ms$^{-1}$; 23 soundings). We compare medians of sounding parameters of both

studies with those of German derechos. Furthermore, box-and-whisker plots of German derechos (Fig. 6) are compared with those presented in Taszarek et al. (2017, their Figures 3 and 8) to analyse the ability of sounding parameters to discriminate between environments of derechos and other categories of thunderstorms.

The 0–6-km vertical wind shear in warm-season proximity soundings in Germany is relatively strong. The median is 20.1

ms$^{-1}$ (Fig. 6d and Table 3) compared to a median of 16.1 and about 13.8 ms$^{-1}$ for severe convective wind gusts indicated by





Púčik et al. (2015) and Taszarek et al. (2017), respectively. Additionally, the median of derecho 0–6-km shear is higher than the median of the extremely severe wind-gust category (about 19.5 ms$^{-1}$ Taszarek et al., 2017). Deep-layer shear discriminates between derecho and non-severe thunderstorm environments, as the median of 20.1 ms$^{-1}$ almost reaches the 90th percentile of non-severe thunderstorm environments which is about 20.5 ms$^{-1}$ (Taszarek et al., 2017). The 0–6-km shear also discriminates

between derecho environments and severe convective wind environments with an upper quartile of about 19.5 ms$^{-1}$ (Taszarek et al., 2017). The 0–1-km shear indicates a similar tendency. The median increases from about 3.5 ms$^{-1}$ over 5 ms$^{-1}$ to about 5.8 ms$^{-1}$ for respectively non-severe thunderstorms, severe convective wind gusts, and extremely severe convective wind gusts (Taszarek et al., 2017). It reaches 6.7 ms$^{-1}$ for derechos (Fig. 6c and Table 3). Púčik et al. (2015) indicated a rather strong median of 0–1-km shear for severe convective wind reports that is about the same magnitude than the median for derechos (6.6

ms$^{-1}$ compared to 6.7 ms$^{-1}$). Nonetheless, 0–1-km shear does not discriminate well between derecho environments and that of other categories. For example, the derecho median of 0–1-km shear is only slightly higher than the upper quartile for the non-severe thunderstorm category (about 6 ms$^{-1}$ Taszarek et al., 2017) and for one proximity derecho sounding, 0–1-km shear is as weak as 1.6 ms$^{-1}$ (Fig. 6c).

In addition to vertical wind shear, the equilibrium-level temperature of derecho soundings could discriminate between non-severe thunderstorm and derecho environments. The median equilibrium-level temperature is –48°C (Fig. 6e and Table 3), which is below the 90th percentile of non-severe thunderstorm environments, and it is also about 20°C below the medians of severe and extremely severe convective-wind environments (Taszarek et al., 2017). The median of mixed-layer CAPE of warm-season type derechos in Germany is 513 Jkg$^{-1}$. This value is higher than the medians of mixed-layer CAPE for non-

severe thunderstorms and severe convective wind gusts presented in (Taszarek et al., 2017) which are about 100 Jkg$^{-1}$ and 250 Jkg$^{-1}$, respectively, but below the median for severe convective wind reports in Púčik et al. (2015) that reaches 695 Jkg$^{-1}$ of most-unstable CAPE[3]. Furthermore, the upper quartile of mixed-layer CAPE for non-severe thunderstorms is well below the median of derechos (375 Jkg$^{-1}$, Taszarek et al., 2017), so that mixed-layer CAPE discriminates between derecho and non-severe thunderstorm environments. Other thermodynamic parameters are less useful to anticipate derecho potential.

Low-level mixing ratio is not useful to discriminate between derecho and non-severe thunderstorm environments, as well as between non-severe thunderstorm and severe convective wind gust environments, although its median is slightly higher for German derechos with 10.6 gkg$^{-1}$ (Fig. 6g and Table 3) compared to approximately 9.7 gkg$^{-1}$ for non-severe thunderstorms and approximately 10.3 gkg$^{-1}$ for severe convective wind gusts (Taszarek et al., 2017). Finally, derechos can form when the level of free convection (LFC) is close to the ground (e.g. 885 hPa) or at greater heights (e.g. 605 hPa). There is no tendency

for derechos to form in association with particularly low or high LFCs (Fig. 6f and Table 3).

In summary, German warm-season type derecho environments are characterised by relatively strong vertical wind shear and cold equilibrium levels compared to other convective events across central Europe. Additionally, mixed-layer CAPE is relatively large. Compared to derecho proximity soundings from the United States, 0–6-km vertical wind shear in German

---

[3]Most-unstable CAPE and mixed-layer CAPE are calculated differently and cannot be compared directly.





warm-season derecho environments is rather strong. For example, the lower and upper quartiles of 0–6-km vertical wind shear of this study are higher compared to the results of Evans and Doswell (2001) (15.9 and 24.7 ms$^{-1}$ compared to 11.8 and 20.0 ms$^{-1}$, respectively). The mixed-layer CAPE values of German warm-season derechos are low compared to those in the United States. For example, for derecho proximity soundings given in Evans and Doswell (2001), the lower quartile of mixed-layer

CAPE[4] is 1097 Jkg$^{-1}$ which slightly exceeds the upper quartile in the German distribution (1061 Jkg$^{-1}$, Fig. 6b). Derechos with low CAPE have been also analysed in the United States (Evans and Doswell, 2001; Burke and Schultz, 2004). These events indicated similar mid-level synoptic-scale flow ahead of advancing high-amplitude troughs (strongly-forced synoptic situations; Evans and Doswell, 2001, their Fig. 1b). Next to differences in sounding parameters, German warm-season type derechos form later in the year compared to the results of Bentley and Sparks (2003), with an annual maximum in early July

in Germany compared to May to July in the United States. For the United States, the location of the monthly maximum of derecho activity moves northward in the spring and summer (Bentley and Sparks, 2003). Likewise, the later derecho maximum in Germany may be related to its high latitude and the associated average seasonal CAPE distribution.

### 3.3 Cold-season type derechos

Eighteen of the 40 derechos (45%) are of the cold-season type that mostly form during the secondary annual frequency maximum from December to February (Fig. 2). Only three derechos are assigned to the cold-season type after 1 March (11 April 1997, 29 April 2002, 20 May 2006). During the winter, diurnal heating is weak in Germany. Consequently, a less distinct diurnal cycle is present for cold-season type derechos in comparison to the warm-season type (Figs. 4a and 4b). Seven of 18 (39%) cold-season type derechos start between 18 and 6 UTC (19 and 7 Central European time, respectively), when there is

no insolation (Fig. 4b). Nonetheless, diurnal heating has some influence as indicated by a sharp frequency maximum in the early afternoon from 12 to 15 UTC (13 to 16 Central European time), when 10 of 18 (56%) cold-season-type derechos start (Fig. 4b). The intensity of the cold-season type is high-end for two of 18 (11%) and low-end for nine of 18 (50%) events. On average, cold-season type derechos are slightly weaker than the warm-season type given the smaller fraction of high-end intensity and a larger fraction of low-end intensity compared to the warm-season type (14 and 32%, respectively). However,

the most intense derecho in the dataset is of the cold-season type and occurred on 01 March 2008. Moreover, the average path length of the cold-season type is 880 km, which is 1.4 times the average path length of the warm-season type (path lengths are given in Table 1).

In contrast to the warm-season type, most cold-season type derechos occur across central Germany (Fig. 5b). Four events

start over the North Sea, and a further 10 start over north-western Germany, Belgium, and the Netherlands. This regional variability could be related to the distance to the North Sea that likely serves as a source of low-level humidity in the cold season. Furthermore, cold-season type derechos move typically from north-west to south-east across Germany; only two have more

---

[4]Evans and Doswell (2001) also include cold-season derechos and calculate mixed-layer CAPE for the lowest 100-m mixed-layer parcel instead of the lowest 500-m mixed layer parcels used in our study.





west–east-oriented paths (Fig. 5b). The motion of cold-season type derechos is related to the large-scale flow. The cold-season type is associated with a westerly to north-westerly 500-hPa flow in the wake of low geopotential heights across eastern Europe (Fig. 3b, c). A comparable flow pattern has been described for cold-season bow echoes in the United States (Johns, 1982, 1984; Burke and Schultz, 2004). In many situations, an intense trough amplifies across Germany during the event, associated with

upper-tropospheric PV intrusions in the mid-level troposphere (not shown). Cold-season type derechos occur in 11 out of 19 cold seasons, and some of these seasons are associated with groups of derechos: 2001/02 (three derechos), 2007/08 (two derechos), and 2013/14 (two derechos). There are no derechos in the cold-seasons 1997/98, 2000/01, 2005/06, 2008/09, 2009/10, 2010/11, 2011/12, and 2012/13.

To characterise the cold-season derecho environment, we analysed 31 proximity soundings (left columns in Table 2). We compared sounding parameters to those of proximity soundings across central Europe given in Púčik et al. (2015) (for severe convective wind events) and across the United States presented by Burke and Schultz (2004) (for cold-season bow echoes). Cold-season type derecho environments are characterised by exceptionally strong 0–6-km shear that is calculated for 28 of the 31 soundings (Fig. 6). It is at least 22 ms$^{-1}$ and peaks at 59 ms$^{-1}$, the median is 34.5 ms$^{-1}$ (Table 3), what is slightly

above the median of 0–6-km shear presented for severe convective-wind environments in Púčik et al. (2015) (33.2 ms$^{-1}$). Strong deep-layer vertical wind shear is also common in cold-season bow-echo environments in the continental United States with a mean 0–5-km shear of 23 ms$^{-1}$ (Burke and Schultz, 2004). The 0–1-km shear is strong as well with a median of 21.2 ms$^{-1}$ (Fig. 6c and Table 3) which is slightly higher compared to cold-season severe convective wind environments (18.1 ms$^{-1}$, Púčik et al., 2015). For cold-season bow echoes in the United States, 0–2.5-km shear is weaker than 0–1-km shear in German

derechos. The mean 0–2.5-km shear is 14 ms$^{-1}$ with 87% of proximity soundings having less than 20 ms$^{-1}$ shear (Burke and Schultz, 2004).

Forty-one percent of proximity soundings in German cold-season type derechos have zero mixed-layer CAPE. The maximum and median of mixed-layer CAPE are 231 and 3 Jkg$^{-1}$, respectively, and the lower quartile is 0 Jkg$^{-1}$ (Fig. 6a and Table

3). This was not unexpected; proximity soundings of severe convective-wind events across central Europe show similar results with a median most-unstable CAPE of 14 Jkg$^{-1}$ (Púčik et al., 2015). Furthermore, although most cold-season bow echoes in the continental United States have higher most-unstable CAPE, there are also long-lived bow-echo proximity soundings with most-unstable CAPE less than 100 Jkg$^{-1}$ (Burke and Schultz, 2004). In addition to low mixed-layer CAPE, German cold-season type derecho environments are characterised by low moisture in the lowest 500-m layer above the ground. The

highest mean mixing ratio is 8.3 gkg$^{-1}$, the median is 5.2 gkg$^{-1}$ (Table 3), and the lowest mixing ratio is 3.5 gkg$^{-1}$ (Fig. 6g). These are rather low values; in the continental United States, Burke and Schultz (2004) presented a minimum of mean lowest 100-hPa mixing ratio of 9 gkg$^{-1}$ for nine long-lived cold-season bow echoes. Nonetheless, there is a low LFC in many German cold-season type derecho soundings (median 867 hPa; Table 3), indicating high relative humidity of the low-level air mass.





The equilibrium-level temperature in proximity soundings of cold-season type derechos is high in some events, peaking at +6°C, with a median of –7.5°C (Fig. 6e, Table 3). Although cold-season type derechos do not need to be associated with thunderstorms by definition, all German events produced lightning, which becomes likely at sufficiently low equilibrium-level temperatures (van den Broeke et al., 2005). Soundings with high equilibrium-level temperatures are likely not representative for

the particular derecho events for this reason. One example is the derecho proximity sounding at Bergen (WMO number 10238) for 12 UTC 28 December 2001. The sounding has zero mixed-layer CAPE, whereas the 12-UTC SYNOP at the same location reports a thunderstorm with graupel (not shown). Rapid changes of the local equilibrium-level temperature can be expected to occur close to this cold-season type derecho in the time between the sounding ballon launch (1047 UTC) and the thunderstorm observation (1150 UTC). Forecasters have to be aware that soundings may not indicate the potential of convective storms in

the vicinity of cold-season type derechos.

In summary, cold-season type derechos in Germany are associated with strong vertical wind shear with a median 0–6-km shear of 35 ms$^{-1}$ and a median 0–1-km shear of 21 ms$^{-1}$. At the same time, mixed-layer CAPE is low and soundings frequently have zero mixed-layer CAPE. Taking into account that German cold-season type derechos have longer path lengths on

average and are of similar intensity than warm-season type derechos, we expected them to have a similar potential impact than warm-season type derechos in Germany.

## 4   Conclusions

This work analyses the derecho potential across Germany. Based on the presented climatology, the derecho risk is higher than

would be expected from previously published cases (four events in 10 years between 2004 and 2014): 40 events were classified in the 18-year period, including 24 moderate and high-end intensity derechos. The highest regional derecho risk occurs in southern Germany with about three moderate and high-end intensity derechos in four years, a value of similar magnitude to the Appalachian mountains in the eastern United States (Coniglio and Stensrud, 2004). The highest yearly derecho number in Germany was four in 2003; no derechos formed in 1998 and 2012. German derechos are most frequent in June and July.

The winter season (October–March) contributes on average to 40% of the annual derecho number, although eight of 18 winter seasons had no derechos. The relative contribution of German winter season derechos to the annual derecho number is large compared to the winter season in the United States, which contributes only 25% to the annual derecho number (Ashley and Mote, 2005).

According to a cluster analysis of the 500-hPa flow across Europe, German derechos form in two distinct synoptic-scale situations. Twenty-two derechos correspond to the first derecho type that forms in strong south-westerly 500-hPa flow downstream of an intense trough across the north-eastern Atlantic Ocean and western Europe. This derecho type only occurs in the summer months (May to August) and is thus named warm-season type accordingly. From proximity soundings, there is less mixed-





layer CAPE in the pre-derecho region than expected from case studies of central European warm-season derechos. Median mixed-layer CAPE in this study is 500 Jkg$^{-1}$ versus mixed-layer CAPE between 1000 and 2500 Jkg$^{-1}$ in previous derecho case studies (Gatzen, 2004; Púčik et al., 2011; Hamid, 2012; Celiński-Mysław and Matuszko, 2014). Derechos in environments with limited CAPE have also been described in the United States (Evans and Doswell, 2001; Burke and Schultz, 2004) and are

characterised by similar mid-level synoptic-scale flow ahead of advancing high-amplitude troughs (strongly-forced synoptic situations; Evans and Doswell, 2001, their Fig. 1b).

We also compared the set of warm-season type derecho proximity soundings with the results of studies on central European sounding analyses by Púčik et al. (2015) and Taszarek et al. (2017) to evaluate if derecho environments differ from those of

non-severe thunderstorms or severe convective wind events. For example, the median mixed-layer CAPE of warm-season type derechos is 513 Jkg$^{-1}$ compared to about 113 Jkg$^{-1}$ for non-severe thunderstorms (Taszarek et al., 2017); their equilibrium-level temperature is –48°C , which is below the 90th percentile of non-severe thunderstorms (Taszarek et al., 2017). The best parameter to distinguish between environments of warm-season type derechos and other convective events in Germany is 0–6-km shear, given that the median of 20.1 ms$^{-1}$ is higher than the median of all severe convective wind gust proximity soundings

which is 16.1 (Púčik et al., 2015) and about 13.8 ms$^{-1}$ (Taszarek et al., 2017). Compared to proximity soundings in the United States, 0–6-km shear of German warm-season type derechos is stronger, with lower and upper quartiles of 15.9 and 24.7 ms$^{-1}$ compared to 11.8 and 20.0 ms$^{-1}$ (Evans and Doswell, 2001). Other sounding parameters showed less discrimination between derechos and other convective situations in Germany, particularly for LFC and the low-level mixing ratio. In conclusion, forecasters need to be aware that warm-season type derechos can develop in environments with CAPE below 500 Jkg$^{-1}$, when

concurring strong vertical wind shear and large-scale forcing are present. Strong south-westerly flow at 500 hPa can also indicate derecho potential in such situations.

The remaining 18 of 40 derechos are of the cold-season type. In contrast to the warm-season type, these derechos form in strong north-westerly 500-hPa flow, mostly at the south-western flank of rapidly amplifying 500-hPa troughs. These troughs

are always associated with PV intrusions at 500 hPa, and derechos form close to the PV intrusions. Cold-season type derechos have a clear frequency peak from December to February, but some events occur as early as October or as late as May. Proximity soundings indicate high-shear–low-CAPE environments, and cold-season type derechos are associated with exceptionally strong 0–1-km and 0–6-km vertical wind shear with a median of 21 and 35 ms$^{-1}$, respectively. These median shear values are stronger than for cold-season severe convective wind events in central Europe (18 and 33 ms$^{-1}$ Púčik et al., 2015) and

cold-season bow-echo environments in the continental United States, with a mean 0–2.5-km shear of 14 ms$^{-1}$ and a mean 0–5-km shear of 23 ms$^{-1}$, respectively (Burke and Schultz, 2004). Mixed-layer CAPE has a median of 14 Jkg$^{-1}$ and is zero in 41% of cold-season type derecho proximity soundings. Moreover, the equilibrium-level temperature has a median of –7.5°C and frequently does not indicate any potential of thunderstorms close to cold-season type derechos. Forecast parameters based on CAPE may thus fail to indicate derecho potential in these cases. The limited capability of CAPE-based forecast parameters

to provide insight on cold-season convective potential has been discussed in the United States (Sherburn et al., 2016). On the



other hand, the relation to mid-tropospheric PV intrusions is promising with respect to forecasting of cold-season type dere-chos. Further work should thus focus on PV intrusions and their relation to cold-season type derechos.

Despite the differences with respect to the large-scale flow and environmental conditions, the comparison of warm-season

and cold-season type derechos reveals no clear differences in terms of their intensity. On the one hand, a larger fraction of warm-season type derechos reach moderate or high-end intensity (54% and 14%, respectively) compared to cold-season type derechos (39% and 11%, respectively). On the other hand, the most intense derecho during the investigation period is of the cold-season type, and cold-season type derecho path lengths are about 1.4 times longer on average (880 km compared to 620 km for the warm-season type). The potential impact of cold-season type derechos in Germany is thus expected to be similar to

that of warm-season type derechos.

*Data availability.*  The ERA-Interim reanalysis data are available publicly from their website https://www.ecmwf.int/en/forecasts/datasets/reanalysis-datasets/era-interim. NCEP Reanalysis 2 data are provided by the NOAA/OAR/ESRL PSD, Boulder, Colorado, USA, from their website at https://www.esrl.noaa.gov/psd/. Radar and wind data are available via the DWD website (www.dwd.de). The MeteoGroup data

are not freely available but can be made available via the provider. All code is available from the authors.

*Author contributions.*  Christoph P. Gatzen, Andreas H. Fink, David M. Schultz, and Joaquim G. Pinto conceived and designed the research. Christoph P. Gatzen performed the analysis, prepared the figures and wrote the initial draft of the paper. All authors contributed with discussions and revisions.

*Competing interests.*  The authors declare that they have no conflict of interest.

*Acknowledgements.*  We thank the ECMWF and NCAR for the provision of reanalysis data. We thank the DWD for radar and wind data and MeteoGroup for additional wind data and the permission to use the wetter4 display tool. Patrick Ludwig prepared the map presented in figure 1a. We are grateful to the European Severe Storms Laboratory (ESSL) for the reports taken from the European Severe Weather Database (ESWD; http://www.eswd.eu/, last access: 18 November 2018). Joaquim G. Pinto thanks the AXA Research Fund for support. Partial funding for David M. Schultz was provided to the University of Manchester by the Natural Environment Research Council through

Grant NE/ N003918/1.



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



**Table 1.** Derechos found between 1997 and 2014. Countries of start and end points are abbreviated as GER (Germany), AUT (Austria), FRA (France), SVK (Slovakia), CZE (Czechia), POL (Poland), DEN (Denmark), SUI (Switzerland), BEL (Belgium), NED (Netherlands), SRB (Serbia), SCO (Scotland), ENG (England), CRO (Croatia).

| Number | Start time (UTC) | Path length (km) | Duration (h) | Intensity | Path | | |
|---|---|---|---|---|---|---|---|
| 1 | 11 April 1997 02:00:00 | 1620 | 16 | moderate | NW GER | - | E AUT |
| 2 | 05 February 1999 00:00:00 | 660 | 10 | moderate | N GER | - | Centr GER |
| 3 | 02 June 1999 14:00:00 | 420 | 6 | moderate | SW GER | - | Centr GER |
| 4 | 02 June 1999 15:00:00 | 600 | 7 | high | E FRA | - | SE GER |
| 5 | 03 December 1999 15:00:00 | 560 | 6 | low | NW GER | - | E GER |
| 6 | 02 July 2000 14:00:00 | 550 | 7 | moderate | N FRA | - | W GER |
| 7 | 04 July 2000 11:00:00 | 630 | 7 | moderate | SE GER | - | N SVK |
| 8 | 06 July 2001 17:00:00 | 450 | 6 | moderate | E FRA | - | Centr GER |
| 9 | 28 December 2001 09:00:00 | 750 | 12 | low | NW GER | - | N CZE |
| 10 | 28 January 2002 12:00:00 | 930 | 12 | low | N GER | - | SE POL |
| 11 | 28 January 2002 13:00:00 | 430 | 5 | low | NW GER | - | E GER |
| 12 | 29 April 2002 12:00:00 | 410 | 6 | low | W GER | - | E GER |
| 13 | 10 July 2002 16:00:00 | 600 | 7 | high | SE GER | - | E DEN |
| 14 | 28 January 2003 06:00:00 | 440 | 6 | low | W GER | - | SE GER |
| 15 | 19 May 2003 15:00:00 | 620 | 6 | low | N SUI | - | Centr CZE |
| 16 | 14 June 2003 06:00:00 | 1000 | 15 | moderate | N FRA | - | SE AUT |
| 17 | 15 December 2003 21:00:00 | 700 | 6 | low | N GER | - | E CZE |
| 18 | 12 August 2004 15:00:00 | 590 | 9 | low | N SUI | - | E AUT |
| 19 | 19 November 2004 06:00:00 | 850 | 12 | moderate | S GER | - | SE POL |
| 20 | 12 February 2005 15:00:00 | 610 | 7 | low | S BEL | - | SE GER |
| 21 | 29 July 2005 14:00:00 | 740 | 11 | high | NW SUI | - | E GER |
| 22 | 20 May 2006 12:00:00 | 600 | 8 | moderate | W GER | - | NW CZE |
| 23 | 18 January 2007 13:00:00 | 1100 | 11 | moderate | N NED | - | SE POL |
| 24 | 21 June 2007 06:00:00 | 1020 | 15 | moderate | W SUI | - | N SVK |
| 25 | 22 February 2008 15:00:00 | 1150 | 13 | moderate | NW DEN | - | E POL |
| 26 | 01 March 2008 02:00:00 | 1520 | 16 | high | W NED | - | Centr SRB |
| 27 | 25 June 2008 16:00:00 | 570 | 6 | low | E GER | - | Centr SVK |
| 28 | 26 May 2009 12:00:00 | 570 | 8 | moderate | SW SUI | - | SW CZE |
| 29 | 23 July 2009 17:00:00 | 520 | 7 | moderate | SE GER | - | N SVK |
| 30 | 23 July 2009 15:00:00 | 670 | 7 | low | E GER | - | Centr POL |
| 31 | 12 July 2010 09:00:00 | 650 | 6 | moderate | S BEL | - | SW DEN |
| 32 | 14 July 2010 12:00:00 | 500 | 5 | moderate | Centr FRA | - | Centr NED |
| 33 | 22 June 2011 12:00:00 | 450 | 6 | low | SW GER | - | E GER |
| 34 | 22 June 2011 13:00:00 | 550 | 5.5 | moderate | E FRA | - | SE GER |
| 35 | 24 August 2011 16:00:00 | 440 | 6 | low | W GER | - | E GER |
| 36 | 04 August 2013 10:00:00 | 790 | 10 | low | N SUI | - | W SVK |
| 37 | 06 August 2013 12:00:00 | 750 | 10 | moderate | SW GER | - | E GER |
| 38 | 05 December 2013 05:00:00 | 1160 | 13 | high | N SCO | - | NW GER |
| 39 | 03 January 2014 14:00:00 | 680 | 7 | low | N FRA | - | N GER |
| 40 | 21 October 2014 13:00:00 | 1600 | 15 | moderate | S ENG | - | E CRO |





**Table 2.** Proximity soundings of derechos between 1997 and 2014. The left column lists soundings of cold-season type derechos, the right colmun lists soundings of warm-season type derechos. Sounding sites are given by their WMO station identifier code. Soundings marked with * are chosen although the derecho moved across the sounding site 3 h after the sounding launch instead of 2 h. For the sounding marked with ** is was no clear evidence that it was crossed by the derecho before or after the launch time.

| Date / Time (UTC) | Sounding # | Date / Time (UTC) | Sounding # |
|---|---|---|---|
| 11 April 1997 12:00:00 | 12425** | 02 June 1999 12:00:00 | 10618 |
| 05 February 1999 00:00:00 | 10035 | 02 June 1999 12:00:00 | 10739 |
| 05 February 1999 06:00:00 | 10393 | 04 July 2000 12:00:00 | 11035 |
| 05 February 1999 12:00:00 | 11520* | 02 July 2000 12:00:00 | 07145 |
| 05 February 1999 12:00:00 | 10771 | 04 July 2000 12:00:00 | 10868 |
| 28 December 2001 12:00:00 | 10238 | 07 July 2001 12:00:00 | 10868 |
| 28 December 2001 12:00:00 | 10410 | 10 July 2002 18:00:00 | 10393 |
| 28 December 2001 18:00:00 | 11520 | 19 May 2003 18:00:00 | 11520 |
| 28 January 2002 12:00:00 | 10410 | 14 June 2003 12:00:00 | 10739 |
| 28 January 2002 18:00:00 | 10393 | 29 July 2005 12:00:00 | 06610 |
| 29 April 2002 12:00:00 | 10410 | 30 July 2005 00:00:00 | 12425 |
| 29 April 2002 12:00:00 | 06260 | 21 June 2007 12:00:00 | 10739 |
| 28 January 2003 12:00:00 | 11035 | 26 May 2009 18:00:00 | 10771 |
| 28 January 2003 12:00:00 | 06610 | 26 May 2009 18:00:00 | 11520 |
| 15 December 2003 00:00:00 | 11520 | 23 July 2009 12:00:00 | 11520 |
| 15 December 2003 00:00:00 | 10548 | 22 June 2011 12:00:00 | 10618 |
| 19 November 2004 06:00:00 | 10771 | 22 June 2011 12:00:00 | 10739 |
| 19 November 2004 12:00:00 | 11952* | 22 June 2011 18:00:00 | 10771 |
| 12 February 2005 18:00:00 | 10618 | 22 June 2011 18:00:00 | 10393 |
| 20 May 2006 12:00:00 | 10410 | 24 August 2011 18:00:00 | 10548 |
| 20 May 2006 12:00:00 | 07145 | 04 August 2013 12:00:00 | 10739 |
| 20 May 2006 12:00:00 | 06260 | 06 August 2013 12:00:00 | 10739 |
| 20 May 2006 18:00:00 | 10771 | 06 August 2013 12:00:00 | 06610 |
| 18 January 2007 18:00:00 | 10393 | | |
| 23 February 2008 00:00:00 | 10393 | | |
| 23 February 2008 00:00:00 | 12425* | | |
| 01 March 2008 00:00:00 | 06260* | | |
| 01 March 2008 06:00:00 | 10618 | | |
| 05 December 2013 12:00:00 | 10035* | | |
| 21 October 2014 12:00:00 | 07145 | | |
| 21 October 2014 18:00:00 | 10771 | | |



**Table 3.** Median values of sounding parameters derived from derechos of the warm-season (top) and cold-season (bottom) type. From left: Mixed-layer CAPE (MLCAPE), mixing ratio (MIXR), 0–6-km shear (DLS), 0–1-km shear (LLS), equilibrium-level temperature (ELT), and level of free convection (LFC).

| Derecho type | MLCAPE | MIXR | DLS | LLS | ELT | LFC |
|---|---|---|---|---|---|---|
| warm-season | 513 Jkg$^{-1}$ | 10.62 gkg$^{-1}$ | 20.1 ms$^{-1}$ | 6.7 ms$^{-1}$ | -48°C | 753 hPa |
| cold-season | 3 Jkg$^{-1}$ | 5.16 gkg$^{-1}$ | 34.5 ms$^{-1}$ | 21.2 ms$^{-1}$ | -7.5°C | 867 hPa |


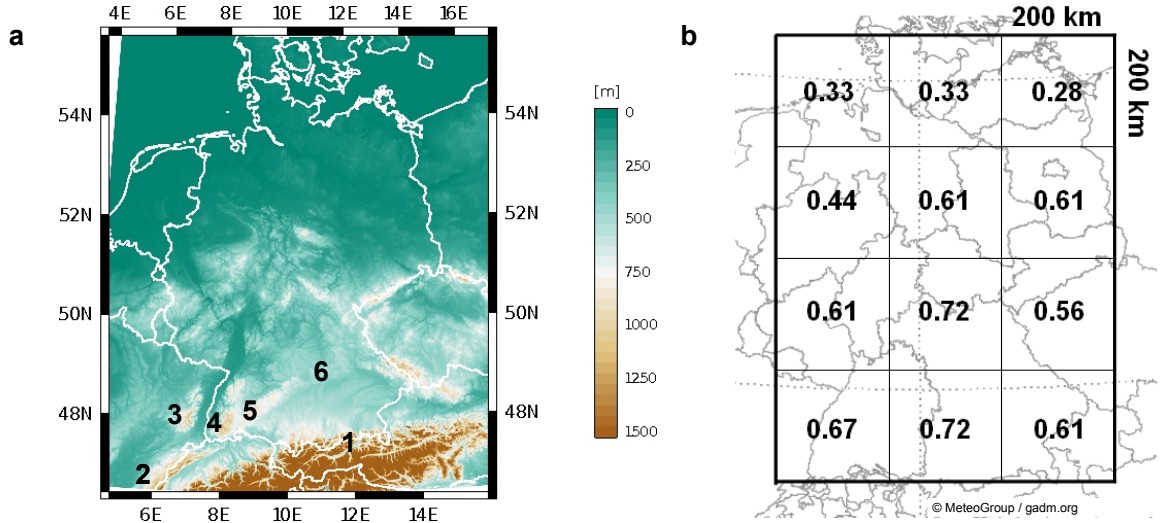

**Figure 1.** (a) Topography of Germany. Terrain height [m] is given by shading according to the colour bar. Mountains mentioned in the text are highlighted by numbers: 1 - Alps, 2 - Swiss Jura, 3 - Vosges, 4 - Black Forest, 5 - Swabian Jura, 6 - Frankonian Jura. (b) Number of moderate and high-end intensity derechos affecting grid boxes with 200 km side length across Germany per year.




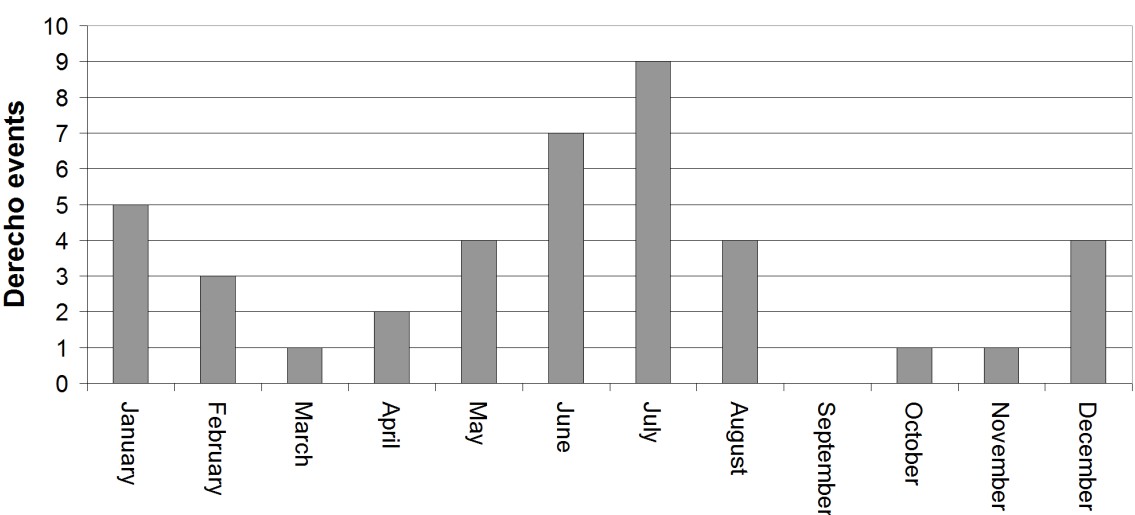

**Figure 2.** Seasonal distribution of derechos in Germany.


**Figure 3.** (a) Averaged 500-hPa geopotential field of the warm-season type derecho cluster. Isohypses are diplayed in 80 m intervals; the 5600 m isohypse is highlighted by the thick line. (b) 500-hPa geopotential field of 11 April 1997 00:00:00 UTC. Isohypses are diplayed in 80 m intervals; the 5600 m isohypse is highlighted by the thick line. (c) Averaged 500-hPa geopotential field of the cold-season type derecho cluster. Isohypses are diplayed in 80 m intervals; the 5600 m isohypse is highlighted by the thick line. (d) Dendrogram of the cluster analysis of 500-hPa geopotential fields in German derecho events. Dates of the analysis fields are given below each event line. The labels of the clusters (a)–(c) are given in the dendogram. The maps in (a)–(c) have been produced with the python module basemap.





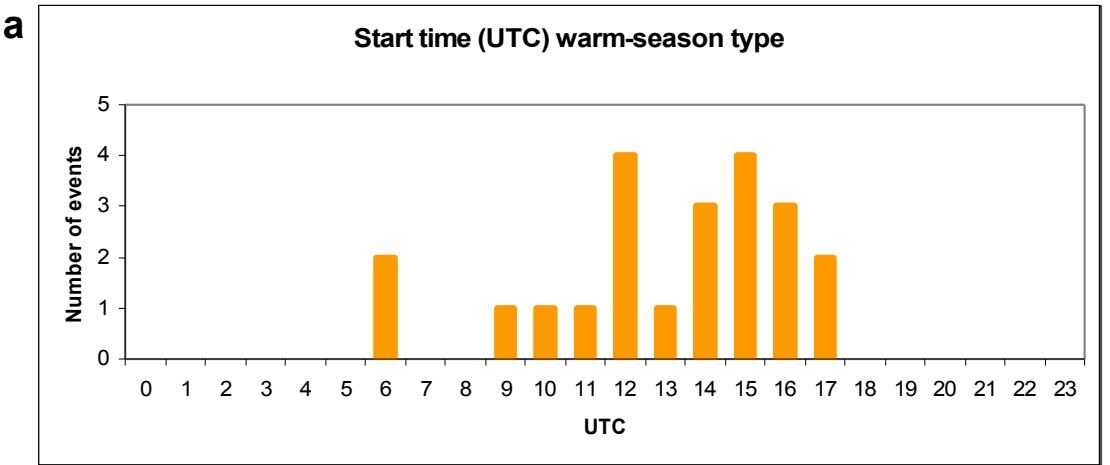

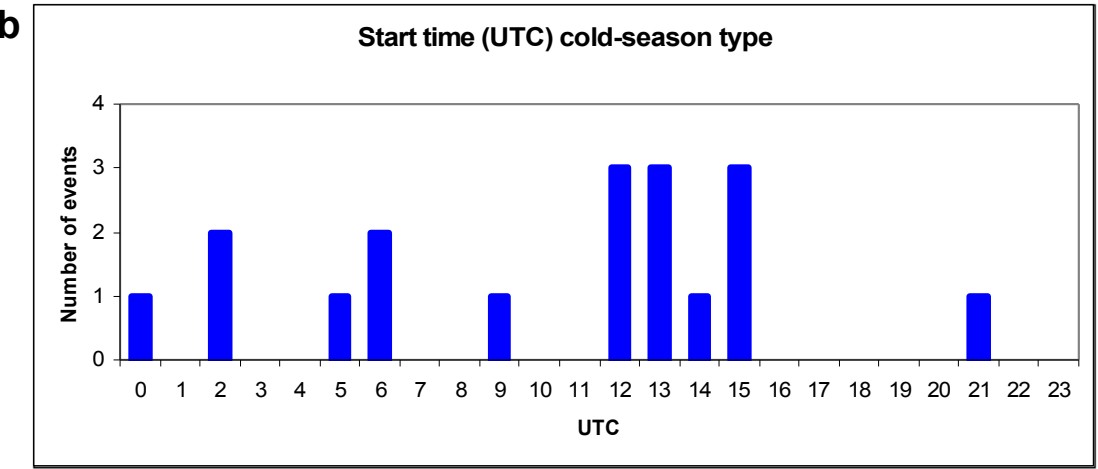

**Figure 4.** Start time (UTC) of derechos of the warm-season type (a) and the cold-season type (b).

**Figure 5.** Tracks of derechos of the warm-season type (a) and the cold-season type (b). Thick lines, thin lines, and broken lines relate to high-end, moderate, and low-end intensity derechos.

**Figure 6.** Box-and-whisker diagrams of warm-season type and cold-season type derecho proximity soundings. Boxes (blue for the cold-season type) give the upper and lower quartiles of the distributions, the end of the thin lines maxima and minima, and the thick lines the medians. (a) Mixed-layer CAPE for the cold-season type; (b) Mixed-layer CAPE for the warm-season type; (c) 0–1-km vertical wind shear; (d) 0–6-km vertical wind shear; (e) Equilibrium-level temperature; (f) Level of free convection; (g) 0–500-m mixing ratio.