# Peer review of "An 18-year climatology of derechos in Germany"

_Natural Hazards and Earth System Sciences, 2019_

## Referee Comment (RC1) · Mateusz Taszarek (Referee) · 14 Nov 2019

Authors addressed climatological aspects of derecho events affecting Germany. Based on the analysis of severe wind reports, radar and lightning data they derived 40 cases that fulfilled derecho criteria. All cases were divided into a warm and cold season derechos. In addition to assessing preliminary derecho climatology, authors used sounding-derived variables derived from proximity observations to investigate thermodynamic and kinematic properties They also used reanalysis data to define typical synoptic patterns supporting development of a derecho. Although the sample size is relatively small, it allowed to obtain a certain signals characterizing atmospheric environments for both categories and their corresponding synoptic patterns.

I need to compliment authors for their efforts in investigating all potential cases over the period of 17 years since they stand for an excellent contribution to severe thunderstorm research over central Europe. Confirmation of each derecho event was certainly a challenging task and required a long and very detailed investigation. Overall, the paper is up to international standards, it is written in a good English, objective of the study is well-formulated and length is adequate. Methodology is described in a great details. References to literature are generally sufficient. However, authors focus mainly on comparing sounding-derived parameters over central Europe and climatological aspects of derechos in USA. Some further discussion on the derecho occurrence over central Europe in the context of general thunderstorm climatology and their corresponding environmental conditions may be considered to assess rarity of such events. Figures in the paper are of acceptable quality, but I believe there is some potential for improvements and space for additional plot. Summing up: the paper is a valuable contribution and should be published in NHESS, but some improvements are needed to increase its quality. Below I present general and specific comments:

General comments:

**1. What was the logic behind choosing a period from 1997 to 2014? Why not until 2018? At some point it may be even possible to investigate 2019 if there were any such cases, so the paper may include the most recent data.**

**2. Did author try other indices? I am missing two very important variables. First is a mid-level shear (e.g. 0-3km) that is commonly used in investigating atmospheric potential for convective windstorms (e.g. https://doi.org/10.1175/811.1, https://doi.org/10.1175/1520-0434(2003)18<502:SCWOTN>2.0.CO;2, https://doi.org/10.1007/s00704-018-2728-6, https://doi.org/10.1175/WAF-D-13-00041.1). Second is a mid-level lapse rate (e.g. 800-500 hPa) that may be useful in characterizing cold-seasons derechos, that usually form in a steep temperature gradients. Also, most of the results in the paper are compared to severe wind events from Pucik et al. (2015) and Taszarek et al.**

[Figure]

(2017), but I am wondering whether authors tried to use a larger number of soundings and define how derecho-related indices differ from the long-term climatological background (null-cases)? Alternatively, this can be derived from a central European sounding-climatology distributions (e.g. https://doi.org/10.1175/JCLI-D-17-0596.1).

**3. Did authors try producing scatterplots of selected pairs of variables? Combination of mixing ratio and lapse rates, or mixing ratio and 0-3km shear may very clearly indicate differences between cold and warm season derechoes (dwo clusters) and provide possibility to compare "coordinates" of these cases with previous or future studies dealing with convective parameters climatology (e.g. using Kernel Density Estimation of climatological background for soundings). In this way, it can be assessed how rare are such environments (see also general comment #2)**

**4. I have a feeling that some further discussion on the climatological background of convective environments over Europe, synoptic patterns and overall thunderstorm frequency can be provided. Authors may discuss how unusual is the situation with a derecho given a background of convective activity over central Europe. I believe that the last section of the paper (conclusions) may be trimmed from repeating what was already written in the results, and instead, some space may be devoted for a deeper discussion.**

Specific comments:

**1. P1: Abstract has 345 words. A usual standard for scientific publishing is around 200-250 words, authors may want to cut it down a bit.**

**2. P1 L4: "mid-tropospheric" may be more appropriate than "500-hPa layer" (same for L11), also what authors mean by "intense"?**

**3. P1 L9: To avoid repetition I suggest "strong 0-6 km wind shear with a median of 20 ms-1 and ..."**

**4. P1 L10: I think that abbreviations shouldn't be used in the abstract, but if authors**

intend to use CAPE, I suggest to mention "mixed-layer" only for the first time, and later just refer to CAPE without adding "mixed-layer" term, also in the whole manuscript body.

**5. P1 L16: It is possible that zero CAPE soundings were not representative enough for sampling a "true" environment in which derecho initially occurred? On the basis of what premise authors came to the conclusion that the choose sounding was representative?**

**6. P2 L3: What do authors mean by "frequently"? I would suggest to remove this word.**

**7. P2 L3-L17: Authors may want to mention also a recent powerful derecho case from Poland from 11 August 2017 where 6 people died being outdoors due to falling trees, including two teenage scouts camping in the forest (https://doi.org/10.1175/MWR-D-18-0330.1). This event had a very unique convective evolution and had a remarkable intensity given European windstorms, it was one of the most significant event over the last 20 years. Also, it took place very close to studied area.**

**8. P2 L12-L17: I don't think references to ESSL twitter reports are needed, authors may just use a reference to ESWD (https://doi.org/10.1016/j.atmosres.2008.10.020) or ESSL (https://doi.org/10.1175/BAMS-D-16-0067.1).**

**9. P3 L13-L15 & L26-L30: Please see comment #7.**

**10. P5 L14: See general comment #1**

**11. P6 L26-L31: Did wind measurements from the water surface were also used to estimate derecho intensity and duration? This may affect results that are presented in second paragraph of section 3.1, especially comparisons with U.S. climatologies where derechos usually do not travel through water surface where friction is lower and promotes stronger wind gusts (that can help reaching derecho criteria).**

**12. P7 L9: Maybe it would be better to use a phrase of: "A similar patterns in 500 hPa geopotential field .."**

**13. P7 L10-L11: Authors used ERA-Interim reanalysis to display derecho positions, GFS for PV intrusions and NCEP/NCAR reanalysis for clustering method. This is quite a mix. Why authors did not use just one source of data? (e.g. ERA-Interim?). Is it possible that the use of various data sources might have introduced inhomogeneities?**

**14. P7 L12: No need to repeat a word "geopotential".**

**15. P8 L3: What does exactly mean "10-m or surface"? Wouldn't be better to write that the shear was computed as a magnitude between first available level from the radiosonde up to 1 or 6 km AGL? (I assume this is the case) .**

**16. P8 L10: "the year" is not needed (to be consistent with earlier part of the sentence "in 1998 and 2012").**

**17. P8 L14: "based on the areas of each derecho path"**

**18. P8 L21-25: Perhaps some short comment on the fraction of warm and cold season derechos in the spatial variability would be good.**

**19. P10 L31: At this point I would like to highlight once again that among 0-1km, 0-3km and 0-6km shear parameters analyzed in a cited study, 0-3km shear demonstrated the best value in discriminating between extremely severe wind events and other categories. See also general comment #2.**

**20. P11 L28-L30: Why authors use pressure values for LFC instead of m AGL? I am not sure if this information adds any value because we don't know what is its relation to the ground level (for each station and in a different synoptic pattern it will be different). Also, we cannot compare it with other studies since majority of them use m AGL. I believe authors should recompute LFC for m AGL. Same for P13 L34.**

**21. P12 L8-L12: This shift may be related to a different peak in thunderstorm activity (compared to USA) based on central European climatolgy of thunderstorms (https://doi.org/10.1007/s00703-013-0285-1, https://doi.org/10.1175/JCLI-D-18-0372.1, https://doi.org/10.5194/nhess-16-607-2016), and European climatology of convective environments (https://doi.org/10.1175/JCLI-D-17-0596.1, https://doi.org/10.1175/JAMC-D-17-0132.1). Peak in derecho frequency seems to overlap with these estimates. Some comparison to synoptic-favouring patterns for thunderstorms in central Europe in relation to patterns obtained in this study (in Figure 3) may be also worth to be done (https://doi.org/10.1016/j.atmosres.2014.07.011). Authors may use these aspects to provide some deeper discussion in the last section (in the exchange for repeating results in the conclusions), especially showing how unusual is the situation with a derecho given a background of convective activity and convective parameter climatology over central Europe (see general comment #4).**

**22. P13 L10-P21: Authors may also provide some comment here on a typical high-shear low-CAPE environments that usually induce such events (https://doi.org/10.1175/WAF-D-13-00041.1).**

**23. P14 L3: Is it possible to provide which value of ELT is "sufficiently low"? Would it be around -10C? (https://doi.org/10.1175/1520-0493(1996)124<0602:CTGLOF>2.0.CO;2).**

**24. P14 L10: Can authors provide some additional discussion why ELT may rapidly change and briefly explain the mechanism leading to steepening of the temperature lapse rates (in a strongly synoptic-scale forced situations) that can "make" CAPE available? I suppose leading author of the paper may have something to say on this process.**

**25. P14 L15: replace "than" with "as".**

**26. P14 L19-L21: Awkward sentence construction, please split it for two sentences for clarity (after the bracket).**

**27. P15 L31: Wasn't ML CAPE median 3 Jkg (Table 3?**

**28. Table 1: Authors may consider including additional column with a measured peak wind gust. Also, "Centr" may be replaced with CNTRL for consistency.**

**29. Figure 1. Color scale for orography may be a very confusing and indicate that Alps are up to 1500m.**

**30. Figure 3. I want to compliment authors for this figure. This is an excellent display of various synoptic patterns supporting derecho occurrence over central Europe. However, it may be a bit confusing for readers that dendrogram indicates two patterns and authors display three at the top. It may be useful to add headings at the top of (a), (b) and (c) to denote something like "Warm-type", "11 April 1997", "Cold-type" - in this way it may be easier to catch what is really displayed on in these plots.**

**31. Figure 5. This is a nice figure. Since this is a German derecho climatology I can understand that authors plot provinces, but I am not sure why provinces of Switzerland and Austria are included while others are not? This is not consistent. Perhaps a thicker line of German border would better highlight a country location.**

**32. Figure 6. This is a very nice looking plot, I have two comments regarding this. (1) Did authors think about adding box-plots with some null-cases (e.g. only situations with ML CAPE > 0 J/kg) that can help to see how the distribution of a certain parameters looks on the background of a climatological mean? (see also general comment #2). (2) I see that due to different data ranges ML CAPE is splited into two different plots. Maybe authors can consider using WMAX (square root of two times CAPE) instead of ML CAPE (similar to Figure 6 in https://doi.org/10.1175/JCLI-D-17-0596.1). Values of ML CAPE can be still included on the axis for better readability. This solution may provide a more comfortable range of values and can fit both distributions on a single plot without a loss of details.**

**33. (optional) – given that only 6 figures are in the paper, authors may consider to add one additional plot with a small histograms of (a) derecho path length (b) derecho duration and (c) derecho intensity categories – all which can be extracted from Table 1. It is nice to have this table, but additional histograms would help to asses distributions at certain values. Additional scatterplot may be also added (see general comment #3).**

---

## Referee Comment (RC2) · Anonymous Referee #2 · 24 Nov 2019

General Comments The entitle "An 18-year climatology of derechos in Germany", presents a detailed analysis of derechos climatology based on dataset from 1997 to 2014. All cases were classified to warm (April-Sep) and cold (Oct-Mar) seasons accompanied with proximity sounding parameters. The paper is very interesting and the results are well depicted. I suggest a Minor revision and encourage the authors to adopt the following points that will enhance the manuscript: Minor Points: 3rd page – line22: Although it is an obvious classification (and authors included later in the manuscript) I suggest to authors add at this point the monthly reference period in order to determine the "warm-season" and "cold-season " types (e.g. April to Sept and Oct to Mar, respectively). 4th page – line 24-25: I recommend to authors to add a sentence regarding the wind surface measurements and clarify that all data were based on 10m

or if there were some station with lower measurements (e.g. 3 or 7m) those data were referenced to 10m measurements. Figures 1-6: I recommend to authors to include also the reference period of study. Figure 6: I suggest to authors to join the a and b plots regardless the huge difference of scale.

———————————————————

---

## Author Comment (AC1) · 16 Jan 2020

We thank the reviewer for his/her suggestions on our original submission, which lead to an improved revised manuscript. Please find our detailed replies to your comments below.

General comment #1 from referee 1

What was the logic behind choosing a period from 1997 to 2014? Why not until 2018? At some point it may be even possible to investigate 2019 if there were any such cases, so the paper may include the most recent data.

Author's response

[Figure]

The laborious manual analysis of German derechos was the first part of the first author's Ph.D thesis (Gatzen, Ch., 2018: Climatology and large-scale Dynamics of Derechos in Germany 173 p., https://kups.ub.uni-koeln.de/8275/) and was already completed in 2015. Even though we share the reviewer's view that adding more recent years would provide an added value, this is not feasible due to other professional obligations of the first author. However, we are confident that the main conclusions would not change if 4 further years would be added to our 18-year climatology.

Author's changes in manuscript

No change to the manuscript.

General comment #2 from referee 1

Did author try other indices? I am missing two very important variables. First is a mid-level shear (e.g. 0-3km) that is commonly used in investigating atmospheric potential for convective windstorms (e.g. https://doi.org/10.1175/811.1, https://doi.org/10.1175/1520-0434(2003)18<502:SCWOTN>2.0.CO;2, https://doi.org/10.1007/s00704-018-2728-6, https://doi.org/10.1175/WAF-D-13-00041.1). Second is a mid-level lapse rate (e.g. 800-500 hPa) that may be useful in characterizing cold-seasons derechos, that usually form in a steep temperature gradients. Also, most of the results in the paper are compared to severe wind events from Pucik et al. (2015) and Taszarek et al. (2017), but I am wondering whether authors tried to use a larger number of soundings and define how derecho-related indices differ from the long-term climatological background (null-cases)? Alternatively, this can be derived from a central European sounding-climatology distributions (e.g. https://doi.org/10.1175/JCLI-D-17-0596.1).

Author's response

We thank the reviewer for this comment. As suggested by the reviewer, the mid-level shear and lapse rates were tested. The major goal was to find environmental parameters in proximity soundings that best discriminate between warm- and cold season derechos over Germany. As can be seen from the box-and-whisker plots in Figure R1 below, the 0-1 km (0-6 km) shear is the best (worst) discriminator, and 0-3 km is in between. We also compared 0-3 km shear of derechos to those of different event classes presented in Pucik et al. (P, 2015) and Taszarek et al. (T, 2017). While (P, 2015) analyzed proximity soundings identified for three categories of ESWD convective-wind reports (non-severe below 25 msˆ-1 , severe, and extremely severe above 32 msˆ-1) through-out the year (warm- and cold season), (T, 2017) analyzed proximity soundings identified for four categories of ESWD wind reports (thunderstorms with non-severe gusts below 25 msˆ-1, severe gusts, and extremely severe gusts above 32 msˆ-1, and reports without thunderstorms and CAPE=0) in the warm-season only. To compare the results of warm-season type derecho proximity soundings, (T, 2017) is the best choice as it concentrates on the warm-season; we used the three categories with thunderstorms. For the cold-season type of derechos, (P, 2015) gives context to compare with. In Table 1 (Fig. 7) are the results for the warm-season type of derecho proximity soundings (in msˆ-1).

As can be seen, 0-3 km shear indeed discriminates best between non-severe thunderstorms and derechos, as the upper quartiles and 90th percentile are below the median of warm-season type derechos. However, 0-6 km shear medians differ most for 0-6 km shear. We appreciate the reviewers comment and add the data of 0-3 km shear in the paper as figure and change the text accordingly.

Figure R2 demonstrates that the mid-level lapse rate from proximity soundings is not a suitable distinction parameter between warm- and cold-season type derechos. To enable comparisons with earlier studies, we opted to add this new figure to the Supplementary Material.

We agree that a comparison to a longer-term climatology of the sounding locations, in particular to null cases is meaningful. However, building sounding climatologies is laborious due to missing and erroneous sounding data particularly in winter when

10% of the balloons obviously exploded. As a compromise, we used the climatologies presented in Taszarek et al. (2017) and Pucik et al. (2015). We consider a comparison to these climatologies to be sufficient since our aim was to put our results into the context of severe convective wind events. Furthermore, both studies compare their results with null cases.

Author's changes in manuscript

The main text has been modified to refer to the supplementary figures:

Page 8, lines 2-6 (data and methods) "... we computed 0–1-km vertical wind shear as the magnitude of the vector difference between the wind at 1 km altitude above ground level and the wind closest to the ground that is always referenced to the 10-m (or surface) wind regardless of the exact measurement height. Similarly, we computed the 0–3 and 0–6-km vertical wind shear by the magnitude of the vector difference between the winds at 3 and 6 km altitude above ground level and the surface. We linearly interpolated the wind vectors at 1, 3, and 6 km based in the closest wind measurements near the corresponding heights."

Page 10, line 34 to page 11, line 13 (results; warm-season type derechos) The 0–6-km vertical wind shear in warm-season type derecho proximity soundings in Germany is relatively strong. The median is 20.1 msˆ-1 (Fig. 6d and Table 3) compared to a median of about 13.8 msˆ-1 for severe convective wind gusts indicated by Taszarek et al. (2017), respectively. Additionally, the median of derecho 0–6-km shear is slightly higher than the median of the extremely severe wind-gust category (about 19.5 msˆ-1; Taszarek et al., 2017). Deep-layer shear discriminates between derecho and non-severe thunderstorm environments, as the median of 20.1 msˆ-1 almost reaches the 90th percentile of non-severe thunderstorm environments which is about 20.5 msˆ-1 (Taszarek et al., 2017). The 0–6-km shear's median of derecho environments is also above the upper quartile of the severe convective wind gust category (about 19.5 msˆ-1; Taszarek et al., 2017). Compared to the work of Pucik et al (2015) that includes

cold-season wind events, the median of 0–6-km shear of warm-season type derechos is still rather high compared to the median of the severe wind gust category (16.1 msˆ-1).

We found similar results based on vertical wind shear at lower levels. For 0–3-km shear, the derecho median is 14.9 msˆ-1 (Fig. 6d and Table 3) compared to a median of about 9.8 msˆ-1 and an upper quartile of 14.2 msˆ-1 for severe convective wind gusts and a 90th percentile of 13.9 msˆ-1 for non-severe thunderstorms given by Taszarek et al. (2017). For 0–1-km shear the median increases from about 3.5 msˆ-1 over 5 msˆ-1 to about 5.8 msˆ-1 for respectively non-severe thunderstorms, severe convective wind gusts, and extremely severe convective wind gusts given by Taszarek et al., 2017. It reaches 6.7 msˆ-1 for derechos (Fig. 6c and Table 3). Pucik et al. (2015) indicated a rather strong median of 0–1-km shear for severe convective wind reports that is about the same magnitude than the median for derechos (6.6 msˆ-1 compared to 6.7 msˆ-1). Nonetheless, 0–1-km shear does not discriminate well between derecho environments and that of other categories. For example, the derecho median of 0–1-km shear is only slightly higher than the upper quartile for the non-severe thunderstorm category (6 msˆ-1, Taszarek et al., 2017) and for one proximity derecho sounding, 0–1-km shear is as weak as 1.6 msˆ-1 (Fig. 6c).

Page 13, line 13-19 (results; cold-season type derechos) Cold-season type derecho environments are characterised by exceptionally strong 0–6-km shear that is calculated for 28 of the 31 soundings (Fig. 6). It is at least 22 msˆ-1 and peaks at 59 msˆ-1, the median is 34.5 msˆ-1 (Table 3), what is slightly above the median of 0–6-km shear presented for severe convective-wind environments in Pucik et al. (2015) (33.2 msˆ-1). Strong deep-layer vertical wind shear is also common in cold-season bow-echo environments in the continental United States with a mean 0–5-km shear of 23 msˆ-1 (Burke and Schultz, 2004). For lower level shear, we found a 0–3-km shear median of 23.1 msˆ-1, and a 0–1-km shear of 21.2 msˆ-1 (Fig. 6c and Table 3). The median of 0-1 km shear of cold-season type derechos is therefore slightly higher compared to

cold-season severe convective-wind environments (18.1 msˆ-1, Pucik et al., 2015).

General comment #3 from referee 1

Did authors try producing scatterplots of selected pairs of variables? Combination of mixing ratio and lapse rates, or mixing ratio and 0-3km shear may very clearly indicate differences between cold and warm season derechoes (dwo clusters) and provide possibility to compare "coordinates" of these cases with previous or future studies dealing with convective parameters climatology (e.g. using Kernel Density Estimation of climatological background for soundings). In this way, it can be assessed how rare are such environments (see also general comment #2).

Author's response

Figures R3-R6 display the scatter plots suggested by the Reviewer. Indeed there are regimes for the two different types that are clearly separated, in particular due to the differences in the moisture between warm- and cold-season events. There is also a tendency for higher shear in the cold-season type, but there is still quite an overlap between both types. Finally, the lapse rate vs mixing ratio diagram indicates the difference in the moisture, but no big differences in the lapse rate.

In summary, the scatter plots basically show the clear dependency of the warm- and cold-season event regimes to the seasonal moisture variation. Therefore, we consider the box-and-whisker diagrams to analyse which parameters discriminate best between the two regimes as being sufficient and have decided not change the manuscript.

Author's changes in manuscript

No changes to the manuscript.

General comment #4 from referee 1

I have a feeling that some further discussion on the climatological background of convective environments over Europe, synoptic patterns and overall thunderstorm fre-
quency can be provided. Authors may discuss how unusual is the situation with a derecho given a background of convective activity over central Europe. I believe that the last section of the paper (conclusions) may be trimmed from repeating what was already written in the results, and instead, some space may be devoted for a deeper discussion.

Author's response

Thanks for this comment. We agree with the referee that it can be helpful to put our results into context, and since derechos are convective phenomena, the thunderstorm or CAPE climatology of Germany can provide some context to the derecho climatology. We will include a review on the thunderstorm climatology of Germany, including the annual frequency, spatial distribution, and seasonal variability and compare our results in this context. The annual number and spatial distribution of thunderstorm days (reference period 2001 to 2014) can be obtained from Piper, D. A., Kunz, M., Allen, J. T., & Mohr, S. (2019): "Investigation of the temporal variability of thunderstorms in Central and Western Europe and the relation to large‐scale flow and teleconnection patterns." Quarterly Journal of the Royal Meteorological Society. Additionally, the seasonal cycle of lightning (reference period 2007 to 2012) is presented in Wapler, K. (2013): "High-resolution climatology of lightning in Central Europe." In EGU General Assembly Conference Abstracts (Vol. 15).

Author's changes in manuscript

Page 14, line 19-33 (conclusions) This work analyses the derecho potential across Germany. Based on the presented climatology, the derecho risk is higher than would be expected from previously published cases (four events in 10 years between 2004 and 2014): 40 events were classified in the 18-year period, including 24 moderate and high-end intensity derechos. The highest regional derecho risk occurs in southern Germany with about three moderate and high-end intensity derechos in four years, a value of similar magnitude to the Appalachian mountains in the eastern United States

(Coniglio and Stensrud, 2004). The spatial derecho density maximum across southern Germany overlaps with the maximum of averaged thunderstorm days across Germany (Piper et al., 2019). The highest yearly derecho number in Germany was four in 2003; no derechos formed in 1998 and 2012. German derechos are most frequent in June and July, consistent with the highest seasonal lightning frequency across Germany (Wapler, 2013). The winter season (October–March) contributes on average to 40% of the annual derecho number, although eight of 18 winter seasons had no derechos. The relative contribution of German winter season derechos to the annual derecho number is large compared to the winter season in the United States, which contributes only 25% to the annual derecho number (Ashley and Mote, 2005), and also large with respect to average frequency of lightning in the winter season (Wapler, 2013).

According to a cluster analysis of the 500-hPa flow across Europe, German derechos form in two distinct synoptic-scale situations. Twenty-two derechos correspond to the first derecho type that forms in strong south-westerly 500-hPa flow downstream of an intense trough across the north-eastern Atlantic Ocean and western Europe. Such weather patterns are also favourable for thunderstorms across Germany (Piper et al., 2019). This first derecho type only occurs in the summer months (May to August) and is thus named warm-season type accordingly.

Page 15, line 23-26 (conclusions) The remaining 18 of 40 derechos are of the cold-season type. In contrast to the warm-season type, these derechos form in strong north-westerly 500-hPa flow, mostly at the south-western flank of rapidly amplifying 500-hPa troughs. These troughs are always associated with PV intrusions at 500 hPa, and derechos form close to the PV intrusions. Such weather pattern is typically not associated with frequent thunderstorms across Germany (Piper et al, 2019). Cold-season type derechos have a clear frequency peak from December to February, but some events occur as early as October or as late as May.

Specific comment #1 from referee 1

**1. P1: Abstract has 345 words. A usual standard for scientific publishing is around 200-250 words, authors may want to cut it down a bit.**

Author's response

We agree and reduced the number of words from 345 to 252 words as requested by the reviewer.

Author's changes in manuscript

Abstract. Derechos are high-impact convective wind events that can cause fatalities and widespread losses. In this study, 40 derechos affecting Germany between 1997 and 2014 are analysed to estimate the derecho risk. Similar to the United States, Germany is affected by two derecho types. The first, called warm-season type, forms in strong south-westerly 500-hPa flow downstream of west-European troughs and accounts for 22 of the 40 derechos. It has a peak occurrence in June and July. Warm-season type derechos frequently start in the afternoon and move either eastward along the Alpine forelands or north-eastward across southern central Germany. Associated proximity soundings indicate strong 0–6-km and 0–3-km vertical wind shear and a median of mixed-layer Convective Available Potential Energy (mixed-layer CAPE) around 500 Jkg-1. The second derecho type, the cold-season type, forms in strong north-westerly 500-hPa flow, frequently in association with mid-tropospheric PV intrusions, and accounts for 18 of the 40 derechos. They are associated with a secondary peak from December to February. Cold-season type derechos start over or close to the North Sea and primarily affect north and central Germany; their start time is not strongly related to the peak of diurnal heating. Proximity soundings indicate high-shear–low-CAPE environments. Fifteen warm-season type and nine cold-season type derechos had wind gusts reaching 33 ms-1 in at least at three locations. Although warm-season derechos are more frequent, the path length of cold-season type derechos is on average 1.4 times longer. Thus, these two types of German derechos are likely to have similar impacts.

[Figure]

Specific comment #2 from referee 1

**2: P1 L4: "mid-tropospheric" may be more appropriate than "500-hPa layer" (same for L11), also what authors mean by "intense"?**

Author's response

Thank you very much for this suggestion. However, since we want to address also forecasters, we think that "500-hPa layer" is better than "mid-tropospheric" since 500-hPa charts are used in operational forecasting.

With "intense troughs" we mean that the trough is associated with a strong flow due to sharp geopotential gradients. We appreciate your suggestion and changed the sentence accordingly.

Author's changes in manuscript

The first, called warm-season type, forms in strong south-westerly 500-hPa flow downstream of west-European troughs and accounts for 22 of the 40 derechos.

Specific comment #3 from referee 1

**3. P1 L9: To avoid repetition I suggest "strong 0-6 km wind shear with a median of 20 ms-1 and ..."**

Author's response

Thank you for this suggestion. We deleted this sentence to shorten the abstract.

Author's changes in manuscript

The sentence was deleted in the manuscript.

Specific comment #4 from referee 1

**4: P1 L10: I think that abbreviations shouldn't be used in the abstract, but if authors intend to use CAPE, I suggest to mention "mixed-layer" only for the first time, and later**

[Figure]

just refer to CAPE without adding "mixed-layer" term, also in the whole manuscript body.

Author's response

There are so many different ways to calculate CAPE, and we want to make sure that the reader knows which CAPE we use. Additionally, we compare our results with respect to CAPE of other studies, that sometimes use different CAPE calculations (e.g. MUCAPE). To avoid confusion with respect to the calculation of CAPE, we use the exact abbreviation everywhere.

Author's changes in manuscript

No changes to the manuscript.

Specific comment #5 from referee 1

**5: P1 L16: It is possible that zero CAPE soundings were not representative enough for sampling a "true" environment in which derecho initially occurred? On the basis of what premise authors came to the conclusion that the choose sounding was representative?**

Author's response

We explain the objective method of sounding choosing and also discuss that some of these may be not representative.

Author's changes in manuscript

No changes to the manuscript.

Specific comment #6 from referee 1

**6: P2 L3: What do authors mean by "frequently"? I would suggest to remove this word.**

Author's response

Yes, we agree and appreciate this comment.

Author's changes in manuscript

Page 2, lines 3-6: The high number of casualties is caused by the sudden unexpected occurrence of severe convective storms due to their rapid development (Doswell, 2001), and there are indications that people caught outdoors are not well-prepared.

Specific comment #7 from referee 1

**7: P2 L3-L17: Authors may want to mention also a recent powerful derecho case from Poland from 11 August 2017 where 6 people died being outdoors due to falling trees, including two teenage scouts camping in the forest (https://doi.org/10.1175/MWR-D18-0330.1). This event had a very unique convective evolution and had a remarkable intensity given European windstorms, it was one of the most significant event over the last 20 years. Also, it took place very close to studied area.**

Author's response

We changed this according to the reviewers suggestions by putting the citation in the introduction as a recent example.

Author's changes in manuscript

Page 2, lines 2-3: Other examples are the convective windstorm that killed eight in Berlin and Brandenburg on 10 July 2002 (Gatzen, 2004) and derecho in Poland on 11 August 2017 that caused six fatalities (Taszarek et al., 2019).

Specific comment #8 from referee 1

**8: P2 L12-L17: I don't think references to ESSL twitter reports are needed, authors may just use a reference to ESWD (https://doi.org/10.1016/j.atmosres.2008.10.020) or ESSL (https://doi.org/10.1175/BAMS-D-16-0067.1).**

Author's response

We agree and changed the text according to the reviewer's suggestion.

Author's changes in manuscript

Changed the citations to ESSL (https://doi.org/10.1175/BAMS-D-16-0067.1).

Specific comment #9 from referee 1

**9: P3 L13-L15 & L26-L30: Please see comment #7**

Author's response

We included the suggested citation at several places.

Author's changes in manuscript

Page 3, lines 13-15: In contrast to the United States, the derecho definition is still not commonly used in Europe although several events have been classified in the past decades (Gatzen, 2004; Punkka et al., 2006; López, 2007; Gatzen et al., 2011; Pistotnik et al., 2011; Pucik et al., 2011; Simon et al., 2011; Hamid, 2012; Celinski-Mysław and Matuszko, 2014; Toll et al., 2015; Gospodinov et al., 2015; Mathias et al., 2018, Taszarek et al., 2019).

Page 3, lines 26-30: These are frequently associated with rather high CAPE (Pucik et al., 2011; Hamid, 2012; Celinski-Mysław and Matuszko, 2014, Taszarek et al., 2019) compared to the majority of severe convective wind environments (Pucik et al., 2015). Five case studies of the warm-season derecho type also mention strong vertical wind shear (Gatzen, 2004; Pucik et al., 2011; Hamid, 2012; Celinski-Mysław and Matuszko, 2014, Taszarek et al., 2019). Publications with radar images describe sustained convection initiation along or ahead of a gust front (Gatzen, 2004; Celinski-Mysław and Matuszko, 2014), bow echoes (Gatzen, 2004; Púcik et al., 2011; Hamid, 2012; Celinski-Mysław and Matuszko, 2014, Taszarek et al., 2019), and rear-inflow jets (Gatzen, 2004, Taszarek et al, 2019). Finally, the large-scale flow was frequently from the south-west (e.g. Gatzen, 2004).

Specific comment #10 from referee 1

**10: P5 L14: See general comment #1**

Author's response

Please refer to the response to general comment #1.

Author's changes in manuscript

No changes to the manuscript.

Specific comment #11 from referee 1

**11: P6 L26-L31: Did wind measurements from the water surface were also used to estimate derecho intensity and duration? This may affect results that are presented in second paragraph of section 3.1, especially comparisons with U.S. climatologies where derechos usually do not travel through water surface where friction is lower and promotes stronger wind gusts (that can help reaching derecho criteria).**

Author's response

We discussed this on page 6, lines 7-15: This study differs from previous derecho climatologies because we included over-water stations such as measurements from buoys, ships, and oil rigs. The threshold of 25 ms-1 might be more easily surpassed at open-water stations, because the effect of friction on the wind speed is weaker over the water compared to the land. Wind measurements on oil rigs and ships are not taken at 10 m but at heights up to 100 m above the sea level. This difference in wind measurements might lead to a higher rating of a wind event when it moves across open water compared to the land. Furthermore, the density of wind measurements affects the detection rate of derechos. Derechos can be followed across the North Sea where the measurement density is high enough, in particular across the southern North Sea. Across the central and northern North Sea, there are fewer wind measurements, and it was more difficult to follow derecho tracks farther out than 100 km away from the

German coast. For this reason, derecho paths across the North Sea may be shorter and/or less reliable compared to derecho paths over land.

Author's changes in manuscript

No changes to the manuscript.

Specific comment #12 from referee 1

**12: P7 L9: Maybe it would be better to use a phrase of: "A similar patterns in 500 hPa geopotential field .."**

Author's response

Thank you for your suggestion. We changed the text accordingly.

Author's changes in manuscript

Page 7, line 9: Similar patterns in 500-hPa geopotential field were grouped together using. . .

Specific comment #13 from referee 1

**13: P7 L10-L11: Authors used ERA-Interim reanalysis to display derecho positions, GFS for PV intrusions and NCEP/NCAR reanalysis for clustering method. This is quite a mix. Why authors did not use just one source of data? (e.g. ERA-Interim?). Is it possible that the use of various data sources might have introduced inhomogeneities?**

Author's response

We agree that the use of different reanalysis sources is not ideal. Since we only analyse the synoptic-scale flow of derecho events, inhomogeneities due to the different data sources are very unlikely. The main reason to use different reanalysis sources was convenience. For example, it was easy to display wind gust reports together with various reanalysis fields from ERA-Interim data using software of MeteoGroup. At the same time, agglomerative hierarchical clustering of the scipy.cluster python package

was easiest to do with NCEP reanalysis data.

Author's changes in manuscript

No changes to the manuscript.

Specific comment #14 from referee 1

**14. P7 L12: No need to repeat a word "geopotential".**

Author's response

We agree and changed the text.

Author's changes in manuscript

We used the analysis time closest before the start time of the derecho and normalized the fields with respect to their minimum and maximum geopotential heights in the region of interest.

Specific comment #15 from referee 1

**15. P8 L3: What does exactly mean "10-m or surface"? Wouldn't be better to write that the shear was computed as a magnitude between first available level from the radiosonde up to 1 or 6 km AGL? (I assume this is the case) .**

Author's response

This is correct, since we do not know the exact measurement height of every observation site. We changed the text accordingly.

Author's changes in manuscript

Page 8, lines 2-6 (data and methods) "... we computed 0–1-km vertical wind shear as the magnitude of the vector difference between the wind at 1 km altitude above ground level and the wind closest to the ground that is always referenced to the 10-m (or surface) wind regardless of the exact measurement height."

Specific comment #16 from referee 1

**16. P8 L10: "the year" is not needed (to be consistent with earlier part of the sentence "in 1998 and 2012").**

Author's response

We agree and changed the text.

Author's changes in manuscript

Page 8, line 10: . . .with a range from zero derechos in 1998 and 2012 to four derechos in 2003.

Specific comment #17 from referee 1

**17. P8 L14: "based on the areas of each derecho path"**

Author's response

We agree with this suggestion.

Author's changes in manuscript

The regional derecho frequency was analysed based on the areas of each path.

Specific comment #18 from referee 1

**18. P8 L21-25: Perhaps some short comment on the fraction of warm and cold season derechos in the spatial variability would be good.**

Author's response

Yes, we agree and added one sentence to the paragraph.

Author's changes in manuscript

We added the following sentence at the end of the paragraph (page 8, line 25): At the same time, the fraction of derechos occurring in the cold-season (October to March)

increased from south to north.

Specific comment #19 from referee 1

**19. P10 L31: At this point I would like to highlight once again that among 0-1km, 0-3km and 0-6km shear parameters analyzed in a cited study, 0-3km shear demonstrated the best value in discriminating between extremely severe wind events and other categories. See also general comment #2.**

Author's response

We changed the manuscript accordingly.

Author's changes in manuscript

Please see response to general comment #2.

Specific comment #20 from referee 1

**20. P11 L28-L30: Why authors use pressure values for LFC instead of m AGL? I am not sure if this information adds any value because we don't know what is its relation to the ground level (for each station and in a different synoptic pattern it will be different). Also, we cannot compare it with other studies since majority of them use m AGL. I believe authors should recompute LFC for m AGL. Same for P13 L34.**

Author's response

We agree. We will change that to address forecasters more directly.

Author's changes in manuscript

Changed the figure 6f and Table 3: Instead of LFC [hPa], LFC [m] is given. Changed the text accordingly:

Page 11, line 28-30: Finally, derechos can form when the level of free convection (LFC) is close to the ground (e.g. 1147 m) or at greater heights (e.g. 4227 m). There is no tendency for derechos to form in association with particularly low or high LFCs (Fig. 6f

and Table 3).

Page 13, line 34: Nonetheless, there is a low LFC in many German cold-season type derecho soundings (median 1061 m; Table 3), indicating high relative humidity of the low-level air mass.

Specific comment #21 from referee 1

**21. P12 L8-L12: This shift may be related to a different peak in thunderstorm activity (compared to USA) based on central European climatology of thunderstorms (https://doi.org/10.1007/s00703-013-0285-1, https://doi.org/10.1175/JCLID-18-0372.1, https://doi.org/10.5194/nhess-16-607-2016), and European climatology of convective environments (https://doi.org/10.1175/JCLI-D-17-0596.1, https://doi.org/10.1175/JAMC-D-17-0132.1). Peak in derecho frequency seems to overlap with these estimates.**

Author's response

We appreciate your comment and thank you for the publication links. We will cite Raedler et al., 2018 who gives an overview to lightning occurrence in western and central Europe. The seasonal frequency maximum is similar to that of the warm-season derecho type.

Author's changes in manuscript

Page 12, lines 8-12: Next to differences in sounding parameters, German warm-season type derechos form later in the year compared to the results of Bentley and Sparks (2003), with an annual maximum in early July in Germany compared to May to July in the United States. For the United States, the location of the monthly maximum of derecho activity moves northward in the spring and summer (Bentley and Sparks, 2003). Likewise, the later derecho maximum in Germany may be related to its high latitude and the associated average seasonal CAPE distribution. This is in accordance to the lightning frequency across central Europe (Rädler et al., 2018).

Specific comment #22 from referee 1

**22. P13 L10-P21: Authors may also provide some comment here on a typical high-shear low-CAPE environments that usually induce such events (https://doi.org/10.1175/WAF-D-13-00041.1).**

Author's response

We included the citation and said that the presented soundings indicate high-shear – low-CAPE environments.

Author's changes in manuscript

Page 13, lines 10-21: To characterise the cold-season derecho environment, we analysed 31 proximity soundings (left columns in Table 2). We compared sounding parameters to those of proximity soundings across central Europe given in Pucik et al. (2015) (for severe convective wind events) and across the United States presented by Burke and Schultz (2004) (for cold-season bow echoes). We found high-shear – low-CAPE environments (Sherburn and Parker, 2013): Cold-season type derecho environments are characterised by exceptionally strong 0–6-km shear that is calculated for 28 of the 31 soundings (Fig. 6). It is at least 22 ms-1 and peaks at 59 ms-1, the median is 34.5 ms-1 (Table 3), what is slightly above the median of 0–6-km shear presented for severe convective-wind environments in Pucik et al. (2015) (33.2 ms-1). Strong deep-layer vertical wind shear is also common in cold-season bow-echo environments in the continental United States with a mean 0–5-km shear of 23 ms-1 (Burke and Schultz, 2004). The 0–1-km shear is strong as well with a median of 21.2 ms-1 (Fig. 6c and Table 3) which is slightly higher compared to cold-season severe convective wind environments (18.1 ms-1, Pucik et al., 2015). For cold-season bow echoes in the United States, 0–2.5-km shear is weaker than 0–1-km shear in German derechos. The mean 0–2.5-km shear is 14 ms-1 with 87% of proximity soundings having less than 20 ms-1 shear (Burke and Schultz, 2004).

Specific comment #23 from referee 1

**23. P14 L3: Is it possible to provide which value of ELT is "sufficiently low"? Would it be around -10C? (https://doi.org/10.1175/1520-0493(1996)124<0602:CTGLOF>2.0.CO;2).**

Author's response

Yes, we will include the numbers from the cited publication.

Author's changes in manuscript

Page 14, lines 2-4: Although cold-season type derechos do not need to be associated with thunderstorms by definition, all German events produced lightning, which becomes likely at sufficiently low equilibrium-level temperatures (i.e. below -10°C; van den Broeke et al., 2005).

Specific comment #24 from referee 1

**24. P14 L10: Can authors provide some additional discussion why ELT may rapidly change and briefly explain the mechanism leading to steepening of the temperature lapse rates (in a strongly synoptic-scale forced situations) that can "make" CAPE available? I suppose leading author of the paper may have something to say on this process.**

Author's response

This would be a lot of speculation since the exact processes are not clear to us so far.

Author's changes in manuscript

No changes to the manuscript.

Specific comment #25 from referee 1

**25. P14 L15: replace "than" with "as"**

Author's response

We changed the word according to the reviewer's suggestion.

Author's changes in manuscript

Taking into account that German cold-season type derechos have longer path lengths on average and are of similar intensity than warm-season type derechos, we expected them to have a similar potential impact as warm-season type derechos in Germany.

Specific comment #26 from referee 1

**26. P14 L19-L21: Awkward sentence construction, please split it for two sentences for clarity (after the bracket).**

Author's response

We changed the text according to the suggestion.

Author's changes in manuscript

Based on the presented climatology, the derecho risk is higher than would be expected from previously published cases. Whereas four events had been published in 10 years between 2004 and 2014, 40 events were classified in the 18-year period, including 24 moderate and high-end intensity derechos.

Specific comment #27 from referee 1

**27. P15 L31: Wasn't ML CAPE median 3 Jkg-1 (Table 3?**

Author's response

That is correct! Thank you for this important point. We have fixed this in the revised manuscript

Author's changes in manuscript

Page 14, line 31: Mixed-layer CAPE has a median of 3 Jkg-1 and is zero in 41% of

cold-season type derecho proximity soundings.

Specific comment #28 from referee 1

**28. Table 1: Authors may consider including additional column with a measured peak wind gust. Also, "Centr" may be replaced with CNTRL for consistency.**

Author's response

We thank for these suggestions. We changes "Centr" to "CENTR" in Table 1. However, we did not include the measured peak wind gust.

Author's changes in manuscript

Changed "Centr" to "CENTR" in Table 1.

Specific comment #29 from referee 1

**29. Figure 1. Color scale for orography may be a very confusing and indicate that Alps are up to 1500m.**

Author's response

We will add a "plus" to the last colour layer to indicate that it includes also higher levels.

Author's changes in manuscript

Will include a "plus" to the 1500 m colour layer in the colour bar.

Specific comment #30 from referee 1

**30. Figure 3. I want to compliment authors for this figure. This is an excellent display of various synoptic patterns supporting derecho occurrence over central Europe. However, it may be a bit confusing for readers that dendrogram indicates two patterns and authors display three at the top. It may be useful to add headings at the top of (a), (b) and (c) to denote something like "Warm-type", "11 April 1997", "Cold-type" - in this way it may be easier to catch what is really displayed on in these plots.**
Author's response

Thanks for this suggestion which lead to an improvement of the figure.

Author's changes in manuscript

Changed according to the reviewer's suggestion.

Specific comment #31 from referee 1

**31. Figure 5. This is a nice figure. Since this is a German derecho climatology I can understand that authors plot provinces, but I am not sure why provinces of Switzerland and Austria are included while others are not? This is not consistent. Perhaps a thicker line of German border would better highlight a country location.**

Author's response

We agree that it is better to delete the provinces of Switzerland and Austria in the figure.

Author's changes in manuscript

Changed according to the reviewer's suggestion.

Specific comment #32 (1) from referee 1

**32. Figure 6. This is a very nice looking plot, I have two comments regarding this. (1) Did authors think about adding box-plots with some null-cases (e.g. only situations with ML CAPE > 0 J/kg) that can help to see how the distribution of a certain parameters looks on the background of a climatological mean? (see also general comment #2).**

Author's response

Thanks for this suggestion. We can understand that such information would increase the context of derecho environments. However, we are not able to calculate these data at this time. The main reason is that sounding data needs a quality control that we cannot effort. Soundings presented in this study have been analysed manually to take care of possible measurement errors.

Author's changes in manuscript

No changes made.

Specific comment #32 (2) from referee 1

(2) I see that due to different data ranges ML CAPE is splited into two different plots. Maybe authors can consider using WMAX (square root of two times CAPE) instead of ML CAPE (similar to Figure 6 in https://doi.org/10.1175/JCLI-D-17-0596.1). Values of ML CAPE can be still included on the axis for better readability. This solution may provide a more comfortable range of values and can fit both distributions on a single plot without a loss of details.

Author's response

Thanks for this helpful comment. We will implement a logarithmic scale to have both plots combined. However, we still present CAPE in order to address forecasters that are not familiar with wmax.

Author's changes in manuscript

We now display CAPE of the two types in one plot in figure 6, but in logarithmic scale.

Specific comment #33 from referee 1

**33. (optional) – given that only 6 figures are in the paper, authors may consider to add one additional plot with a small histograms of (a) derecho path length (b) derecho duration and (c) derecho intensity categories – all which can be extracted from Table 1. It is nice to have this table, but additional histograms would help to asses distributions at certain values. Additional scatterplot may be also added (see general comment #3).**

Author's response

We included additional plots to the supplementary material as requested.

Author's changes in manuscript

Added additional plots of derecho path length, derecho duration, derecho intensity categories to the supplementary material.

[Figure]

[Figure]

Fig. R1: Box-and-whisker plots of vertical wind shear of proximity soundings for cold-season (blue boxes) and warm-season (white boxes) derechos.

**Fig. 1.**

[Figure]

**Lapse Rate
800_500 (K/km)**

Fig. R2: Box-and-whisker plot of lapse rates of proximity soundings for cold-season (blue box) and warm-season (white box) derechos.

**Fig. 2.**

[Figure]

Fig. R3: Scatter plot of 0-1 km shear and 0-500 m mixing ratio of proximity soundings for cold-season (symbols in dark blue) and warm-season (symbols in magenta) derechos.

**Fig. 3.**

[Figure]

Fig. R4: Scatter plot of 0-3 km shear and 0-500 m mixing ratio of proximity soundings for cold-season (symbols in dark blue) and warm-season (symbols in magenta) derechos.

**Fig. 4.**

[Figure]

Fig. R5: Scatter plot of 0-6 km shear and 0-500 m mixing ratio of proximity soundings for cold-season (symbols in dark blue) and warm-season (symbols in magenta) derechos.

**Fig. 5.**

[Figure]

Fig. R6: Scatter plot of 800-500 hPa lapse rate and 0-500 m mixing ratio of proximity soundings for cold-season (symbols in dark blue) and warm-season (symbols in magenta) derechos.

**Fig. 6.**

| Warm season | 0-6 km shear | 0-3 km shear | 0-1 km shear |
|---|---|---|---|
| Non-severe median | **~9.4 (T, 2017)**
~13.3 (P, 2015) | ~6.3 (T, 2017)
~9.1 (P, 2015) | ~3.5 (T, 2017)
~5.5 (P, 2015) |
| Non-severe upper quartile | ~15 (T, 2017)
~19.2 (P, 2015) | **~9.8 (T, 2017)**
~13 (P, 2015) | ~6 (T, 2017)
~8.3 (P, 2015) |
| Non-severe 90th percentile | ~20.5 (T, 2017)
~25.4 (P, 2015) | **~13.9 (T, 2017)**
~17.5 (P, 2015) | |
| **Derecho median** | **20.1** | **14.9** | **6.7** |

| Warm season | 0-6 km shear | 0-3 km shear | 0-1 km shear |
|---|---|---|---|
| Severe median | ~13.8 (T, 2017)
16.1 (P, 2015) (also only warm-season) | ~9.8 (T, 2017)
~13 (P, 2015) | ~5 (T, 2017)
6.6 (P, 2015) (also only warm-season) |
| Severe upper quartile | ~19.5 (T, 2017)
~23.7 (P, 2015) | ~14.2 (T, 2017)
~17.5 (P, 2015) | ~8.5 (T, 2017)
~10.9 (P, 2015) |
| **Derecho median** | **20.1** | **14.9** | **6.7** |

| Warm season | 0-6 km shear | 0-3 km shear | 0-1 km shear |
|---|---|---|---|
| Extr. severe median | ~19.5 (T, 2017)
~20 (P, 2015) | ~12.3 (T, 2017)
~14.1 (P, 2015) | ~5.8 (T, 2017)
~7.4 (P, 2015) |
| Extr. severe upper quartile | ~22 (T, 2017)
~24.7 (P, 2015) | ~16.6 (T, 2017)
~19.1 (P, 2015) | ~10.9 (T, 2017)
~11.5 (P, 2015) |
| **Derecho median** | **20.1** | **14.9** | **6.7** |

Table 1: Shear values for summer-season convective categories as taken from Pucik et al. (P, 2015) and Taszarek et al. (T,2018). Shear is given in ms$^{-1}$.

**Fig. 7.**

---

## Author Comment (AC2) · 16 Jan 2020

We thank the reviewer for his/her suggestions on our original submission, which lead to an improved revised manuscript. Please find our detailed replies to your comments below.

Specific comment #1 from referee 2

Minor Points: 3rd page – line22: Although it is an obvious classification (and authors included later in the manuscript) I suggest to authors add at this point the monthly reference period in order to determine the "warm-season" and "cold-season " types (e.g. April to Sept and Oct to Mar, respectively).

Author's response

[Figure]

The occurrence of warm-season and cold-season type events is not limited to the defining seasons (e.g., a cold-season type event can occur in May). Please refer to e.g. page 9, lines 21-22: "The cold-season cluster contains 16 of 17 derechos that occurred between October and April plus the one on 20 May 2006."

Author's changes in manuscript

No changes made to the manuscript.

Specific comment #2 from referee 2

4th page – line 24-25: I recommend to authors to add a sentence regarding the wind surface measurements and clarify that all data were based on 10m or if there were some station with lower measurements (e.g. 3 or 7m) those data were referenced to 10m measurements.

Author's response

We agree with this suggestion. Since there can be indeed differences in the height of surface wind measurements that we do not give, we have added this information in the data and methods section.

Author's changes in manuscript

Page 8, lines 2-6 (data and methods) ". . . we computed 0–1-km vertical wind shear as the magnitude of the vector difference between the wind at 1 km altitude above ground level and the wind closest to the ground that is always referenced to the 10-m (or surface) wind regardless of the exact measurement height. Similarly, we computed the 0–3 and 0–6-km vertical wind shear by the magnitude of the vector difference between the winds at 3 and 6 km altitude above ground level and the surface. We linearly interpolated the wind vectors at 1, 3, and 6 km based in the closest wind measurements near the corresponding heights."

Specific comment #3 from referee 2

Figures 1-6: I recommend to authors to include also the reference period of study.

Author's response

Thank you, we included the reference period as requested.

Author's changes in manuscript

We included the reference period in the figures as requested.

Specific comment #4 from referee 2

Figure 6: I suggest to authors to join the a and b plots regardless the huge difference of scale.

Author's response

Thank you, we changed the figure accordingly: we now display CAPE of the two types in one plot, but in logarithmic scale.

Author's changes in manuscript

We now display CAPE of the two types in one plot in figure 6, but in logarithmic scale.

---

## Author Response (AR1)

Response to reviewer 1

We thank the reviewer for his/her suggestions on our original submission, which lead to an improved .revised manuscript. Please find our detailed replies to your comments below

General comment #1 from referee 1

What was the logic behind choosing a period from 1997 to 2014? Why not until

2018? At some point it may be even possible to investigate 2019 if there were any such

cases, so the paper may include the most recent data.

Author's response

The laborious manual analysis of German derechos was the first part of the first author's Ph.D thesis (Gatzen, Ch., 2018: Climatology and large-scale Dynamics of Derechos in Germany 173 p., https://kups.ub.uni-koeln.de/8275/) and was already completed in 2015. Even though we share the reviewer's view that adding more recent years would provide an added value, this is not feasible due to other professional obligations of the first author. However, we are confident that the main conclusions would not change if 4 further years would be added to our 18-year climatology.

Author's changes in manuscript

No change to the manuscript.

General comment #2 from referee 1

Did author try other indices? I am missing two very important variables. First is a mid-level shear (e.g. 0-3km) that is commonly used in investigating atmospheric potential for convective windstorms (e.g. https://doi.org/10.1175/811.1, https://doi.org/10.1175/1520-0434(2003)18<502:SCWOTN>2.0.CO;2, https://doi.org/10.1007/s00704-018-2728-6, https://doi.org/10.1175/WAF-D-13-00041.1). Second is a mid-level lapse rate (e.g. 800-500 hPa) that may be useful in characterizing cold-seasons derechos, that usually form in a steep temperature gradients. Also, most of the results in the paper are compared to severe wind events from Pucik et al. (2015) and Taszarek et al. (2017), but I am wondering whether authors tried to use a larger number of soundings and define how derecho-related indices differ from the long-term climatological background (null-cases)? Alternatively, this can be derived from a central European sounding-climatology distributions (e.g. https://doi.org/10.1175/JCLI-D-17-0596.1).

Author's response

We thank the reviewer for this comment. As suggested by the reviewer, the mid-level shear and lapse rates were tested. The major goal was to find environmental parameters in proximity soundings that best discriminate between warm- and cold season derechos over Germany. As can be seen from the box-and-whisker plots in Figure R1 below, the 0-1 km (0-6 km) shear is the best (worst) discriminator, and 0-3 km is in between. We also compared 0-3 km shear of derechos to those of different event classes presented in Pucik et al. (P, 2015) and Taszarek et al. (T, 2017). While (P, 2015) analyzed proximity soundings identified for three categories of ESWD convective-wind reports (non-severe below 25 m/s, severe, and extremely severe above 32 m/s) through-out the year (warm- and cold season), (T, 2017) analyzed proximity soundings identified for four categories of ESWD wind reports (thunderstorms with non-severe gusts below 25 m/s, severe gusts, and extremely severe gusts above 32 m/s, and reports without thunderstorms and CAPE=0) in the warm-season only. To compare the results of warm-season type derecho proximity soundings, (T, 2017) is the best choice as it concentrates on the warm-season; we used the three categories with thunderstorms. For the cold-season type of derechos, (P, 2015) gives context to compare with. Below are the results for the warm-season type of derecho proximity soundings.

[Figure]

Fig. R1: Box-and-whisker plots of vertical wind shear of proximity soundings for cold-season (blue boxes) and warm-season (white boxes) derechos.

| Warm season | 0-6 km shear | 0-3 km shear | 0-1 km shear |
|---|---|---|---|
| Non-severe median | ~9.4 (T, 2017)
~13.3 (P, 2015) | ~6.3 (T, 2017)
~9.1 (P, 2015) | ~3.5 (T, 2017)
~5.5 (P, 2015) |
| Non-severe upper quartile | ~15 (T, 2017)
~19.2 (P, 2015) | ~9.8 (T, 2017)
~13 (P, 2015) | ~6 (T, 2017)
~8.3 (P, 2015) |
| Non-severe 90th percentile | ~20.5 (T, 2017)
~25.4 (P, 2015) | ~13.9 (T, 2017)
~17.5 (P, 2015) | |
| Derecho median | 20.1 | 14.9 | 6.7 |

| Warm season | 0-6 km shear | 0-3 km shear | 0-1 km shear |
|---|---|---|---|
| Severe median | ~13.8 (T, 2017)
16.1 (P, 2015) (also only warm-season) | ~9.8 (T, 2017)
~13 (P, 2015) | ~5 (T, 2017)
6.6 (P, 2015) (also only warm-season) |
| Severe upper quartile | ~19.5 (T, 2017)
~23.7 (P, 2015) | ~14.2 (T, 2017)
~17.5 (P, 2015) | ~8.5 (T, 2017)
~10.9 (P, 2015) |
| Derecho median | 20.1 | 14.9 | 6.7 |

| Warm season | 0-6 km shear | 0-3 km shear | 0-1 km shear |
|---|---|---|---|
| Extr. severe median | ~19.5 (T, 2017)
~20 (P, 2015) | ~12.3 (T, 2017)
~14.1 (P, 2015) | ~5.8 (T, 2017)
~7.4 (P, 2015) |
| Extr. severe upper quartile | ~22 (T, 2017)
~24.7 (P, 2015) | ~16.6 (T, 2017)
~19.1 (P, 2015) | ~10.9 (T, 2017)
~11.5 (P, 2015) |
| Derecho median | 20.1 | 14.9 | 6.7 |

As can be seen, 0-3 km shear indeed discriminates best between non-severe thunderstorms and derechos, as the upper quartiles and 90th percentile are below the median of warm-season type derechos. However, 0-6 km shear medians differ most for 0-6 km shear. We appreciate the reviewers comment and add the data of 0-3 km shear in the paper as figure and change the text accordingly.

Figure R2 demonstrates that the mid-level lapse rate from proximity soundings is not a suitable distinction parameter between warm- and cold-season type derechos. To enable comparisons with earlier studies, we opted to add this new figure to the Supplementary Material.

[Figure]

**Lapse Rate
800_500 (K/km)**

Fig. R2: Box-and-whisker plot of lapse rates of proximity soundings for cold-season (blue box) and warm-season (white box) derechos.

We agree that a comparison to a longer-term climatology of the sounding locations, in particular to null cases is meaningful. However, building sounding climatologies is laborious due to missing and erroneous sounding data particularly in winter when 10% of the balloons obviously exploded. As a compromise, we used the climatologies presented in Taszarek et al. (2017) and Pucik et al. (2015). We consider a comparison to these climatologies to be sufficient since our aim was to put our results into the context of severe convective wind events. Furthermore, both studies compare their results with null cases.

**Author's changes in manuscript**

The main text has been modified to refer to the supplementary figures:

**Page 8, lines 2-6 (data and methods)**

[revised manuscript text omitted]

**General comment #3 from referee 1**

Did authors try producing scatterplots of selected pairs of variables? Combination of mixing ratio and lapse rates, or mixing ratio and 0-3km shear may very clearly indicate differences between cold and warm season derechoes (dwo clusters) and provide possibility to compare "coordinates" of these cases with previous or future studies dealing with convective parameters climatology (e.g. using Kernel Density Estimation of climatological background for soundings). In this way, it can be assessed how rare are such environments (see also general comment #2).

[Figure]

Fig. R3: Scatter plot of 0-1 km shear and 0-500 m mixing ratio of proximity soundings for cold-season (symbols in dark blue) and warm-season (symbols in magenta) derechos.

[Figure]

Fig. R4: Scatter plot of 0-3 km shear and 0-500 m mixing ratio of proximity soundings for cold-season (symbols in dark blue) and warm-season (symbols in magenta) derechos.

[Figure]

Fig. R5: Scatter plot of 0-6 km shear and 0-500 m mixing ratio of proximity soundings for cold-season (symbols in dark blue) and warm-season (symbols in magenta) derechos.

[Figure]

Fig. R6: Scatter plot of 800-500 hPa lapse rate and 0-500 m mixing ratio of proximity soundings for cold-season (symbols in dark blue) and warm-season (symbols in magenta) derechos.

**Author's response**

Figures R3-R6 display the scatter plots suggested by the Reviewer. Indeed there are regimes for the two different types that are clearly separated, in particular due to the differences in the moisture between warm- and cold-season events. There is also a tendency for higher shear in the cold-season type, but there is still quite an overlap between both types. Finally, the lapse rate vs mixing ratio diagram indicates the difference in the moisture, but no big differences in the lapse rate.

In summary, the scatter plots basically show the clear dependency of the warm- and cold-season event regimes to the seasonal moisture variation. Therefore, we consider the box-and-whisker diagrams to analyse which parameters discriminate best between the two regimes as being sufficient and have decided not change the manuscript.

**Author's changes in manuscript**

No changes to the manuscript.

**General comment #4 from referee 1**

I have a feeling that some further discussion on the climatological background of convective environments over Europe, synoptic patterns and overall thunderstorm frequency can be provided. Authors may discuss how unusual is the situation with a derecho given a background of convective activity over central Europe. I believe that the last section of the paper (conclusions) may be trimmed from repeating what was already written in the results, and instead, some space may be devoted for a deeper discussion.

**Author's response**

Thanks for this comment. We agree with the referee that it can be helpful to put our results into context, and since derechos are convective phenomena, the thunderstorm or CAPE climatology of Germany can provide some context to the derecho climatology. We will include a review on the thunderstorm climatology of Germany, including the annual frequency, spatial distribution, and seasonal variability and compare our results in this context. The annual number and spatial distribution of thunderstorm days (reference period 2001 to 2014) can be obtained from Piper, D. A., Kunz, M., Allen, J. T., & Mohr, S. (2019): "Investigation of the temporal variability of thunderstorms in Central and Western Europe and the relation to large-scale flow and teleconnection patterns." Quarterly Journal of the Royal Meteorological Society. Additionally, the seasonal cycle of lightning (reference period 2007 to 2012) is presented in Wapler, K. (2013): "High-resolution climatology of lightning in Central Europe." In EGU General Assembly Conference Abstracts (Vol. 15).

**Author's changes in manuscript**

**Page 14, line 19-33 (conclusions)**

This work analyses the derecho potential across Germany. Based on the presented climatology, the derecho risk is higher than would be expected from previously published cases (four events in 10 years between 2004 and 2014): 40 events were classified in the 18-year period, including 24 moderate and high-end intensity derechos. The highest regional derecho risk occurs in southern Germany with about three moderate and high-end intensity derechos in four years, a value of similar magnitude to the Appalachian mountains in the eastern United States (Coniglio and Stensrud, 2004). The spatial derecho density maximum across southern Germany overlaps with the maximum of averaged thunderstorm days across Germany (Piper et al., 2019). The highest yearly derecho number in Germany was four in 2003; no derechos formed in 1998 and 2012. German derechos are most frequent in June and July, consistent with the highest seasonal lightning frequency across Germany (Wapler, 2013). The winter season (October–March) contributes on average to 40% of the annual derecho number, although eight of 18 winter seasons had no derechos. The relative contribution of German winter season derechos to the annual derecho number is large compared to the winter season in the United States, which contributes only 25% to the annual derecho number (Ashley and Mote, 2005), and also large with respect to average frequency of lightning in the winter season (Wapler, 2013).

According to a cluster analysis of the 500-hPa flow across Europe, German derechos form in two distinct synoptic-scale situations. Twenty-two derechos correspond to the first derecho type that forms in strong south-westerly 500-hPa flow downstream of an intense trough across the north-eastern Atlantic Ocean and western Europe. Such weather patterns are also favourable for thunderstorms across Germany (Piper et al., 2019). This first derecho type only occurs in the summer months (May to August) and is thus named warm-season type accordingly.

**Page 15, line 23-26 (conclusions)**

The remaining 18 of 40 derechos are of the cold-season type. In contrast to the warm-season type, these derechos form in strong north-westerly 500-hPa flow, mostly at the south-western flank of rapidly amplifying 500-hPa troughs. These troughs are always associated with PV intrusions at 500 hPa, and derechos form close to the PV intrusions. Such weather pattern is typically not associated with frequent thunderstorms across Germany (Piper et al, 2019). Cold-season type derechos have a clear frequency peak from December to February, but some events occur as early as October or as late as May.

**Specific comment #1 from referee 1**

**#1. P1: Abstract has 345 words. A usual standard for scientific publishing is around 200-250 words, authors may want to cut it down a bit.**

**Author's response**

We agree and reduced the number of words from 345 to 252 words as requested by the reviewer.

**Author's changes in manuscript**

Abstract. Derechos are high-impact convective wind events that can cause fatalities and widespread losses. In this study, 40 derechos affecting Germany between 1997 and 2014 are analysed to estimate the derecho risk. Similar to the United States, Germany is affected by two derecho types. The first, called warm-season type, forms in strong south-westerly 500-hPa flow downstream of west-European troughs and accounts for 22 of the 40 derechos. It has a peak occurrence in June and July. Warm-season type derechos frequently start in the afternoon and move either eastward along the Alpine forelands or north-eastward across southern central Germany. Associated proximity soundings indicate strong 0–6-km and 0–3-km vertical wind shear and a median of mixed-layer Convective Available Potential Energy (mixed-layer CAPE) around 500 Jkg$^{-1}$. The second derecho type, the cold-season type, forms in strong north-westerly 500-hPa flow, frequently in association with mid-tropospheric PV intrusions, and accounts for 18 of the 40 derechos. They are associated with a secondary peak from December to February. Cold-season type derechos start over or close to the North Sea and primarily affect north and central Germany; their start time is not strongly related to the peak of diurnal heating. Proximity soundings indicate high-shear–low-CAPE environments. Fifteen warm-season type and nine cold-season type derechos had wind gusts reaching 33 ms$^{-1}$ in at least at three locations. Although warm-season derechos are more frequent, the path length of cold-season type derechos is on average 1.4 times longer. Thus, these two types of German derechos are likely to have similar impacts.

**Specific comment #2 from referee 1**

**#2: P1 L4: "mid-tropospheric" may be more appropriate than "500-hPa layer"[MOU2] (same for L11), also what authors mean by "intense"?**

Author's response

Thank you very much for this suggestion. However, since we want to address also forecasters, we think that "500-hPa layer" is better than "mid-tropospheric" since 500-hPa charts are used in operational forecasting.

With "intense troughs" we mean that the trough is associated with a strong flow due to sharp geopotential gradients. We appreciate your suggestion and changed the sentence accordingly.

Author's changes in manuscript

The first, called warm-season type, forms in strong south-westerly 500-hPa flow downstream of  west-European troughs and accounts for 22 of the 40 derechos.

**Specific comment #3 from referee 1**

**#3. P1 L9: To avoid repetition I suggest "strong 0-6 km wind shear with a median of 20 ms-1 and …"**

Author's response

Thank you for this suggestion. We deleted this sentence to shorten the abstract.

Author's changes in manuscript

The sentence was deleted in the manuscript.

**Specific comment #4 from referee 1**

**#4: P1 L10: I think that abbreviations shouldn't be used in the abstract, but if authors intend to use CAPE, I suggest to mention "mixed-layer" only for the first time, and later just refer to CAPE without adding "mixed-layer" term, also in the whole manuscript body.**

Author's response

There are so many different ways to calculate CAPE, and we want to make sure that the reader knows which CAPE we use. Additionally, we compare our results with respect to CAPE of other studies, that sometimes use different CAPE calculations (e.g. MUCAPE). To avoid confusion with respect to the calculation of CAPE, we use the exact abbreviation everywhere.

Author's changes in manuscript

No changes to the manuscript.

**Specific comment #5 from referee 1**

**#5: P1 L16: It is possible that zero CAPE soundings were not representative enough for sampling a "true" environment in which derecho initially occurred? On the basis of what premise authors came to the conclusion that the choose sounding was representative?**

Author's response

We explain the objective method of sounding choosing and also discuss that some of these may be not representative.

Author's changes in manuscript

No changes to the manuscript.

**Specific comment #6 from referee 1**

**#6: P2 L3: What do authors mean by "frequently"? I would suggest to remove this word.**

Author's response

Yes, we agree and appreciate this comment.

Author's changes in manuscript

**Page 2, lines 3-6:**

The high number of casualties is caused by the sudden   unexpected occurrence of severe convective storms due to their rapid development (Doswell, 2001), and there are indications that people caught outdoors are not well-prepared.

**Specific comment #7 from referee 1**

**#7: P2 L3-L17: Authors may want to mention also a recent powerful derecho case from Poland from 11 August 2017 where 6 people died being outdoors due to falling trees, including two teenage scouts camping in the forest (https://doi.org/10.1175/MWR-D18-0330.1). This event had a very unique convective evolution and had a remarkable intensity given European windstorms, it was one of the most significant event over the last 20 years. Also, it took place very close to studied area.**

Author's response

We changed this according to the reviewers suggestions by putting the citation in the introduction as a recent example.

Author's changes in manuscript

 Other examples are the convective windstorm that killed eight in Berlin and Brandenburg on 10 July 2002 (Gatzen, 2004) and derecho in Poland on 11 August 2017 that caused six fatalities (Taszarek et al., 2019).

**Specific comment #8 from referee 1**

**#8: P2 L12-L17: I don't think references to ESSL twitter reports are needed, authors may just use a reference to ESWD (https://doi.org/10.1016/j.atmosres.2008.10.020) or ESSL (https://doi.org/10.1175/BAMS-D-16-0067.1).**

Author's response

We agree and changed the text according to the reviewer's suggestion.

Author's changes in manuscript

Changed the citations to ESSL (https://doi.org/10.1175/BAMS-D-16-0067.1).

**Specific comment #9 from referee 1**

**#9: P3 L13-L15 & L26-L30: Please see comment #7**

Author's response

We included the suggested citation at several places.

Author's changes in manuscript

**Page 3, lines 13-15:**

In contrast to the United States, the derecho definition is still not commonly used in Europe although several events have been classified in the past decades (Gatzen, 2004; Punkka et al., 2006; López, 2007; Gatzen et al., 2011; Pistotnik et al., 2011; Pucik et al., 2011; Simon et al., 2011; Hamid, 2012; Celinski-Mysław and Matuszko, 2014; Toll et al., 2015; Gospodinov et al., 2015; Mathias et al., 2018, Taszarek et al., 2019).

**Page 3, lines 26-30:**

These are frequently associated with rather high CAPE (Pucik et al., 2011; Hamid, 2012; Celinski-Mysław and Matuszko, 2014, Taszarek et al., 2019) compared to the majority of severe convective wind environments (Pucik et al., 2015). Five case studies of the warm-season derecho type also mention strong vertical wind shear (Gatzen, 2004; Pucik et al., 2011; Hamid, 2012; Celinski-Mysław and Matuszko, 2014, Taszarek et al., 2019). Publications with radar images describe sustained

convection initiation along or ahead of a gust front (Gatzen, 2004; Celinski-Mysław and Matuszko, 2014), bow echoes (Gatzen, 2004; Púcik et al., 2011; Hamid, 2012; Celinski-Mysław and Matuszko, 2014, Taszarek et al., 2019), and rear-inflow jets (Gatzen, 2004, Taszarek et al, 2019). Finally, the large-scale flow was frequently from the south-west (e.g. Gatzen, 2004).

**Specific comment #10 from referee 1**

**#10: P5 L14: See general comment #1**

Author's response

Please refer to the response to general comment #1.

Author's changes in manuscript

No changes to the manuscript.

**Specific comment #11 from referee 1**

**#11: P6 L26-L31: Did wind measurements from the water surface were also used to estimate derecho intensity and duration? This may affect results that are presented in second paragraph of section 3.1, especially comparisons with U.S. climatologies where derechos usually do not travel through water surface where friction is lower and promotes stronger wind gusts (that can help reaching derecho criteria).**

Author's response

We discussed this on page 6, lines 7-15:

This study differs from previous derecho climatologies because we included over-water stations such as measurements from buoys, ships, and oil rigs. The threshold of 25 ms$^{-1}$ might be more easily surpassed at open-water stations, because the effect of friction on the wind speed is weaker over the water compared to the land. Wind measurements on oil rigs and ships are not taken at 10 m but at heights up to 100 m above the sea level. This difference in wind measurements might lead to a higher rating of a wind event when it moves across open water compared to the land. Furthermore, the density of wind measurements affects the detection rate of derechos. Derechos can be followed across the North Sea where the measurement density is high enough, in particular across the southern North Sea. Across the central and northern North Sea, there are fewer wind measurements, and it was more difficult to follow derecho tracks farther out than 100 km away from the German coast. For this reason, derecho paths across the North Sea may be shorter and/or less reliable compared to derecho paths over land.

Author's changes in manuscript

No changes to the manuscript.

**Specific comment #12 from referee 1**

**#12: P7 L9: Maybe it would be better to use a phrase of: "A similar patterns in 500 hPa geopotential field .."**

Author's response

Thank you for your suggestion. We changed the text accordingly.

Author's changes in manuscript

**Page 7, line 9:**

Similar patterns in 500-hPa geopotential field  were grouped together using...

**Specific comment #13 from referee 1**

**#13: P7 L10-L11: Authors used ERA-Interim reanalysis to display derecho positions, GFS for PV intrusions and NCEP/NCAR reanalysis for clustering method. This is quite a mix. Why authors did not use just one source of data? (e.g. ERA-Interim?). Is it possible that the use of various data sources might have introduced inhomogeneities?**

Author's response

We agree that the use of different reanalysis sources is not ideal. Since we only analyse the synoptic-scale flow of derecho events, inhomogeneities due to the different data sources are very unlikely.

Author's changes in manuscript

No changes to the manuscript.

**Specific comment #14 from referee 1**

**#14. P7 L12: No need to repeat a word "geopotential".**

Author's response

We agree and changed the text.

Author's changes in manuscript

We used the analysis time closest before the start time of the derecho and normalized the  fields with respect to their minimum and maximum geopotential heights in the region of interest.

**Specific comment #15 from referee 1**

**#15. P8 L3: What does exactly mean "10-m or surface"? Wouldn't be better to write that the shear was computed as a magnitude between first available level from the radiosonde up to 1 or 6 km AGL? (I assume this is the case) .**

Author's response

This is correct, since we do not know the exact measurement height of every observation site. We changed the text accordingly.

Author's changes in manuscript

**Page 8, lines 2-6 (data and methods)**

"… we computed 0–1-km vertical wind shear as the magnitude of the vector difference between the wind at 1 km altitude above ground level and the wind closest to the ground that is always referenced to the 10-m (or surface) wind regardless of the exact measurement height."

**Specific comment #16 from referee 1**

**#16. P8 L10: "the year" is not needed (to be consistent with earlier part of the sentence "in 1998 and 2012").**

Author's response

We agree and changed the text.

Author's changes in manuscript

**Page 8, line 10:**

…with a range from zero derechos in 1998 and 2012 to four derechos in  2003.

**Specific comment #17 from referee 1**

**#17. P8 L14: "based on the areas of each derecho path"**

Author's response

We agree with this suggestion.

Author's changes in manuscript

The regional derecho frequency was analysed based on the areas of each  path.

**Specific comment #18 from referee 1**

**#18. P8 L21-25: Perhaps some short comment on the fraction of warm and cold season derechos in the spatial variability would be good.**

Author's response

Yes, we agree and added one sentence to the paragraph.

Author's changes in manuscript

**We added the following sentence at the end of the paragraph (page 8, line 25):**

At the same time, the fraction of derechos occurring in the cold-season (October to March) increased from south to north.

**Specific comment #19 from referee 1**

**#19. P10 L31: At this point I would like to highlight once again that among 0-1km,**

**0-3km and 0-6km shear parameters analyzed in a cited study, 0-3km shear demonstrated the best value in discriminating between extremely severe wind events and**

**other categories. See also general comment #2.**

Author's response

We changed the manuscript accordingly.

Author's changes in manuscript

Please see response to general comment #2.

**Specific comment #20 from referee 1**

**#20. P11 L28-L30: Why authors use pressure values for LFC instead of m AGL? I am not sure if this information adds any value because we don't know what is its relation to the ground level (for each station and in a different synoptic pattern it will be different). Also, we cannot compare it with other studies since majority of them use m AGL. I believe authors should recompute LFC for m AGL. Same for P13 L34.**

**Author's response**

We agree. We will change that to address forecasters more directly.

**Author's changes in manuscript**

Changed the figure 6f and Table 3: Instead of LFC [hPa], LFC [m] is given. Changed the text accordingly:

**Page 11, line 28-30:**

Finally, derechos can form when the level of free convection (LFC) is close to the ground (e.g. 1147 m) or at greater heights (e.g. 4227 m). There is no tendency for derechos to form in association with particularly low or high LFCs (Fig. 6f and Table 3).

**Page 13, line 34:**

Nonetheless, there is a low LFC in many German cold-season type derecho soundings (median 1061 m; Table 3), indicating high relative humidity of the low-level air mass.

**Specific comment #21 from referee 1**

**#21. P12 L8-L12: This shift may be related to a different peak in thunderstorm activity (compared to USA) based on central European climatology of thunderstorms (https://doi.org/10.1007/s00703-013-0285-1, https://doi.org/10.1175/JCLID-18-0372.1, https://doi.org/10.5194/nhess-16-607-2016), and European climatology of convective environments (https://doi.org/10.1175/JCLI-D-17-0596.1, https://doi.org/10.1175/JAMC-D-17-0132.1). Peak in derecho frequency seems to overlap with these estimates.**

**Author's response**

We appreciate your comment and thank you for the publication links. We will cite Raedler et al., 2018 who gives an overview to lightning occurrence in western and central Europe. The seasonal frequency maximum is similar to that of the warm-season derecho type.

**Author's changes in manuscript**

**Page 12, lines 8-12:**

Next to differences in sounding parameters, German warm-season type derechos form later in the year compared to the results of Bentley and Sparks (2003), with an annual maximum in early July in Germany compared to May to July in the United States. For the United States, the location of the monthly maximum of derecho activity moves northward in the spring and summer (Bentley and Sparks, 2003). Likewise, the later derecho maximum in Germany may be related to its high latitude and the associated average seasonal CAPE distribution. This is in accordance to the lightning frequency across central Europe (Rädler et al., 2018).

comment #22 from referee 1

**#22. P13 L10-P21: Authors may also provide some comment here on**

**a typical high-shear low-CAPE environments that usually induce such events**

**(https://doi.org/10.1175/WAF-D-13-00041.1).**

Author's response

We included the citation and said that the presented soundings indicate high-shear – low-CAPE environments.

Author's changes in manuscript

**Page 13, lines 10-21:**

To characterise the cold-season derecho environment, we analysed 31 proximity soundings (left columns in Table 2). We compared sounding parameters to those of proximity soundings across central Europe given in Pucik et al. (2015) (for severe convective wind events) and across the United States presented by Burke and Schultz (2004) (for cold-season bow echoes). We found high-shear – low-CAPE environments (Sherburn and Parker, 2013): Cold-season type derecho environments are characterised by exceptionally strong 0–6-km shear that is calculated for 28 of the 31 soundings (Fig. 6). It is at least 22 ms$^{-1}$ and peaks at 59 ms$^{-1}$, the median is 34.5 ms$^{-1}$ (Table 3), what is slightly above the median of 0–6-km shear presented for severe convective-wind environments in Pucik et al. (2015) (33.2 ms$^{-1}$). Strong deep-layer vertical wind shear is also common in cold-season bow-echo environments in the continental United States with a mean 0–5-km shear of 23 ms$^{-1}$ (Burke and Schultz, 2004). The 0–1-km shear is strong as well with a median of 21.2 ms$^{-1}$ (Fig. 6c and Table 3) which is slightly higher compared to cold-season severe convective wind environments (18.1 ms-1, Pucik et al., 2015). For cold-season bow echoes in the United States, 0–2.5-km shear is weaker than 0–1-km shear in German derechos. The mean 0–2.5-km shear is 14 ms$^{-1}$ with 87% of proximity soundings having less than 20 ms$^{-1}$ shear (Burke and Schultz, 2004).

Specific comment #23 from referee 1

**#23. P14 L3: Is it possible to provide which value of ELT is "sufficiently low"? Would it be around -10C? (https://doi.org/10.1175/1520-0493(1996)124<0602:CTGLOF>2.0.CO;2).**

Author's response

Yes, we will include the numbers from the cited publication.

Author's changes in manuscript

**Page 14, lines 2-4:**

Although cold-season type derechos do not need to be associated with thunderstorms by definition, all German events produced lightning, which becomes likely at sufficiently low equilibrium-level temperatures (i.e. below -10°C; van den Broeke et al., 2005).

Specific comment #24 from referee 1

**#24. P14 L10: Can authors provide some additional discussion why ELT may rapidly**

**change and briefly explain the mechanism leading to steepening of the temperature**

**lapse rates (in a strongly synoptic-scale forced situations) that can "make" CAPE available? I suppose leading author of the paper may have something to say on this process.**

Author's response

This would be a lot of speculation since the exact processes are not clear to us so far.

Author's changes in manuscript

No changes to the manuscript.

Specific comment #25 from referee 1

**#25. P14 L15: replace "than" with "as"**

Author's response

We changed the word according to the reviewer's suggestion.

Author's changes in manuscript

Taking into account that German cold-season type derechos have longer path lengths on average and are of similar intensity than warm-season type derechos, we expected them to have a similar potential impact  as warm-season type derechos in Germany.

Specific comment #26 from referee 1

**#26. P14 L19-L21: Awkward sentence construction, please split it for two sentences or clarity (after the bracket).**

Author's response

We changed the text according to the suggestion.

Author's changes in manuscript

Based on the presented climatology, the derecho risk is higher than would be expected from previously published cases Whereas (four events had been published in 10 years between 2004 and 2014. 40 events were classified in the 18-year period, including 24 moderate and high-end intensity derechos.

Specific comment #27 from referee 1

**#27. P15 L31: Wasn't ML CAPE median 3 Jkg$^{-1}$ (Table 3?**

Author's response

That is correct! Thank you for this important point. We have fixed this in the revised manuscript

Author's changes in manuscript

**Page 14, line 31:**

Mixed-layer CAPE has a median of  3 Jkg$^{-1}$ and is zero in 41% of cold-season type derecho proximity soundings.

Specific comment #28 from referee 1

**#28. Table 1: Authors may consider including additional column with a measured peak**

**wind gust. Also, "Centr" may be replaced with CNTRL for consistency.**

Author's response

We thank for these suggestions. We changes "Centr" to "CENTR" in Table 1. However, we did not include the measured peak wind gust.

Author's changes in manuscript

Changed "Centr" to "CENTR" in Table 1.

**Specific comment #29 from referee 1**

**29. Figure 1. Color scale for orography may be a very confusing and indicate that Alps are up to 1500m.**

**Author's response**

We will add a "plus" to the last colour layer to indicate that it includes also higher levels.

**Author's changes in manuscript**

Will include a "plus" to the 1500 m colour layer in the colour bar.

**Specific comment #30 from referee 1**

**30. Figure 3. I want to compliment authors for this figure. This is an excellent display of various synoptic patterns supporting derecho occurrence over central Europe. However, it may be a bit confusing for readers that dendrogram indicates two patterns and authors display three at the top. It may be useful to add headings at the top of (a), (b) and (c) to denote something like "Warm-type", "11 April 1997", "Cold-type" - in this way it may be easier to catch what is really displayed on in these plots.**

**Author's response**

Thanks for this suggestion which lead to an improvement of the figure.

**Author's changes in manuscript**

Changed according to the reviewer's suggestion.

**Specific comment #31 from referee 1**

**31. Figure 5. This is a nice figure. Since this is a German derecho climatology I can understand that authors plot provinces, but I am not sure why provinces of Switzerland and Austria are included while others are not? This is not consistent. Perhaps a thicker line of German border would better highlight a country location.**

**Author's response**

We agree that it is better to delete the provinces of Switzerland and Austria in the figure.

**Author's changes in manuscript**

Changed according to the reviewer's suggestion.

**Specific comment #32 (1) from referee 1**

**32. Figure 6. This is a very nice looking plot, I have two comments regarding this. (1) Did authors think about adding box-plots with some null-cases (e.g. only situations with ML CAPE > 0 J/kg) that can help to see how the distribution of a certain parameters looks on the background of a climatological mean? (see also general comment #2).**

**Author's response**

Thanks for this suggestion. We can understand that such information would increase the context of derecho environments. However, we are not able to calculate these data at this time. The main reason is that sounding data needs a quality control that we cannot effort. Soundings presented in this study have been analysed manually to take care of possible measurement errors.

**Author's changes in manuscript**

No changes made.

**Specific comment #32 (2) from referee 1**

(2) I see that due to different data ranges ML CAPE is splited into two different plots. Maybe authors can consider using WMAX (square root of two times CAPE) instead of ML CAPE (similar to Figure 6 in https://doi.org/10.1175/JCLI-D-17-0596.1). Values of ML CAPE can be still included on the axis for better readability. This solution may provide a more comfortable range of values and can fit both distributions on a single plot without a loss of details.

**Author's response**

Thanks for this helpful comment. We will implement a logarithmic scale to have both plots combined. However, we still present CAPE in order to address forecasters that are not familiar with wmax.

**Author's changes in manuscript**

We now display CAPE of the two types in one plot in figure 6, but in logarithmic scale.

**Specific comment #33 from referee 1**

**33. (optional) – given that only 6 figures are in the paper, authors may consider to add one additional plot with a small histograms of (a) derecho path length (b) derecho duration and (c) derecho intensity categories – all which can be extracted from Table 1. It is nice to have this table, but additional histograms would help to asses distributions at certain values. Additional scatterplot may be also added (see general comment #3).**

**Author's response**

We included additional plots as requested.

**Author's changes in manuscript**

Added additional plots of derecho path length, derecho duration, derecho intensity categories.

Response to reviewer 2

We thank the reviewer for his/her suggestions on our original submission, which lead to an improved .revised manuscript. Please find our detailed replies to your comments below

Specific comment #1 from referee 2

**Minor Points: 3rd page**

**– line22: Although it is an obvious classification (and authors included later in the manuscript) I suggest to authors add at this point the monthly reference period in order to determine the "warm-season" and "cold-season " types (e.g. April to Sept and Oct to Mar, respectively).**

Author's response

The occurrence of warm-season and cold-season type events is not limited to the defining seasons (e.g., a cold-season type event can occur in May). Please refer to e.g. page 9, lines 21-22: "The cold-season cluster contains 16 of 17 derechos that occurred between October and April plus the one on 20 May 2006."

Author's changes in manuscript

No changes made to the manuscript.

Specific comment #2 from referee 2

**4th page – line 24-25: I recommend to authors to add a sentence regarding the wind surface measurements and clarify that all data were based on 10m or if there were some station with lower measurements (e.g. 3 or 7m) those data were referenced to 10m measurements.**

Author's response

We agree with this suggestion. Since there can be indeed differences in the height of surface wind measurements that we do not give, we have added this information in the data and methods section.

Author's changes in manuscript

**Page 8, lines 2-6 (data and methods)**

"… we computed 0–1-km vertical wind shear as the magnitude of the vector difference between the wind at 1 km altitude above ground level and the wind closest to the ground that is always referenced to the 10-m (or surface) wind regardless of the exact measurement height. Similarly, we computed the

0–3 and 0–6-km vertical wind shear by the magnitude of the vector difference between the winds at 3 and 6 km altitude above ground level and the surface. We linearly interpolated the wind vectors at 1, 3, and 6 km based in the closest wind measurements near the corresponding heights."

Specific comment #3 from referee 2

**Figures 1-6: I recommend to authors to include also the reference period of study.**

Author's response

Thank you, we included the reference period as requested.

Author's changes in manuscript

We included the reference period in the figures as requested.

Specific comment #4 from referee 2

**Figure 6: I suggest to authors to join the a and b plots regardless the huge difference of scale.**

**Author's response**

Thank you, we changed the figure accordingly: we now display CAPE of the two types in one plot, but in logarithmic scale.

**Author's changes in manuscript**

We now display CAPE of the two types in one plot in figure 6, but in logarithmic scale.

%% Copernicus Publications Manuscript Preparation Template for LaTeX Submissions
%DIF LATEXDIFF DIFFERENCE FILE
%DIF DEL Derecho_Climate_Paper.tex          Tue Feb 11 10:10:24 2020
%DIF ADD Derecho_Climate_Paper_reviewed.tex   Tue Feb 11 10:10:22 2020
%% ----------------------------
%% This template should be used for copernicus.cls
%% The class file and some style files are bundled in the Copernicus Latex Package, which can be downloaded from the different journal webpages.
%% For further assistance please contact Copernicus Publications at: production@copernicus.org
%% https://publications.copernicus.org/for_authors/manuscript_preparation.html
%% https://publications.copernicus.org/for_authors/manuscript_preparation.html

%% Please use the following documentclass and journal abbreviations for discussion papers and final revised papers.

%% 2-column papers and discussion papers
\documentclass[journal abbreviation, manuscript]{copernicus}

%% Journal abbreviations (please use the same for discussion papers and final revised papers)

% Advances in Geosciences (adgeo)
% Advances in Radio Science (ars)
% Advances in Science and Research (asr)
% Advances in Statistical Climatology, Meteorology and Oceanography (ascmo)
% Annales Geophysicae (angeo)
% Archives Animal Breeding (aab)
% ASTRA Proceedings (ap)
% Atmospheric Chemistry and Physics (acp)
% Atmospheric Measurement Techniques (amt)
% Biogeosciences (bg)
% Climate of the Past (cp)
% Drinking Water Engineering and Science (dwes)
% Earth Surface Dynamics (esurf)
% Earth System Dynamics (esd)
% Earth System Science Data (essd)
% E&G Quaternary Science Journal (egqsj)
% Fossil Record (fr)
% Geographica Helvetica (gh)
% Geoscience Communication (gc)
% Geoscientific Instrumentation, Methods and Data Systems (gi)
% Geoscientific Model Development (gmd)
% History of Geo- and Space Sciences (hgss)
% Hydrology and Earth System Sciences (hess)
% Journal of Micropalaeontology (jm)
% Journal of Sensors and Sensor Systems (jsss)
% Mechanical Sciences (ms)
% Natural Hazards and Earth System Sciences (nhess)
% Nonlinear Processes in Geophysics (npg)
% Ocean Science (os)
% Primate Biology (pb)
% Proceedings of the International Association of Hydrological Sciences (piahs)
% Scientific Drilling (sd)
% SOIL (soil)
% Solid Earth (se)
% The Cryosphere (tc)
% Web Ecology (we)
% Wind Energy Science (wes)

```
%% \usepackage commands included in the copernicus.cls:
%\usepackage[german, english]{babel}
%\usepackage{tabularx}
%\usepackage{cancel}
%\usepackage{multirow}
%\usepackage{supertabular}
%\usepackage{algorithmic}
%\usepackage{algorithm}
%\usepackage{amsthm}
%\usepackage{float}
%\usepackage{subfig}
%\usepackage{rotating}
%DIF PREAMBLE EXTENSION ADDED BY LATEXDIFF
%DIF UNDERLINE PREAMBLE %DIF PREAMBLE
\RequirePackage[normalem]{ulem} %DIF PREAMBLE
\RequirePackage{color}\definecolor{RED}{rgb}{1,0,0}\definecolor{BLUE}{rgb}{0,0,1} %DIF PREAMBLE
\providecommand{\DIFadd}[1]{{\protect\color{blue}\uwave{#1}}} %DIF PREAMBLE
\providecommand{\DIFdel}[1]{{\protect\color{red}\sout{#1}}}              %DIF PREAMBLE
%DIF SAFE PREAMBLE %DIF PREAMBLE
\providecommand{\DIFaddbegin}{} %DIF PREAMBLE
\providecommand{\DIFaddend}{} %DIF PREAMBLE
\providecommand{\DIFdelbegin}{} %DIF PREAMBLE
\providecommand{\DIFdelend}{} %DIF PREAMBLE
%DIF FLOATSAFE PREAMBLE %DIF PREAMBLE
\providecommand{\DIFaddFL}[1]{\DIFadd{#1}} %DIF PREAMBLE
\providecommand{\DIFdelFL}[1]{\DIFdel{#1}} %DIF PREAMBLE
\providecommand{\DIFaddbeginFL}{} %DIF PREAMBLE
\providecommand{\DIFaddendFL}{} %DIF PREAMBLE
\providecommand{\DIFdelbeginFL}{} %DIF PREAMBLE
\providecommand{\DIFdelendFL}{} %DIF PREAMBLE
%DIF END PREAMBLE EXTENSION ADDED BY LATEXDIFF

\begin{document}

\title{An 18-year climatology of derechos in Germany}

% \Author[affil]{given_name}{surname}

\Author[1]{Christoph P.}{Gatzen}
\Author[2]{Andreas H.}{Fink}
\Author[3]{David M.}{Schultz}
\Author[2]{Joaquim G.}{Pinto}

\affil[1]{Institut für Meteorologie, Freie Universit\"at Berlin, Berlin, Germany}
\affil[2]{Institute of Meteorology and Climate Research - Department Troposphere Research, Karlsruhe
Institute of Technology,
Karlsruhe, Germany
 }
\affil[3
]{Center for Atmospheric Science, School of Earth and Environmental Sciences, University of Manchester,
Manchester, United Kingdom}

%% The [] brackets identify the author with the corresponding affiliation. 1, 2, 3, etc. should be inserted.

\runningtitle{An 18-year climatology of derechos in Germany}
```

\runningauthor{Christoph Gatzen}

\correspondence{Christoph Gatzen (gatzen@met.fu-berlin.de)}

\received{}
\pubdiscuss{} %% only important for two-stage journals
\revised{}
\accepted{}
\published{}

%% These dates will be inserted by Copernicus Publications during the typesetting process.

\firstpage{1}

\maketitle

\begin{abstract}

Derechos are high-impact convective wind events that can cause fatalities and widespread losses. In this study, 40 derechos affecting Germany between 1997 and 2014 are analysed to estimate the derecho risk. Similar to the United States, Germany is affected by two derecho types. The first\DIFdelbegin \DIFdel{derecho typeforms in }\DIFdelend \DIFaddbegin \DIFadd{, called warm-season type, forms in strong }\DIFaddend south-westerly 500-hPa flow downstream of \DIFdelbegin \DIFdel{intense }\DIFdelend west-European troughs and accounts for 22 of the 40 derechos. \DIFdelbegin \DIFdel{These derechos are named warm-season type due to their }\DIFdelend \DIFaddbegin \DIFadd{It has a }\DIFaddend peak occurrence in June and July. Warm-season type derechos frequently start \DIFdelbegin \DIFdel{over southwestern Germany }\DIFdelend in the afternoon and move either eastward along the Alpine forelands or north-eastward across southern central Germany. \DIFdelbegin \DIFdel{Only one warm-season derecho moved across the North Sea and one moved across the Baltic Sea in the 18-year period. Proximity soundings of German warm-season type derechos indicate strong deep-layer }\DIFdelend \DIFaddbegin \DIFadd{Associated proximity soundings indicate strong 0--6-km and 0--3-km }\DIFaddend vertical wind shear \DIFdelbegin \DIFdel{with }\DIFdelend \DIFaddbegin \DIFadd{and }\DIFaddend a median of \DIFdelbegin \DIFdel{20 ms$^{-1}$ 0--6-km shear and }\DIFdelend mixed-layer Convective Available Potential Energy (mixed-layer CAPE) \DIFdelbegin \DIFdel{between 20 and 2600 Jkg$^{-1}$ with a median }\DIFdelend around 500 Jkg$^{-1}$. The second derecho type\DIFdelbegin \DIFdel{forms in north-westerly 500-hPa flow and accounts for 18 of the 40 derechos. These derechos form }\DIFdelend \DIFaddbegin \DIFadd{, the cold-season type, forms }\DIFaddend in strong north-westerly \DIFaddbegin \DIFadd{500-hPa }\DIFaddend flow, frequently in association with mid-tropospheric PV intrusions\DIFaddbegin \DIFadd{, and accounts for 18 of the 40 derechos}\DIFaddend . They are \DIFdelbegin \DIFdel{named cold-season type because they are }\DIFdelend associated with a secondary peak from December to February. Cold-season type derechos start over or close to the North Sea and primarily affect north and central Germany; their start time is not strongly related to the peak of diurnal heating. Proximity soundings indicate high-shear--low-CAPE environments\DIFdelbegin \DIFdel{with a median 0--6-km shear of 35 ms$^{-1}$ and a median mixed-layer CAPE of 3 Jkg$^{-1}$. Environmental CAPE is zero in almost half of cold-season type proximity soundings}\DIFdelend . Fifteen warm-season type and nine cold-season type derechos had wind gusts reaching 33 ms$^{-1}$ in at least at three locations. Although warm-season derechos are more frequent, the path length of cold-season type derechos is on average 1.4 times longer. Thus, these two types of German derechos are likely to have similar impacts.\\
\DIFaddbegin

\DIFaddend \end{abstract}

%\copyrightstatement{TEXT}

\introduction  %% \introduction[modified heading if necessary]

Convective wind events can produce high losses and fatalities in Germany. One example is the Pentecost storm in 2014 \citep{luca2017} with six fatalities in the region of D\"usseldorf in \DIFdelbegin \DIFdel{north-western }\DIFdelend \DIFaddbegin \DIFadd{western }\DIFaddend Germany and high impacts in particular to the railway network. Trains were stopped due to blocked tracks and trees were blown down and hit overhead power lines, severing connections \citep{welt2014}. \DIFdelbegin \DIFdel{Another example is }\DIFdelend \DIFaddbegin \DIFadd{Other examples are }\DIFaddend the convective windstorm that killed eight in Berlin and Brandenburg on 10 July 2002 \citep{Gatzen2004} \DIFaddbegin \DIFadd{and derecho in Poland on 11 August 2017 that caused six fatalities \mbox{%DIFAUXCMD
\citep{taszarek2019derecho}}%DIFAUXCMD
}\DIFaddend . The high number of casualties is caused by the sudden \DIFdelbegin \DIFdel{and frequently }\DIFdelend unexpected occurrence of severe convective storms due to their rapid development \citep{doswell2001}, and there are indications that people caught outdoors are not well-prepared. Open-air events are at risk in particular as demonstrated by incidents such as on 06 July 2001 near Strasbourg, France, when trees fell on tents where visitors took cover, killing 12 \citep[European Severe Weather Database (ESWD),][]{strasbourg,2009ESWD} and at the Pukkelpop open-air festival in Hasselt, Belgium, when estimated wind gusts of 47 ms$^{-1}$ during the 18 August 2011 storm caused five fatalities due to the collapse of tents and parts of the stage \citep{pukkelpop}. A common characteristic of such events is that they travel over large distances of several hundred kilometers, which greatly increases their potential impact due to the large area affected.\\

Data from the European Severe Storms Laboratory (ESSL) can be used to determine the impact of large-scale convective wind events. In 2017, there were 131 fatalities and

 783 injuries due to severe weather \DIFdelbegin \DIFdel{\mbox{%DIFAUXCMD
\citep[including severe winds, heavy rain, tornadoes, and large hail,][]{essl2018a,essl2018b}}%DIFAUXCMD
}\DIFdelend \DIFaddbegin \DIFadd{\mbox{%DIFAUXCMD
\citep[including severe winds, heavy rain, tornadoes, and large hail,][]{essl2018}}%DIFAUXCMD
}\DIFaddend . Of these, 41 fatalities and 456 injuries were caused by the most significant 12 large-scale convective windstorms \DIFdelbegin \DIFdel{\mbox{%DIFAUXCMD
\citep{essl2018c,essl2018d}}%DIFAUXCMD
}\DIFdelend \DIFaddbegin \DIFadd{\mbox{%DIFAUXCMD
\citep{essl2018}}%DIFAUXCMD
}\DIFaddend . The high potential for fatalities and substantial economic losses due to widespread high-impact weather in combination with their rapid development make these convective windstorms among the most challenging forecast situations for weather services and their forecasters. \\

Similar convective windstorms are known in the United States, where the most intense are called \textit{derechos} \DIFdelbegin \DIFdel{\mbox{%DIFAUXCMD
\citep[after][]{Hinrichs1888} }%DIFAUXCMD
\mbox{%DIFAUXCMD
\citep{JohnsHirt1987}}%DIFAUXCMD
}\DIFdelend \DIFaddbegin \DIFadd{\mbox{%DIFAUXCMD
\citep[after \citet{Hinrichs1888};][]{JohnsHirt1987}}%DIFAUXCMD
}\DIFaddend . To classify derechos based on severe wind gust reports and radar reflectivity data, \citet{JohnsHirt1987} require that

 \begin{itemize}
\item deep moist convection is associated with concentrated wind occurrence over a path that extends over at least 400 km along the major length axis. Wind gusts must exceed 26 ms$^{-1}$ as indicated by measurements and/or damage reports,
\item associated wind gusts can be related to the same event, so that a chronological progression of singular wind swaths or a series of wind swaths is indicated,
\item there must be at least 3 reports of wind gusts of at least 33 ms$^{-1}$ (and/or related wind damage) separated by 64 km or more within the area of severe wind gusts, and
\item all wind reports have to occur within 3h of the other wind reports of the same derecho event.
 \end{itemize}

For the United States, \citet{ashley2005} estimate that the derecho hazard potential can be as high as that from tropical storms and tornadoes except for the highest impact tornadoes and hurricanes. Up to four derechos in three years can be expected for 200 km x 200 km grid boxes in the most affected region southwest of the Great Lakes \citep{coniglio2004}. Derechos form in association with mesoscale convective systems that develop bow echoes on plan-view radar displays \citep{fujita1978}. Bow echoes indicate strong flow from the rear to the front of a convective system, a so-called rear-inflow jet \citep[e.g.][] {Smull1987,przybylinski1995}. The development of rear-inflow jets is fostered by persistent convection initiation along the gust front of a convective system \citep{mahoney2009}. Studies on the \DIFdelbegin \DIFdel{environmental }\DIFdelend \DIFaddbegin \DIFadd{ambient }\DIFaddend conditions of convective storms have shown that a combination of high Convective Available Potential Energy (CAPE) and strong vertical wind shear is one environment that can support this upscale growth \citep[e.g.][] {rotunno1988,corfidi2003,klimowski2003}. In particular, a combination of strong shear and high CAPE can occur during May to July in the United States when derechos commonly occur \citep{ashley2005}. The environment may become so favourable (as during heat waves) that derechos may occur in clusters of several events called \textit{derecho families} during the same large-scale flow situation \citep{bentley2003,ashley2004}. However, derechos can also develop when instability is weak \citep{johns1993,burke2004}. Instead of high shear and high CAPE, these derechos form in situations with high shear and low CAPE \citep{sherburn2014} called high-shear--low-CAPE environments \citep[e.g.][] {evans2010,king2014}. \\

In contrast to the United States, the \textit{derecho} definition is still not commonly used in Europe although several events have been classified in the past decades \DIFdelbegin \DIFdel{\mbox{%DIFAUXCMD \citep{Gatzen2004,Punkka2006,Lopez2007,gatzen2011a,pistotnik2011,Puvcik2011,Simon2011,Hamid20 12,Celinski2014,Toll2015,Gospodinov2015,nhess-2018-365}}%DIFAUXCMD }\DIFdelend \DIFaddbegin \DIFadd{\mbox{%DIFAUXCMD \citep{Gatzen2004,Punkka2006,Lopez2007,gatzen2011a,pistotnik2011,Puvcik2011,Simon2011,Hamid20 12,Celinski2014,Toll2015,Gospodinov2015,nhess-2018-365,taszarek2019derecho}}%DIFAUXCMD }\DIFaddend . Four of these derechos affected large portions of Germany \citep{Gatzen2004,gatzen2011a,pistotnik2011,nhess-2018-365}. Before 2004, German convective windstorms were not classified as derechos, but there are indications of potential derechos in the previous literature \citep[e.g.][]{Koeppen1882,Koeppen1896,Faust1948,Kurz1993,Haase-Straub1997,Kaltenboeck2004}. The low number of classified events does not allow for a risk estimate of derechos in Germany, because their characteristics (e.g. the spatio-temporal distribution, typical environmental conditions, large-scale flow patterns) are not known.\\

Based on the case studies listed above, European derechos possess some similar characteristics to derechos in the United States. For example, they can be divided into a ``warm-season type'' and a ``cold-season type''. The majority of European derecho case studies can be attributed to the warm-season type. These are frequently associated with rather high CAPE \DIFdelbegin \DIFdel{\mbox{%DIFAUXCMD \citep{Puvcik2011,Hamid2012,Celinski2014} }%DIFAUXCMD }\DIFdelend \DIFaddbegin \DIFadd{\mbox{%DIFAUXCMD \citep{Puvcik2011,Hamid2012,Celinski2014,taszarek2019derecho} }%DIFAUXCMD }\DIFaddend compared to the \DIFaddbegin \DIFadd{environments of the }\DIFaddend majority of severe convective wind \DIFdelbegin \DIFdel{environments \mbox{%DIFAUXCMD \citep{puvcik2015}}%DIFAUXCMD . Four }\DIFdelend \DIFaddbegin \DIFadd{events \mbox{%DIFAUXCMD \citep{puvcik2015}}%DIFAUXCMD . Five }\DIFaddend case studies of the warm-season derecho type also mention strong vertical wind shear \DIFdelbegin \DIFdel{\mbox{%DIFAUXCMD \citep{Gatzen2004,Puvcik2011,Hamid2012,Celinski2014}}%DIFAUXCMD }\DIFdelend \DIFaddbegin \DIFadd{\mbox{%DIFAUXCMD \citep{Gatzen2004,Puvcik2011,Hamid2012,Celinski2014,taszarek2019derecho}}%DIFAUXCMD }\DIFaddend . Publications with radar images describe sustained convection initiation along or ahead of a gust front \citep{Gatzen2004,Celinski2014}, bow echoes \DIFdelbegin \DIFdel{\mbox{%DIFAUXCMD \citep{Gatzen2004,Puvcik2011,Hamid2012,Celinski2014}}%DIFAUXCMD }\DIFdelend \DIFaddbegin \DIFadd{\mbox{%DIFAUXCMD \citep{Gatzen2004,Puvcik2011,Hamid2012,Celinski2014,taszarek2019derecho}}%DIFAUXCMD }\DIFaddend , and rear-inflow jets \DIFdelbegin \DIFdel{\mbox{%DIFAUXCMD \citep{Gatzen2004}}%DIFAUXCMD }\DIFdelend \DIFaddbegin \DIFadd{\mbox{%DIFAUXCMD \citep{Gatzen2004,taszarek2019derecho}}%DIFAUXCMD }\DIFaddend . Finally, the large-scale flow was frequently from the south-west \citep[e.g.][]{Gatzen2004}. \\

[revised manuscript text omitted]

surface) wind regardless of the exact measurement height\DIFdelbegin \DIFdel{the 10-m (or surface) wind}\DIFdelend . Similarly, we computed the \DIFaddbegin \DIFadd{0--3-km and }\DIFaddend 0--6-km vertical wind shear by the magnitude of the vector difference between the winds at \DIFaddbegin \DIFadd{3 and }\DIFaddend 6 km altitude above ground level and the surface. We linearly interpolated the wind vectors at 1\DIFdelbegin \DIFdel{km }\DIFdelend \DIFaddbegin \DIFadd{, 3, }\DIFaddend and 6 km based in the closest wind measurements near the corresponding heights.\\

\section{Results}

Between 1997 and 2014, 40 derechos at least partly affected Germany (Table \ref{derecho_list}). This is more than two derechos every year on average, with a range from zero derechos \DIFdelbegin \DIFdel{in1998 }\DIFdelend \DIFaddbegin \DIFadd{in 1998 }\DIFaddend and 2012 to four derechos in \DIFdelbegin \DIFdel{the year }\DIFdelend 2003. 
[revised manuscript text omitted]
.~\ref{box-whisker_pressure}). We used \citet{puvcik2015} and \citet{taszarek2017sounding} to compare \DIFdelbegin \DIFdel{the }\DIFdelend \DIFaddbegin \DIFadd{our }\DIFaddend results with a larger set of sounding parameters across central Europe. \citet{puvcik2015} identified 1135 proximity soundings by using a maximum distance of 150 km to severe wind reports occurring up to 3 h after the sounding time. His study covered western and central Europe for March--October \DIFdelbegin \DIFdel{of }\DIFdelend 2008--2013. \citet{taszarek2017sounding} found 828 proximity soundings by using a maximum distance of 125 km and time differences of up to 2 h before and 4

h after the sounding time. This study covered \DIFaddbegin \DIFadd{the }\DIFaddend March--October 2009--2015 warm season across western and central Europe. Additionally, \citet{taszarek2017sounding} present further categories of proximity soundings such as non-severe thunderstorms (8361 soundings), and extremely severe convective wind gusts (33+ ms$^{-1}$; 23 soundings). We compare medians of sounding parameters of both studies with those of German derechos. Furthermore, box-and-whisker plots of German derechos (Fig.~\ref{box-whisker_pressure}) are compared with those presented in \citet[][their Figures 3 and 8]{taszarek2017sounding} to analyse the ability of sounding parameters to discriminate between environments of derechos and other categories of thunderstorms.\\

The 0--6-km vertical wind shear in warm-season \DIFaddbegin \DIFadd{type derecho }\DIFaddend proximity soundings in Germany is relatively strong. The median is 20.1 ms$^{-1}$ (Fig.~\ref{box-whisker_pressure}\DIFdelbegin \DIFdel{d }\DIFdelend \DIFaddbegin \DIFadd{h }\DIFaddend and Table \ref{Table_sounding_medians}) compared to a median of \DIFdelbegin \DIFdel{16.1 and about }\DIFdelend 13.8 ms$^{-1}$ for severe convective wind gusts indicated by \DIFdelbegin \DIFdel{\mbox{%DIFAUXCMD
\citet{puvcik2015} }%DIFAUXCMD
and \mbox{%DIFAUXCMD
\citet{taszarek2017sounding}}%DIFAUXCMD
, respectively}\DIFdelend \DIFaddbegin \DIFadd{\mbox{%DIFAUXCMD
\citet{taszarek2017sounding}}%DIFAUXCMD
}\DIFaddend . Additionally, the median of derecho 0--6-km shear is \DIFaddbegin \DIFadd{slightly }\DIFaddend higher than the median of the extremely severe wind-gust category \DIFdelbegin \DIFdel{\mbox{%DIFAUXCMD
\citep[about 19.5 ms$^{-1}$][]{taszarek2017sounding}}%DIFAUXCMD
}\DIFdelend \DIFaddbegin \DIFadd{\mbox{%DIFAUXCMD
\citep[about 19.5 ms$^{-1}$,][]{taszarek2017sounding}}%DIFAUXCMD
}\DIFaddend . Deep-layer shear discriminates between derecho and non-severe thunderstorm environments, as the median of 20.1 ms$^{-1}$ almost reaches the 90th percentile of non-severe thunderstorm environments which is about 20.5 ms$^{-1}$ \citep{taszarek2017sounding}. The 0--6-km shear\DIFdelbegin \DIFdel{also discriminates between derecho environments and }\DIFdelend \DIFaddbegin \DIFadd{'s median of derecho environments is also above the upper quartile of the }\DIFaddend severe convective wind \DIFdelbegin \DIFdel{environments with }\DIFdelend \DIFaddbegin \DIFadd{gust category \mbox{%DIFAUXCMD
\citep[about 19.5 ms$^{-1}$,][]{taszarek2017sounding}}%DIFAUXCMD
. Compared to the work of \mbox{%DIFAUXCMD
\citet{puvcik2015} }%DIFAUXCMD
that includes cold-season wind events, the median of 0--6-km shear of warm-season type derechos is still rather high compared to the median of the severe wind gust category (16.1 ms$^{-1}$).}\\

\DIFadd{We found similar results based on vertical wind shear at lower levels. For 0--3-km shear, the derecho median is 14.9 ms$^{-1}$ (Fig.~\ref{box-whisker_pressure}g and Table \ref{Table_sounding_medians}) compared to a median of about 9.8 ms$^{-1}$ and }\DIFaddend an upper quartile of \DIFdelbegin \DIFdel{about 19.5 }\DIFdelend \DIFaddbegin \DIFadd{14.2 }\DIFaddend ms$^{-1}$ \DIFdelbegin \DIFdel{\mbox{%DIFAUXCMD
\citep{taszarek2017sounding}}%DIFAUXCMD
. The }\DIFdelend \DIFaddbegin \DIFadd{for severe convective wind gusts and a 90th percentile of 13.9 ms$^{-1}$ for non-severe thunderstorms given by \mbox{%DIFAUXCMD
\citet{taszarek2017sounding}}%DIFAUXCMD
. For the }\DIFaddend 0--1-km shear\DIFdelbegin \DIFdel{indicates a similar tendency. The }\DIFdelend \DIFaddbegin \DIFadd{, the }\DIFaddend median increases from about 3.5 ms$^{-1}$ over 5 ms$^{-1}$ to about 5.8 ms$^{-1}$ for respectively non-severe thunderstorms, severe convective wind gusts, and extremely severe convective wind gusts \DIFdelbegin \DIFdel{\mbox{%DIFAUXCMD
\citep{taszarek2017sounding}}%DIFAUXCMD
}\DIFdelend \DIFaddbegin \DIFadd{given by \mbox{%DIFAUXCMD
\citet{taszarek2017sounding}}%DIFAUXCMD
}\DIFaddend . It reaches 6.7 ms$^{-1}$ for derechos (Fig.~\ref{box-whisker_pressure}\DIFdelbegin \DIFdel{c }\DIFdelend \DIFaddbegin \DIFadd{f }\DIFaddend and Table \ref{Table_sounding_medians}).
\citet{puvcik2015} indicated a rather strong median of 0--1-km shear for severe convective wind reports that is about the same magnitude than the median for derechos (6.6 ms$^{-1}$ compared to 6.7 ms$^{-1}$). Nonetheless, 0--1-km shear does not discriminate well between derecho environments and that of other categories. For example, the derecho median of 0--1-km shear is only slightly higher than the upper quartile for the non-severe thunderstorm category \DIFdelbegin \DIFdel{\mbox{%DIFAUXCMD

\citep[about 6 ms$^{-1}$][]{taszarek2017sounding} }%DIFAUXCMD
}\DIFdelend \DIFaddbegin \DIFadd{\mbox{%DIFAUXCMD
\citep[6 ms$^{-1}$][]{taszarek2017sounding} }%DIFAUXCMD
}\DIFaddend and for one proximity derecho sounding, 0--1-km shear is as weak as 1.6 ms$^{-1}$
(Fig.~\ref{box-whisker_pressure}\DIFdelbegin \DIFdel{c}\DIFdelend \DIFaddbegin \DIFadd{f}\DIFaddend ).\\

In addition to vertical wind shear, the equilibrium-level temperature of derecho soundings could discriminate between non-severe thunderstorm and derecho environments. The median equilibrium-level temperature is --48$^{\circ}$C (Fig.~\ref{box-whisker_pressure}\DIFdelbegin \DIFdel{e }\DIFdelend \DIFaddbegin \DIFadd{b }\DIFaddend and Table \ref{Table_sounding_medians}), which is below the 90th percentile of non-severe thunderstorm environments, and it is also about 20$^{\circ}$C below the medians of severe and extremely severe convective-wind environments \citep{taszarek2017sounding}. The median of mixed-layer CAPE of warm-season type derechos in Germany is 513 Jkg$^{-1}$. This value is higher than the medians of mixed-layer CAPE for non-severe thunderstorms and severe convective wind gusts presented in \DIFdelbegin \DIFdel{\mbox{%DIFAUXCMD
\citep{taszarek2017sounding} }%DIFAUXCMD
}\DIFdelend \DIFaddbegin

 \DIFadd{\mbox{%DIFAUXCMD
\citet{taszarek2017sounding} }%DIFAUXCMD
}\DIFaddend which are about 100 Jkg$^{-1}$ and 250 Jkg$^{-1}$, respectively, but below the median for severe convective wind reports in \citet{puvcik2015} that reaches 695 Jkg$^{-1}$ of most-unstable CAPE\footnote{Most-unstable CAPE and mixed-layer CAPE are calculated differently and cannot be compared directly.}. Furthermore, the upper quartile of mixed-layer CAPE for non-severe thunderstorms is well below the median of derechos \citep[375 Jkg$^{-1}$,][]{taszarek2017sounding}, so that mixed-layer CAPE discriminates between derecho and non-severe thunderstorm environments. Other thermodynamic parameters are less useful to anticipate derecho potential. Low-level mixing ratio is not useful to discriminate between derecho and non-severe thunderstorm environments, as well as between non-severe thunderstorm and severe convective wind gust environments, although its median is slightly higher for German derechos with 10.6 gkg$^{-1}$ (Fig.~\ref{box-whisker_pressure}\DIFdelbegin \DIFdel{g }\DIFdelend \DIFaddbegin \DIFadd{c }\DIFaddend and Table \ref{Table_sounding_medians}) compared to approximately 9.7 gkg$^{-1}$ for non-severe thunderstorms and approximately 10.3 gkg$^{-1}$ for severe convective wind gusts \citep{taszarek2017sounding}. Finally, derechos can form when the level of free convection (LFC) is close to the ground (e.g. \DIFdelbegin \DIFdel{885 hPa}\DIFdelend \DIFaddbegin \DIFadd{1147 m}\DIFaddend ) or at greater heights (e.g. \DIFdelbegin \DIFdel{605 hPa}\DIFdelend \DIFaddbegin \DIFadd{4227 m}\DIFaddend ). There is no tendency for derechos to form in association with particularly low or high LFCs (Fig.~\ref{box-whisker_pressure}\DIFdelbegin \DIFdel{f }\DIFdelend \DIFaddbegin \DIFadd{e }\DIFaddend and Table \ref{Table_sounding_medians}).\\

In summary, German warm-season type derecho environments are characterised by relatively strong vertical wind shear and cold equilibrium levels compared to other convective events across central Europe. Additionally, mixed-layer CAPE is relatively large. Compared to derecho proximity soundings from the United States, 0--6-km vertical wind shear in German warm-season derecho environments is rather strong. For example, the lower and upper quartiles of 0--6-km vertical wind shear of this study are higher compared to the results of \citet{evans2001} (15.9 and 24.7 ms$^{-1}$ compared to 11.8 and 20.0 ms$^{-1}$, respectively). The mixed-layer CAPE values of German warm-season derechos are low compared to those in the United States. For example, for derecho proximity soundings given in \citet{evans2001}, the lower quartile of mixed-layer CAPE\footnote{\citet{evans2001} also include cold-season derechos and calculate mixed-layer CAPE for the lowest 100-m mixed-layer parcel instead of the lowest 500-m mixed layer parcels used in our study.} is 1097 Jkg$^{-1}$ which slightly exceeds the upper quartile in the German distribution (1061 Jkg$^{-1}$, Fig.~\ref{box-whisker_pressure}\DIFdelbegin \DIFdel{b}\DIFdelend \DIFaddbegin \DIFadd{a}\DIFaddend ). Derechos with low CAPE have been also analysed in the United States \citep{evans2001,burke2004}. These events indicated similar mid-level synoptic-scale flow ahead of advancing high-amplitude troughs \citep[strongly-forced synoptic situations;][ their Fig. 1b]{evans2001}. Next to differences in sounding parameters, German warm-season type derechos form later in the year compared to the results of \citet{bentley2003}, with an annual maximum in early July in Germany compared to May to July in the United States. For the United States, the location of the monthly maximum of derecho activity moves northward in the spring and summer \citep{bentley2003}. Likewise, the later derecho maximum in Germany may be related to its high latitude and the associated average seasonal CAPE distribution.

\DIFaddbegin \DIFadd{This is in accordance to the lightning frequency across central Europe
\mbox{%DIFAUXCMD
\citep{radler2018}}%DIFAUXCMD
.}\DIFaddend \\

\subsection{Cold-season type derechos}

Eighteen of the 40 derechos (45\%) are of the cold-season type that mostly form during the secondary annual frequency \DIFdelbegin \DIFdel{maximum }\DIFdelend \DIFaddbegin \DIFadd{maxi\-mum }\DIFaddend from December to February (Fig.~\ref{cluster_season}). Only three derechos are assigned to the cold-season type after 1 March (11 April 1997, 29 April 2002, 20 May 2006). During the winter, diurnal heating is weak in Germany. Consequently, a less distinct diurnal cycle is present for cold-season type derechos in comparison to the warm-season type (Figs.~\ref{Initiationtime}a and \ref{Initiationtime}b). Seven of 18 (39\%) cold-season type derechos start between 18 and 6 UTC (19 and 7 Central European time, respectively), when there is no insolation (Fig.\ \ref{Initiationtime}b). Nonetheless, diurnal heating has some influence as indicated by a sharp frequency maximum in the early afternoon from 12 to 15 UTC (13 to 16 Central European time), when 10 of 18 (56\%) cold-season-type derechos start (Fig.\ \ref{Initiationtime}b). The intensity of the cold-season type is high-end for two of 18 (11\%) and low-end for nine of 18 (50\%) events. On average, cold-season type derechos are slightly weaker than the warm-season type given the smaller fraction of high-end intensity and a larger fraction of low-end intensity compared to the warm-season type (14 and 32\%, respectively). However, the most intense derecho in the dataset is of the cold-season type and occurred on 01 March 2008. Moreover, the average path length of the cold-season type is 880 km, which is 1.4 times the average path length of the warm-season type (path lengths are given in Table \ref{derecho_list}).\\

In contrast to the warm-season type, most cold-season type derechos occur across central Germany (Fig.~\ref{derechos}b). Four events start over the North Sea, and a further 10 start over north-western Germany, Belgium, and the Netherlands. This regional variability could be related to the distance to the North Sea that likely serves as a source of low-level humidity in the cold season. Furthermore, cold-season type derechos move typically from north-west to south-east across Germany; only two have more west--east-oriented paths (Fig.~\ref{derechos}b). The motion of cold-season type derechos is related to the large-scale flow. The cold-season type is associated with a westerly to north-westerly 500-hPa flow in the wake of low geopotential heights across eastern Europe (Fig.~\ref{Cluster_and_dendogram}b, c). A comparable flow pattern has been described for cold-season bow echoes in the United States \citep{johns1982,johns1984synoptic,burke2004}. In many situations, \DIFdelbegin \DIFdel{an intense }\DIFdelend \DIFaddbegin \DIFadd{a }\DIFaddend trough amplifies across Germany during the event, associated with upper-tropospheric PV intrusions \DIFdelbegin \DIFdel{in }\DIFdelend \DIFaddbegin \DIFadd{down to }\DIFaddend the mid-level troposphere (not shown).
Cold-season type derechos occur in 11 out of 19 cold seasons, and some of these seasons are associated with groups of derechos: 2001/02 (three derechos), 2007/08 (two derechos), and 2013/14 (two derechos). There are no derechos in the cold-seasons 1997/98, 2000/01, 2005/06, 2008/09, 2009/10, 2010/11, 2011/12, and 2012/13. \\

To characterise the cold-season derecho environment, we analysed 31 proximity soundings (left columns in Table \ref{Table_sounding_list}). We compared sounding parameters to those of proximity soundings across central Europe given in \citet{puvcik2015} (for severe convective wind events) and across the United States presented by \citet{burke2004} (for cold-season bow echoes). \DIFaddbegin \DIFadd{We found high-shear--low-CAPE environments \mbox{%DIFAUXCMD
\citep{sherburn2014}}%DIFAUXCMD
: }\DIFaddend Cold-season type derecho environments are characterised by exceptionally strong 0--6-km shear that is calculated for 28 of the 31 soundings (Fig.~\ref{box-whisker_pressure}\DIFaddbegin \DIFadd{h}\DIFaddend ). It is at least 22 ms$^{-1}$ and peaks at 59 ms$^{-1}$, the median is 34.5 ms$^{-1}$ (Table \ref{Table_sounding_medians}), \DIFdelbegin \DIFdel{what }\DIFdelend \DIFaddbegin \DIFadd{which }\DIFaddend is slightly above the median of 0--6-km shear presented for severe convective-wind environments in \citet{puvcik2015} (33.2 ms$^{-1}$). Strong deep-layer vertical wind shear is also common in cold-season bow-echo environments in the continental United States with a mean 0--5-km shear of 23 ms$^{-1}$ \citep{burke2004}. \DIFdelbegin \DIFdel{The 0--1-km shearis strong as well with a }\DIFdelend \DIFaddbegin \DIFadd{For lower level shear, we found a 0--3-km shear median of 23.1 ms$^{-1}$, and a 0--1-km shear }\DIFaddend median of 21.2 ms$^{-1}$ (Fig.~\ref{boxwhisker_pressure}\DIFdelbegin \DIFdel{c }\DIFdelend \DIFaddbegin \DIFadd{f,g }\DIFaddend and Table \ref{Table_sounding_medians})\DIFdelbegin \DIFdel{which is }\DIFdelend \DIFaddbegin \DIFadd{. The median of 0--1 km shear of cold-season type derechos is therefore }\DIFaddend slightly higher compared to cold-season severe \DIFdelbegin \DIFdel{convective wind }\DIFdelend \DIFaddbegin \DIFadd{convective-wind }\DIFaddend environments \citep[18.1 ms$^{-1}$,][]{puvcik2015}. For cold-season bow echoes in the United States, 0--2.5-km shear is weaker than 0--1-km shear in German derechos. The mean 0--2.5-km shear is 14 ms$^{-1}$ with 87\% of proximity soundings having less than 20 ms$^{-1}$ shear \citep{burke2004}.\\

Forty-one percent of proximity soundings in German cold-season type derechos have zero mixed-layer CAPE. The maximum and median of mixed-layer CAPE are 231 and 3 Jkg$^{-1}$, respectively, and the lower quartile is 0 Jkg$^{-1}$ (Fig.\ \ref{box-whisker_pressure}a and Table \ref{Table_sounding_medians}). This was not unexpected; proximity

 soundings of severe convective-wind events across central Europe show similar results with a median most-unstable CAPE of 14 Jkg$^{-1}$ \citep{puvcik2015}. Furthermore, although most cold-season bow echoes in the continental United States have higher most-unstable CAPE, there are also long-lived bow-echo proximity soundings with most-unstable CAPE less than 100 Jkg$^{-1}$ \citep{burke2004}. In addition to low mixed-layer CAPE, German cold-season type derecho environments are characterised by low moisture in the lowest 500-m layer above the ground. The highest mean mixing ratio is 8.3 gkg$^{-1}$, the median is 5.2 gkg$^{-1}$ (Table \ref{Table_sounding_medians}), and the lowest mixing ratio is 3.5 gkg$^{-1}$ (Fig.~\ref{box-whisker_pressure}\DIFdelbegin \DIFdel{g}\DIFdelend \DIFaddbegin \DIFadd{c and Table \ref{Table_sounding_medians}}\DIFaddend ). These are rather low values; in the continental United States, \citet{burke2004} presented a minimum of mean lowest 100-hPa mixing ratio of 9 gkg$^{-1}$ for nine long-lived cold-season bow echoes. Nonetheless, there is a low LFC in many German cold-season type derecho soundings (median \DIFdelbegin \DIFdel{867 hPa; }\DIFdelend \DIFaddbegin \DIFadd{1061 m; Fig.~\ref{box-whisker_pressure}e and }\DIFaddend Table \ref{Table_sounding_medians}), indicating high relative humidity of the low-level air mass.\\

The equilibrium-level temperature in proximity soundings of cold-season type derechos is high in some events, peaking at +6$^{\circ}$C, with a median of --7.5$^{\circ}$C (Fig.~\ref{box-whisker_pressure}\DIFdelbegin \DIFdel{e}\DIFdelend \DIFaddbegin \DIFadd{b}\DIFaddend , Table \ref{Table_sounding_medians}). Although cold-season type derechos do not need to be associated with thunderstorms by definition, all German events produced lightning, which becomes likely at sufficiently low equilibrium-level temperatures \DIFdelbegin \DIFdel{\mbox{%DIFAUXCMD
\citep{vandenbroeke2005}}%DIFAUXCMD
}\DIFdelend \DIFaddbegin \DIFadd{\mbox{%DIFAUXCMD
\citep[i.e. below --10$^{\circ}$C;][]{vandenbroeke2005}}%DIFAUXCMD
}\DIFaddend . Soundings with high equilibrium-level temperatures are likely not representative for the particular derecho events for this reason. One example is the derecho proximity sounding at Bergen (WMO number 10238) for 12 UTC 28 December 2001. The sounding has zero mixed-layer CAPE, whereas the 12-UTC SYNOP at the same location reports a thunderstorm with graupel (not shown). Rapid changes of the local equilibrium-level temperature can be expected to occur close to this cold-season type derecho in the time between the \DIFdelbegin \DIFdel{sounding }\DIFdelend ballon launch (1047 UTC) and the thunderstorm observation (1150 UTC). Forecasters have to be aware that soundings may not indicate the potential of convective storms in the vicinity of cold-season type derechos.\\

In summary, cold-season type derechos in Germany are associated with strong vertical wind shear with a median 0--6-km shear of 35 ms$^{-1}$ and a median 0--1-km shear of 21 ms$^{-1}$. At the same time, mixed-layer CAPE is low and soundings frequently have zero mixed-layer CAPE. Taking into account that German cold-season type derechos have longer path lengths on average and are of similar intensity than warm-season type derechos, we expected them to have a similar potential impact \DIFdelbegin \DIFdel{than }\DIFdelend \DIFaddbegin \DIFadd{as }\DIFaddend warm-season type derechos in Germany. \\

\section{Conclusions}

This work analyses the derecho potential across Germany. Based on the presented climatology, the derecho risk is higher than would be expected from previously published cases\DIFdelbegin \DIFdel{(four events }\DIFdelend \DIFaddbegin \DIFadd{. Whereas four events had been published }\DIFaddend in 10

years between 2004 and 2014\DIFdelbegin \DIFdel{): }\DIFdelend \DIFaddbegin \DIFadd{, }\DIFaddend 40 
[revised manuscript text omitted]

%\subsection{Discussion}
%German derecho risk: Need for improved awareness. Most affected regions and time of the year. Importance of cold season.\\

%Comparison of spatio-temporal density with usa. Comparison of seasonal peaks and weather patterns/environmental conditions. Interpretation for German derecho climate. \\

%Interpretation of low-instability derechos: Why do they form? And should we name these derechos althoug they form in different conditions and with different processes?\\

%\subsubsection{HEADING}
%TEXT

%\conclusions  %% \conclusions[modified heading if necessary]
%TEXT

%% The following commands are for the statements about the availability of data sets and/or software code corresponding to the manuscript.
%% It is strongly recommended to make use of these sections in case data sets and/or software code have been part of your research the article is based on.

%\codeavailability{TEXT} %% use this section when having only software code available

\dataavailability{The ERA-Interim reanalysis data are available publicly from their website https://www.ecmwf.int/en/forecasts/datasets/ reanalysis-datasets/era-interim. NCEP Reanalysis 2 data are provided by the NOAA/OAR/ESRL PSD, Boulder,

Colorado, USA, from their website at https://www.esrl.noaa.gov/psd/. Radar and wind data are available via the DWD website (www.dwd.de). The MeteoGroup data are not freely available but can be made available via the provider. All code is available from the authors. }

%\codedataavailability{TEXT} %% use this section when having data sets and software code available

%\sampleavailability{TEXT} %% use this section when having geoscientific samples available

%\appendix
%\section{}    %% Appendix A

%\subsection{}     %% Appendix A1, A2, etc.

%\noappendix       %% use this to mark the end of the appendix section

%% Regarding figures and tables in appendices, the following two options are possible depending on your general handling of figures and tables in the manuscript environment:

%% Option 1: If you sorted all figures and tables into the sections of the text, please also sort the appendix figures and appendix tables into the respective appendix sections.
%% They will be correctly named automatically.

%% Option 2: If you put all figures after the reference list, please insert appendix tables and figures after the normal tables and figures.
%% To rename them correctly to A1, A2, etc., please add the following commands in front of them:

%\appendixfigures  %% needs to be added in front of appendix figures

%\appendixtables   %% needs to be added in front of appendix tables

%% Please add \clearpage between each table and/or figure. Further guidelines on figures and tables can be found below.

\authorcontribution{Christoph P. Gatzen, Andreas H. Fink, David M. Schultz, and Joaquim G. Pinto conceived and designed the research. Christoph P. Gatzen performed the analysis, prepared the figures and wrote the initial draft of the paper. All authors contributed with discussions and revisions.}

\competinginterests{The authors declare that they have no conflict of interest.}

%\disclaimer{TEXT} %% optional section

\begin{acknowledgements}

We thank the ECMWF and NCAR for the provision of
reanalysis data. We thank the DWD for radar and wind data and MeteoGroup for additional wind data and the permission to use the wetter4 display tool. Patrick Ludwig prepared the map presented in figure 1a.
We are grateful to the European Severe Storms Laboratory (ESSL)
for the reports taken from the European Severe Weather Database
(ESWD; http://www.eswd.eu/, last access: 18 November 2018). Joaquim G. Pinto thanks the AXA Research Fund for support.
Partial funding for David M. Schultz was provided to the University of Manchester by the Natural Environment Research Council through Grant NE/ N003918/1.

\end{acknowledgements}

%REFERENCES

%% The reference list is compiled as follows:
%\bibliographystyle{apalike}%{plainnat}
\bibliographystyle{copernicus}%{plainnat}
\bibliography{bibl_ChrisDiss_paper}
\clearpage
%\begin{thebibliography}{}

%\bibitem[AUTHOR(YEAR)]{LABEL1}
%REFERENCE 1

%\bibitem[AUTHOR(YEAR)]{LABEL2}
%REFERENCE 2

%\end{thebibliography}

%% Since the Copernicus LaTeX package includes the BibTeX style file copernicus.bst,
%% authors experienced with BibTeX only have to include the following two lines:
%%
%% \bibliographystyle{copernicus}
%% \bibliography{example.bib}
%%
%% URLs and DOIs can be entered in your BibTeX file as:
%%
%% URL = {http://www.xyz.org/~jones/idx_g.htm}
%% DOI = {10.5194/xyz}

%% LITERATURE CITATIONS
%%
%% command                & example result
%% \citet{jones90}|          & Jones et al. (1990)
%% \citep{jones90}|           & (Jones et al., 1990)
%% \citep{jones90,jones93}|       & (Jones et al., 1990, 1993)
%% \citep[p.~32]{jones90}|       & (Jones et al., 1990, p.~32)
%% \citep[e.g.,][]{jones90}|     & (e.g., Jones et al., 1990)
%% \citep[e.g.,][p.~32]{jones90}| & (e.g., Jones et al., 1990, p.~32)
%% \citeauthor{jones90}|         & Jones et al.
%% \citeyear{jones90}|          & 1990

%% FIGURES

%% When figures and tables are placed at the end of the MS (article in one-column style), please add \clearpage
%% between bibliography and first table and/or figure as well as between each table and/or figure.

%% ONE-COLUMN FIGURES

%%f
%\begin{figure}[t]
%\includegraphics[width=8.3cm]{FILE NAME}
%\caption{TEXT}
%\end{figure}

```
%
%%% TWO-COLUMN FIGURES
%
%%f
%\begin{figure*}[t]
%\includegraphics[width=12cm]{FILE NAME}
%\caption{TEXT}
%\end{figure*}
%
%
%%%TABLES
\begin{table}[btp]
\centering
\caption{Derechos found between 1997 and 2014. Countries of start and end points are abbreviated as GER (Germany), AUT (Austria), FRA (France), SVK (Slovakia), CZE (Czechia), POL (Poland), DEN (Denmark), SUI (Switzerland), BEL (Belgium), NED (Netherlands), SRB (Serbia), SCO (Scotland), ENG (England), CRO (Croatia)\DIFaddbeginFL \DIFaddFL{, respectively}\DIFaddendFL . \label{derecho_list}}
\renewcommand{\arraystretch}{0.75} % this reduces the vertical spacing between rows
\linespread{1.25}\selectfont\centering
\begin{tabular}{lccccrcl}%{P{2.5cm}P{2.2cm}P{1.8cm}P{2cm}rcl}
\hline
Number & Start time (UTC) &      Path length (km) &       Duration (h)      & Intensity       & Path&&\\
\hline
1 &11 April 1997 02:00:00       &1620 &16      &moderate        &NW GER &-& E AUT\\
2 &05 February 1999 00:00:00&660  &10     &moderate        &N GER &-& \DIFdelbeginFL \DIFdelFL{Centr }\DIFdelendFL \DIFaddbeginFL \DIFaddFL{CENTR }\DIFaddendFL GER\\
3 &02 June 1999 14:00:00        &420  &6       &moderate        &SW GER &-& \DIFdelbeginFL \DIFdelFL{Centr }\DIFdelendFL \DIFaddbeginFL \DIFaddFL{CENTR }\DIFaddendFL GER\\
4 &02 June 1999 15:00:00        &600  &7       &high    &E FRA &-& SE GER\\
5 &03 December 1999 15:00:00        &560  &6       &low&  NW GER &-& E GER\\
6 &02 July 2000 14:00:00        &550  &7       &moderate        &N FRA &-& W GER\\
7 &04 July 2000 11:00:00        &630  &7       &moderate        &SE GER &-& N SVK\\
8 &06 July 2001 17:00:00        &450  &6       &moderate        &E FRA &-& \DIFdelbeginFL \DIFdelFL{Centr }\DIFdelendFL \DIFaddbeginFL \DIFaddFL{CENTR }\DIFaddendFL GER\\
9 &28 December 2001 09:00:00        &750  &12      &low     &NW GER &-& N CZE\\
10 &28 January 2002 12:00:00      &930  &12      &low     &N GER &-& SE POL\\
11 &28 January 2002 13:00:00      &430  &5       &low     &NW GER &-& E GER\\
12 &29 April 2002 12:00:00     &410  &6       &low     &W GER &-& E GER\\
13 &10 July 2002 16:00:00      &600  &7       &high    &SE GER &-& E DEN\\
14 &28 January 2003 06:00:00      &440  &6       &low     &W GER &-& SE GER\\
15 &19 May 2003 15:00:00      &620  &6       &low     &N SUI &-& \DIFdelbeginFL \DIFdelFL{Centr }\DIFdelendFL \DIFaddbeginFL \DIFaddFL{CENTR }\DIFaddendFL CZE\\
16 &14 June 2003 06:00:00     &1000 &15      &moderate        &N FRA &-& SE AUT\\
17 &15 December 2003 21:00:00     &700  &6       &low     &N GER &-& E CZE\\
18 &12 August 2004 15:00:00 &590  &9       &low     &N SUI &-& E AUT\\
19 &19 November 2004 06:00:00     &850  &12      &moderate        &S GER &-& SE POL\\
20 &12 February 2005 15:00:00     &610  &7       &low             &S BEL &-& SE GER\\
21 &29 July 2005 14:00:00      &740  &11      &high    &NW SUI &-& E GER\\
22 &20 May 2006 12:00:00      &600  &8       &moderate        &W GER &-& NW CZE\\
23 &18 January 2007 13:00:00      &1100 &11      &moderate        &N NED &-& SE POL\\
24 &21 June 2007 06:00:00      &1020 &15      &moderate        &W SUI &-& N SVK\\
25 &22 February 2008 15:00:00     &1150 &13      &moderate        &NW

 DEN &-& E POL\\
26 &01 March 2008 02:00:00  &1520 &16      &high    &W NED &-& \DIFdelbeginFL \DIFdelFL{Centr }\DIFdelendFL \DIFaddbeginFL \DIFaddFL{CENTR }\DIFaddendFL SRB\\
27 &25 June 2008 16:00:00      &570  &6       &low             &E GER &-& \DIFdelbeginFL \DIFdelFL{Centr }\DIFdelendFL \DIFaddbeginFL \DIFaddFL{CENTR }\DIFaddendFL SVK\\
28 &26 May 2009 12:00:00      &570  &8       &moderate        &SW SUI &-& SW CZE\\
29 &23 July 2009 17:00:00      &520  &7       &moderate        &SE GER &-& N SVK\\
30 &23 July 2009 15:00:00      &670  &7       &low             &E GER &-& \DIFdelbeginFL \DIFdelFL{Centr
```

```
}\DIFdelendFL \DIFaddbeginFL \DIFaddFL{CENTR }\DIFaddendFL POL\\
31 &12 July 2010 09:00:00     &650   &6      &moderate      &S BEL &-& SW DEN\\
32 &14 July 2010 12:00:00     &500   &5      &moderate      &\DIFdelbeginFL
\DIFdelFL{Centr }\DIFdelendFL \DIFaddbeginFL \DIFaddFL{CENTR }\DIFaddendFL FRA &-&
\DIFdelbeginFL \DIFdelFL{Centr }\DIFdelendFL \DIFaddbeginFL \DIFaddFL{CENTR }\DIFaddendFL NED\\
33 &22 June 2011 12:00:00     &450   &6      &low           &     SW GER &-& E GER\\
34 &22 June 2011 13:00:00     &550   &5.5    &moderate      &     E FRA &-& SE GER\\
35 &24 August 2011 16:00:00 &440   &6      &low           &     W GER &-& E GER\\
36 &04 August 2013 10:00:00 &790   &10     &low           &     N SUI &-& W SVK\\
37 &06 August 2013 12:00:00 &750   &10     &moderate      &     SW GER &-& E GER\\
38 &05 December 2013 05:00:00       &1160&      13      &high  &     N SCO &-& NW GER\\
39 &03 January 2014 14:00:00        &680& 7      &low           &     N FRA &-& N GER\\
40 &21 October 2014 13:00:00        &1600 &15     &moderate      &S ENG &-& E CRO\\
\end{tabular}
\end{table}
\clearpage

\begin{table}[btp]
\caption[Proximity soundings of warm- and cold-season type derechos between 1997 and 2014]{Proximity
soundings of derechos between 1997 and 2014. The left column lists soundings of cold-season type
derechos, the right colmun lists soundings of warm-season type derechos. Sounding sites are given by their
WMO station identifier code. Soundings marked with * are chosen although the derecho moved across the
sounding site 3 h after the sounding launch instead of 2 h. For the sounding marked with **
\DIFdelbeginFL \DIFdelFL{is }\DIFdelendFL \DIFaddbeginFL \DIFaddFL{there }\DIFaddendFL was no clear
evidence that it was crossed by the derecho before or after the launch time.\label{Table_sounding_list}}
\renewcommand{\arraystretch}{0.85} % this reduces the vertical spacing between rows
\linespread{1.5}\selectfont\centering
\scriptsize
\footnotesize
\begin{tabular}{lcclcc}%{P{2.3cm}P{1.3cm}P{3.cm}P{2.3cm}P{1.3cm}P{3.cm}}
\hline
Date / Time (UTC) &     Sounding \#    &       Date / Time (UTC) &      Sounding \# \\
\hline
11 April 1997 12:00:00&          12425**              &02 June 1999 12:00:00       &10618        \\
05 February 1999 00:00:00&   10035          &02 June 1999 12:00:00       &10739        \\
05 February 1999 06:00:00&   10393   &04 July 2000 12:00:00       &11035        \\
05 February 1999 12:00:00&   11520*          &02 July 2000 12:00:00       &07145        \\
05 February 1999 12:00:00&   10771   &04 July 2000 12:00:00       &10868        \\
28 December 2001 12:00:00& 10238   &07 July 2001 12:00:00       &10868        \\
28 December 2001 12:00:00& 10410          &10 July 2002 18:00:00       &10393        \\
28 December 2001 18:00:00& 11520          &19 May 2003 18:00:00       &11520        \\
28 January 2002 12:00:00&    10410          &14 June 2003 12:00:00       &10739        \\
28 January 2002 18:00:00&    10393          &29 July 2005 12:00:00       &06610        \\
29 April 2002 12:00:00&      10410          &30 July 2005 00:00:00       &12425        \\
29 April 2002 12:00:00&      06260          &21 June 2007 12:00:00       &10739        \\
28 January 2003 12:00:00&    11035          &26 May 2009 18:00:00       &10771        \\
28 January 2003 12:00:00&    06610          &26 May 2009 18:00:00       &11520        \\
15 December 2003 00:00:00& 11520          &23 July 2009 12:00:00       &11520        \\
15 December 2003 00:00:00& 10548          &22 June 2011 12:00:00       &10618        \\
19 November 2004 06:00:00& 10771          &22 June 2011 12:00:00       &10739        \\
19 November 2004 12:00:00& 11952*          &22 June 2011 18:00:00       &10771        \\
12 February 2005 18:00:00&   10618          &22 June 2011 18:00:00       &10393        \\
20 May 2006 12:00:00&        10410          &24 August 2011 18:00:00       &10548        \\
20 May 2006 12:00:00&        07145          &04 August 2013 12:00:00       &10739        \\
20 May 2006 12:00:00&        06260          &06 August 2013 12:00:00       &10739        \\
20 May 2006 18:00:00&        10771          &06 August 2013 12:00:00       &06610        \\
18 January 2007 18:00:00&    10393          &&&      \\
23 February 2008 00:00:00&   10393          &&&      \\
23 February 2008 00:00:00&   12425*          &&&      \\
01 March 2008 00:00:00&      06260*          &&&      \\
```

```
01 March 2008 06:00:00&      10618          &&&     \\
05 December 2013 12:00:00& 10035*              &&&     \\
21 October 2014 12:00:00&   07145          &&&     \\
21 October 2014 18:00:00&   10771          &&&     \\
%\hline
\end{tabular}
\end{table}
\normalsize
\clearpage

\begin{table}[btp]
\caption[Median values of sounding parameters derived from derechos]{Median values of sounding
parameters derived from derechos of the warm-season (top) and cold-season (bottom) type. From left: Mixed-
layer CAPE (MLCAPE), mixing ratio (MIXR), 0--6-km shear (\DIFdelbeginFL \DIFdelFL{DLS}\DIFdelendFL
\DIFaddbeginFL \DIFaddFL{SHR 0--6), 0--3-km shear (SHR 0--3)\DIFaddendFL ), 0--1-km shear
(\DIFdelbeginFL \DIFdelFL{LLS}\DIFdelendFL \DIFaddbeginFL \DIFaddFL{SHR 0--1}\DIFaddendFL ),
equilibrium-level temperature (ELT), and level of free convection (LFC).\label{Table_sounding_medians}}
\renewcommand{\arraystretch}{0.85} % this reduces the vertical spacing between rows
\linespread{1.5}\selectfont\centering
\scriptsize
\footnotesize
\DIFdelbeginFL
\DIFdelendFL \DIFaddbeginFL \begin{tabular}{lccccccc}\DIFaddendFL
\hline
Derecho type & MLCAPE         & MIXR &         \DIFdelbeginFL \DIFdelFL{DLS }\DIFdelendFL
\DIFaddbeginFL \DIFaddFL{SHR 0--6 }\DIFaddendFL &  \DIFdelbeginFL \DIFdelFL{LLS  }\DIFdelendFL
\DIFaddbeginFL \DIFaddFL{SHR 0--3 }& \DIFaddFL{SHR 0--1 }\DIFaddendFL & ELT & LFC\\
\hline
warm-season&  513 Jkg$^{-1}$ &10.62 gkg$^{-1}$        &20.1 ms$^{-1}$          &\DIFaddbeginFL
\DIFaddFL{14.9 ms$^{-1}$ }&\DIFaddendFL 6.7 ms$^{-1}$       &-48$^{\circ}$C          &753 hPa \\
cold-season&    3 Jkg$^{-1}$ &5.16 gkg$^{-1}$        &34.5 ms$^{-1}$          &\DIFaddbeginFL
\DIFaddFL{23.1 ms$^{-1}$ }&\DIFaddendFL 21.2 ms$^{-1}$     &-7.5$^{\circ}$C         &867 hPa \\
\end{tabular}
\end{table}
\normalsize
\clearpage

\begin{figure}[t]
\includegraphics[width=18.3cm]{pictures/Fig_1_topo_density.pdf}
\caption{(a) Topography of Germany. Terrain height [m] is given by shading according to the colour bar.
Mountains mentioned in the text are highlighted by numbers: 1 - Alps, 2 - Swiss Jura, 3 - Vosges, 4 - Black
Forest, 5 - Swabian Jura, 6 - Frankonian Jura. (b) Number of moderate and high-end intensity derechos
affecting grid boxes with 200 km side length across Germany per year \DIFaddbeginFL \DIFaddFL{(1997--
2014)}\DIFaddendFL .}
\label{Fig_1_topo_density}
\end{figure}
\clearpage

\begin{figure}[t]
\DIFaddbeginFL \includegraphics[width=14.3cm]{pictures/derecho_characteristics.pdf}
\caption{\DIFaddFL{Characteristics of derechos (1997--2014): (a) Intensity, (b) path length, and (c)
duration.}}
\label{derecho_characteristics}
\end{figure}
\clearpage

\begin{figure}[t]
\DIFaddendFL \includegraphics[width=15.3cm]{pictures/cluster_season_1.png}
\caption{Seasonal distribution of \DIFdelbeginFL \DIFdelFL{derechos }\DIFdelendFL \DIFaddbeginFL
\DIFaddFL{derecho events }\DIFaddendFL in Germany \DIFaddbeginFL \DIFaddFL{(1997--
```

2014)}\DIFaddendFL .}
\label{cluster_season}
\end{figure}

\begin{figure}[t]
\includegraphics[width=14.3cm]{pictures/Cluster_and_dendogram.pdf}
\caption{(a) Averaged 500-hPa geopotential field of the warm-season type derecho cluster
\DIFaddbeginFL \DIFaddFL{(1997--2014)}\DIFaddendFL . Isohypses are diplayed in 80 m intervals; the 5600
m isohypse is highlighted by the thick line. (b) 500-hPa geopotential field of 11 April 1997 00:00:00 UTC.
\DIFdelbeginFL \DIFdelFL{Isohypses are diplayed in 80 m intervals; the 5600 m isohypse is highlighted by
the thick line. }\DIFdelendFL (c) Averaged 500-hPa geopotential field of the cold-season type derecho
cluster \DIFdelbeginFL \DIFdelFL{. Isohypses are diplayed in 80 m intervals; the 5600 m isohypse is
highlighted by the thick line}\DIFdelendFL \DIFaddbeginFL \DIFaddFL{(1997--2014)}\DIFaddendFL . (d)
Dendrogram of the cluster analysis of 500-hPa geopotential fields in German derecho events
\DIFaddbeginFL \DIFaddFL{(1997--2014)}\DIFaddendFL . Dates of the analysis fields are given below each
event line. The \DIFdelbeginFL \DIFdelFL{labels }\DIFdelendFL \DIFaddbeginFL
\DIFaddFL{names }\DIFaddendFL of the clusters (a)--(c) are \DIFaddbeginFL \DIFaddFL{also }\DIFaddendFL
given in the dendogram. The maps in (a)--(c) have been produced with the python module basemap.}
\label{Cluster_and_dendogram}
\end{figure}

\begin{figure}[t]
\includegraphics[width=16.3cm]{pictures/Intiationtime.pdf}
\caption{Start time (UTC) of derechos \DIFaddbeginFL \DIFaddFL{(1997--2014) }\DIFaddendFL of the warm-
season type (a) and the cold-season type (b).}
\label{Initiationtime}
\end{figure}

\begin{figure}[t]
\includegraphics[width=12.3cm]{pictures/derechos.pdf}
\caption{Tracks of derechos \DIFaddbeginFL \DIFaddFL{(1997--2014) }\DIFaddendFL of the warm-season
type (a) and the cold-season type (b). Thick lines, thin lines, and broken lines relate to high-end, moderate,
and low-end intensity derechos.}
\label{derechos}
\end{figure}

\begin{figure}[t]
\includegraphics[width=18.3cm]{pictures/box-whisker_pressure_neu.pdf}
\caption{Box-and-whisker diagrams of warm-season type and cold-season type derecho proximity
soundings \DIFaddbeginFL \DIFaddFL{(1997--2014)}\DIFaddendFL . Boxes (blue for the cold-season type)
give the upper and lower quartiles of the distributions, the end of the \DIFaddbeginFL
\DIFaddFL{vertical }\DIFaddendFL thin lines maxima and minima, and the \DIFaddbeginFL
\DIFaddFL{horizontal }\DIFaddendFL thick lines the medians. (a) Mixed-layer CAPE\DIFdelbeginFL
\DIFdelFL{for the cold-season type}\DIFdelendFL ; (b) \DIFdelbeginFL \DIFdelFL{Mixed-layer CAPE for the
warm-season type}\DIFdelendFL \DIFaddbeginFL \DIFaddFL{Equilibrium-level temperature}\DIFaddendFL ;
(c) \DIFdelbeginFL \DIFdelFL{0--1-km vertical wind shear}\DIFdelendFL \DIFaddbeginFL \DIFaddFL{0--500-m
mixing ratio}\DIFaddendFL ; (d) \DIFdelbeginFL \DIFdelFL{0--6-km vertical wind shear}\DIFdelendFL

 \DIFaddbeginFL \DIFaddFL{800--500 hPa lapse rate}\DIFaddendFL ; (e) \DIFdelbeginFL
\DIFdelFL{Equilibrium-level temperature; (f) }\DIFdelendFL Level of free convection; (\DIFaddbeginFL
\DIFaddFL{f) 0--1-km vertical wind shear; (}\DIFaddendFL g) \DIFdelbeginFL \DIFdelFL{0--500-m mixing
ratio}\DIFdelendFL \DIFaddbeginFL \DIFaddFL{0--3-km vertical wind shear; (h) 0--6-km vertical wind
shear}\DIFaddendFL .}
\label{box-whisker_pressure}
\end{figure}

\end{document}

---

## Author Response (AR3)

Dear Piero Lionello,

thank you very much for your comment. Indeed, there is some unnecessary confusion about the use of different data sources in the manuscript. In the revised version, we made clear that the cluster analysis of the large-scale flow was solely based on NCAR/NCEP reanalysis data. ERA-Interim has not been used in the manuscript.

The only exception to the NCAR/NCEP dataset is the analysis of the PV intrusions. Because the resolution of the NCAR/NCEP reanalysis data (2.5 x 2.5 degrees) is too low to analyse small-scale PV intrusions, we have used archived GFS data (horizontal resolution up to 0.25 x 0.25 degrees in recent years). We have clarified this now in the text.

The reference to ERA-Interim data is thus not needed since no results presented in the study are based on it. We will delete the passages in the methods section to avoid confusion. Additionally, we will delete the reference to ERA-Interim data (Dee et al., 2011) in the acknowledgements, where we include the use of GFS data instead.

Changes made in the text

Methods

Page 7, line 1 / Page 4, line 22

We removed the information about the wetter4 tool on page 7 and included it on page 4, line 22:

For the derecho classification, we used maximum wind gusts together with radar data -using the wetter4 visualization tool of MeteoGroup.

Page 6, line 34 - Page 7, line 15

We analysed the large-scale flow in derecho situations indicated by the 500-hPa geopotential charts following Evans and Doswell (2001), Burke and Schultz (2004), and Coniglio and Stensrud (2004). Similar patterns in 500-hPa geopotential field were grouped together using the agglomerative hierarchical clustering of the scipy.cluster python package (Berkhin, 2006; Müllner, 2011). Base fields for the clustering were the NCEP NCAR Reanalysis 2 (available at 00, 06, 12, and 18 UTC$_1$; Kalnay et al., 1996) 500-hPa geopotential height fields. We used the analysis time closest before the start time of the derecho and normalized the fields with respect to their minimum and maximum geopotential heights in the region of interest. The clustering was done in a latitude–longitude box between 20°W–20°E and 40°N–60°N. The distance between two normalized

geopotential height fields is calculated as the sum of the Gaussian distance between the single grid points. Clusters were joined agglomeratively (bottom-up) by calculating the average distance between clusters. We used 500 hPa to analyse synoptic-scale situations to compare the clusters with results from the United States (Coniglio et al., 2004).  Because PV intrusions have been observed in association with cold-season derechos (e.g. Pistotnik et al., 2011), we also analysed fields of potential vorticity (PV) on the 320 K isentropic surface. As the horizontal resolution of the NCAR/NCEP reanalysis (2.5° x 2.5°) data is low, we used archived GFS (horizontal resolution up to 0.25° x 0.25° in recent years) charts of wetter3.de (http://www1.wetter3.de/Archiv/), concentrating on PV intrusions as indicated by sharp PV gradients and large PV values of more than 1.5 PV units (PVU; 1 PVU = 10 Kkg$^{-1}$m$^2$s$^{-1}$); PV was only available for years after 1998.

~~Similar patterns in 500-hPa geopotential field were grouped together using the agglomerative hierarchical clustering of the scipy.cluster python package (Berkhin, 2006; Müllner, 2011). Base fields for the clustering were the NCEP NCAR Reanalysis 2 (available at 00, 06, 12, and 18 UTC$_1$; Kalnay et al., 1996) 500-hPa geopotential height fields. We used the analysis time closest before the start time of the derecho and normalized the fields with respect to their minimum and maximum geopotential~~

~~heights in the region of interest. The clustering was done in a latitude-longitude box between 20 W-20 E and 40 N-60 N. The distance between two normalized geopotential height fields is calculated as the sum of the Gaussian distance between the single grid points. Clusters were joined agglomeratively (bottom-up) by calculating the average distance between clusters. We used 500 hPa to analyse synoptic-scale situations to compare the clusters with results from the United States (Coniglio et al., 2004).~~

**Acknowledgements**

**Page 17, line 2:**

Acknowledgements. We thank the  NCAR for the provision of reanalysis and GFS data.

**References**

**Page 18, lines 27-29:**

[revised manuscript text omitted]